# Elementary 3D organization of active and silenced *E. coli* genome

Alexey A. Gavrilov[1,2,7], Ilya Shamovsky[2,7], Irina Zhegalova[1,6], Sergey Proshkin[3], Yosef Shamovsky[2], Grigory Evko[1], Vitaly Epshtein[2], Aviram Rasouly[2], Anna Blavatnik[2], Sudipta Lahiri[2], Eli Rothenberg[2], Sergey V. Razin[1,4 ✉] & Evgeny Nudler[2,5 ✉]

Unravelling how genomes are spatially organized and how their three-dimensional (3D) architecture drives cellular functions remains a major challenge in biology[1,2]. In bacteria, genomic DNA is compacted into a highly ordered, condensed state called nucleoid[3–5]. Despite progress in characterizing bacterial 3D genome architecture over recent decades[6–8], the fine structure and functional organization of the nucleoid remain elusive due to low-resolution contact maps from methods such as Hi-C[9–11]. Here we developed an enhanced Micro-C chromosome conformation capture, achieving 10-base pair (bp) resolution. This ultra-high-resolution analysis reveals elemental spatial structures in the *Escherichia coli* nucleoid, including chromosomal hairpins (CHINs) and chromosomal hairpin domains (CHIDs). These structures, organized by histone-like proteins H-NS and StpA, have key roles in repressing horizontally transferred genes. Disruption of H-NS causes drastic reorganization of the 3D genome, decreasing CHINs and CHIDs, whereas removing both H-NS and StpA results in their complete disassembly, increased transcription of horizontally transferred genes and delayed growth. Similar effects are observed with netropsin, which competes with H-NS and StpA for AT-rich DNA binding. Interactions between CHINs further organize the genome into isolated loops, potentially insulating active operons. Our Micro-C analysis reveals that all actively transcribed genes form distinct operon-sized chromosomal interaction domains (OPCIDs) in a transcription-dependent manner. These structures appear as square patterns on Micro-C maps, reflecting continuous contacts throughout transcribed regions. This work unveils the fundamental structural elements of the *E. coli* nucleoid, highlighting their connection to nucleoid-associated proteins and transcription machinery.

The development of chromosome conformation capture (3C)[12] and other proximity ligation protocols[13] has substantially advanced the study of 3D genome organization in both eukaryotic and prokaryotic organisms. Although prokaryotes lack a nucleus, their genome is compacted into a structure known as the nucleoid. Chromosomal DNA within the nucleoid undergoes compaction at several levels. Hi-C analysis of various bacteria has revealed the presence of megabase-sized macrodomains and chromatin interaction domains (CIDs)[6]. The mechanisms responsible for the formation of these domains remain poorly understood, although both positive and negative supercoiling appear to have important roles[6,14].

The most typical 3D structures observed in all prokaryotic genomes studied so far are CIDs[6,7]. The size of CIDs ranges from a dozen to several hundred kilobases (kb)[15–17]. These domains often encompass multiple genes and resemble topologically associated domains seen in eukaryotes[18]. The boundaries of CIDs typically contain highly expressed genes, such as rRNA genes[7,19]. It has been proposed that CIDs arise from the interactions of plectonemal DNA structures formed by supercoiling, with highly transcribed genes acting as barriers to supercoiling diffusion[6]. Highly transcribed genes are themselves organized into smaller (1–20 kb) transcriptionally induced domains[8].

Bacterial cells contain various proteins that facilitate genome folding, referred to as nucleoid-associated proteins (NAPs). These include subunits of structural maintenance of chromosome complexes[20,21] and various DNA-binding proteins known as histone-like proteins.

The histone-like proteins, including H-NS, StpA, Fis and Hu, bind to DNA and contribute to chromosomal structure by mediating bridging, wrapping and clustering between adjacent DNA segments[5,22–25]. These bacterial proteins, unlike eukaryotic histones, display varying degrees of sequence specificity. H-NS forms homodimers and oligomers[26] and preferentially binds to conserved AT-rich motifs in DNA, often spreading into adjacent regions[27]. Binding of H-NS typically results in transcriptional repression[28–30] particularly targeting horizontally transferred genes (HTGs) with higher AT content than the rest of the genome[31,32].

[1]Institute of Gene Biology, Russian Academy of Sciences, Moscow, Russia. [2]Department of Biochemistry and Molecular Pharmacology, New York University Grossman School of Medicine, New York, NY, USA. [3]Engelhardt Institute of Molecular Biology, Russian Academy of Sciences, Moscow, Russia. [4]Department of Molecular Biology, Faculty of Biology, M.V. Lomonosov Moscow State University, Moscow, Russia. [5]Howard Hughes Medical Institute, NYU Langone Health, New York, NY, USA. [6]Present address: Institute for Medical Engineering and Science, Massachusetts Institute of Technology, Cambridge, MA, USA. [7]These authors contributed equally: Alexey A. Gavrilov, Ilya Shamovsky. ✉e-mail: sergey.v.razin@gmail.com; evgeny.nudler@nyulangone.org

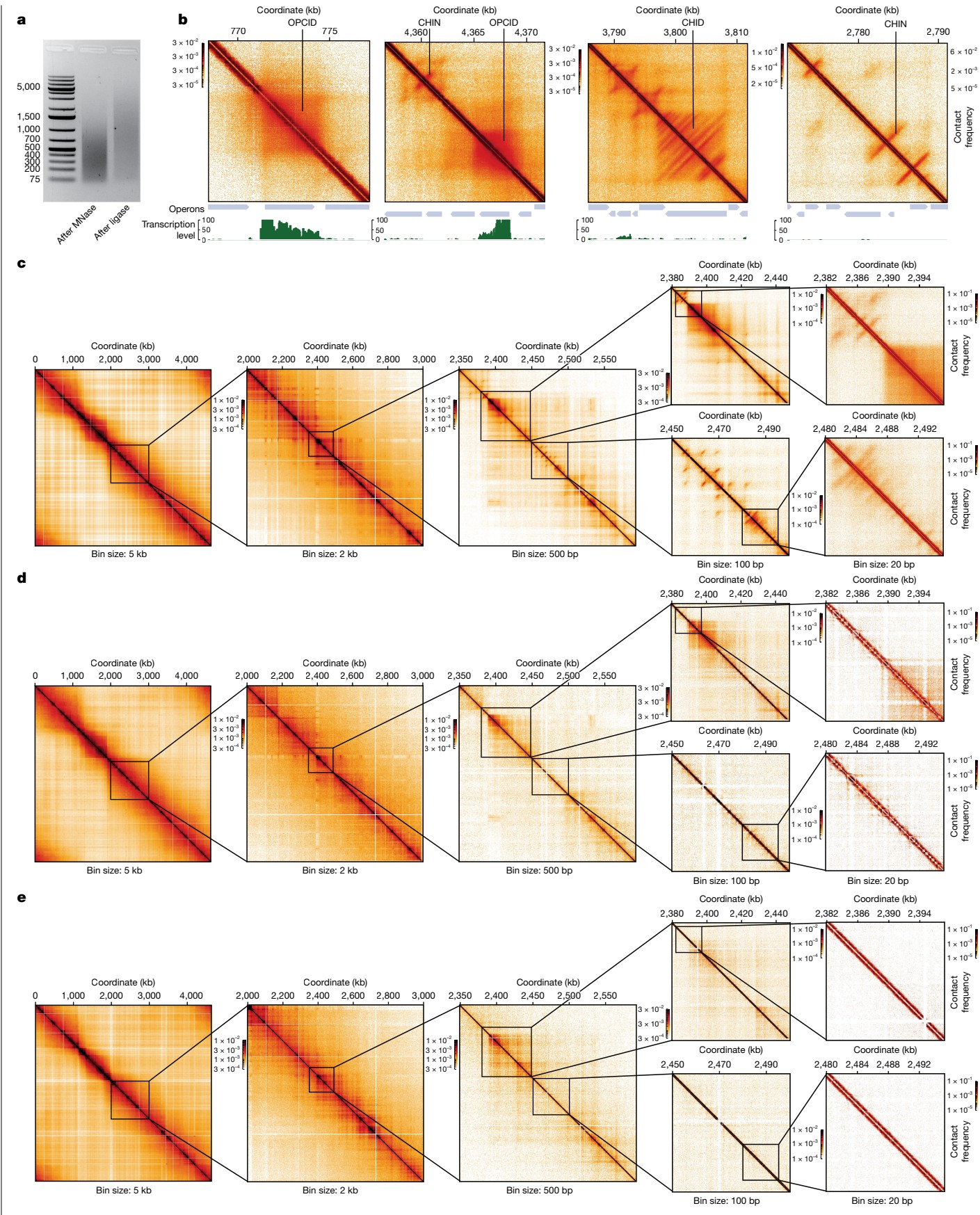

**Fig. 1** | See next page for caption.

**Fig. 1 | Ultra-high-resolution Micro-C maps reveal the complex 3D organization of the E. coli genome. a**, Agarose gel electrophoresis of DNA after different steps of the Micro-C procedure, demonstrating the fragmentation pattern required for ultra-high-resolution mapping (for the uncropped gel, see Supplementary Fig. 1). As *E. coli* lacks nucleosomes, MNase does not produce the characteristic 200-bp DNA ladder but instead generates a smear of fragments. All Micro-C experiments were tested to show similar separation patterns. **b**, Examples of Micro-C contact maps showing OPCIDs, CHINs and CHIDs alongside transcription activity profiles and operon tracks. **c–e**, Comparison between Micro-C and Hi-C data: Micro-C maps generated from WT *E. coli* cells under normal growth conditions (**c**); Hi-C contact maps generated using the standard Hi-C protocol on the same cells (**d**); and previously published[8,55] Hi-C maps for *E. coli* MG1655 (**e**). The left panels show the whole-genome view. Note that reducing bin sizes in Hi-C (below 500 bp) does not significantly enhance resolution, whereas Micro-C maps reveal much finer details, particularly at sub-kilobase resolution.

By contrast, Fis can bind both specifically and non-specifically to DNA[33]. Fis also binds to AT-rich sequences, and depending on cellular conditions, can either activate and suppress transcription[34]. Of note, Fis-binding and H-NS-binding sites are mutually exclusive, suggesting that Fis may compete with H-NS to relieve its repressive effects on gene expression[35,36].

Although the role of NAPs in bacterial nucleoid organization is well known, the specific effect of individual proteins on the 3D genome architecture remains poorly understood. Furthermore, the primary elements of chromosomal organization have yet to be fully characterized. A key challenge has been the limited resolution of Hi-C maps, which even at their best, offer a resolution of 0.5–1 kb. To overcome this limitation, we generated ultra-high-resolution Micro-C maps of the *E. coli* chromosome. Our comprehensive genomic and functional analysis uncovered fundamental 3D features of the active and silenced genome, previously undetected, which are maintained by RNA polymerases (RNAPs) and NAPs, respectively, providing insights into how these chromosomal 3D features affect gene expression and genome stability.

## Elementary features of the *E. coli* 3D genome

Using the Micro-C technique, we generated contact maps of the *E. coli* genome with a resolution of up to 10 bp (Fig. 1a–c and Extended Data Fig. 1a; see Methods). At 5-kb resolution (Fig. 1c), our Micro-C map recapitulates previously reported CIDs[8]. When comparing the higher-resolution Micro-C map with previously published Hi-C maps[8] and those generated in the present study, we observed that the Micro-C map reveals substantially more structural features especially at sub-kilobase resolutions, even with comparable bin sizes (Fig. 1c–e). This observation aligns with previous reports suggesting that, at identical bin sizes, Micro-C provides more details than Hi-C matrices[10,37]. This increased resolution is probably due to the ability of MNase to cleave DNA almost independently of the sequence, resulting in a more uniform distribution of cuts, combined with the improvements in our protocol, which include double crosslinking and the exclusion of detergents (see Methods).

A visual analysis of the Micro-C maps allowed us to identify several previously unrecognized basic features of the 3D structure of the *E. coli* genome:

- OPCIDs: these domains precisely colocalize with highly transcribed operons (Figs. 1b and 2a–c), as confirmed by Red-C analysis, which detects nascent transcripts proximal to cognate transcription units[38] (see Methods).
- CHINs: visible as vertical clusters of contacts; CHINs emerge at or near diagonal of the contact matrix, indicating compact genome folding in non-transcribed regions (Figs. 1b and 2b,c).
- CHIDs: these are composed of multiple CHINs, also located in non-transcribed areas of the genome (Figs. 1b and 2b,c).

OPCIDs preferentially interact with one another, merging into larger domains and creating plaid patterns on heat maps (Extended Data Fig. 2).

## OPCIDs: transcription-driven structures

To explore the relationship between active transcription and the appearance of OPCIDs on operons, we generated Micro-C maps of *E. coli* exposed to heat shock. Activation of heat shock ($\sigma^{32}$) operons at 44 °C led to the formation of OPCIDs at these operons (Fig. 2d–f and Extended Data Fig. 3a). By contrast, transcriptional inhibition of certain $\sigma^{70}$ operons during heat shock resulted in the disappearance of OPCIDs from these operons (Extended Data Fig. 3b–d), whereas active operons unaffected by heat shock continued to form OPCIDs (Extended Data Fig. 3e,f). Of note, CHINs and CHIDs were unaffected by heat shock (Extended Data Fig. 3g,h).

Transcription-dependent 3D structures have been previously reported in the *E. coli* genome[8]. However, unlike operon-colocalizing square contact domains observed in our Micro-C maps, those structures exhibited a different shape being stretched along the diagonal in the Hi-C maps. In addition, the earlier study used an artificial system to demonstrate transcription dependence of those structures, whereas activation of $\sigma^{32}$ operons by heat shock is a natural phenomenon[39]. Unlike transcriptional-dependent bundled domains previously reported[8], OPCIDs exhibited elevated interaction between transcription start sites (TSSs), or promoters, and transcription end sites (TESs), or terminators (Fig. 2g,h). Furthermore, the intensity of TSS–TES contacts increases with higher transcription levels (Fig. 2g,h), suggesting a possibility of rapid RNAP recycling to support sustained high transcriptional output.

We further constructed Micro-C maps of *E. coli* treated with varying doses of the specific RNAP inhibitor rifampicin (Rif). A lower dose of Rif (25 µg ml⁻¹ for 40 min), which suppresses most transcription, caused OPCIDs to disappear, whereas CHINs and CHIDs were largely unaffected. A high dose of Rif (750 µg ml⁻¹ for 40 min), which inhibits all detectable transcription, eliminated most 3D structures except of CHIDs and some prominent CHINs, which became less pronounced but remained detectable on Micro-C maps (Fig. 2i,j and Extended Data Fig. 3i).

## Organization of CHINs and CHIDs

Given that CHINs and CHIDs had not been reported, we sought to characterize these structures in detail. Micro-C maps revealed individual CHINs, clusters of CHINs (CHIDs) and more complex arrangements formed by contacts between individual CHINs (Fig. 3a). To determine whether a specific protein is involved in assembling these structures, we examined chromatin immunoprecipitation followed by sequencing (ChIP–seq) data from *E. coli* strain MG1655 and compared the genomic distribution of various proteins with the manually annotated locations of CHINs and CHIDs (Fig. 3a and Extended Data Fig. 4).

We found that H-NS and its paralogue StpA, which are known to mediate the silencing of specific *E. coli* genes[22,35,40,41], colocalized precisely with CHINs and CHIDs (Fig. 3a,b,f and Extended Data Fig. 4a). These proteins were excluded from transcribed regions across the genome (Fig. 3b and Extended Data Fig. 4a). Consistent with these observations, a motif search using the HOMER tool[42] identified a common motif in the CHIN arms that closely resembles the H-NS-binding site (Fig. 3c). As expected, RNAP was present on OPCIDs and active operons, but absent from CHINs and CHIDs (Extended Data Fig. 4b).

In addition to H-NS and StpA, we found that MukB, a bacterial SMC protein[43], also colocalizes with the arms of CHINs (Fig. 3a), although only a fraction of CHIN arms demonstrates prominent association with this protein (Extended Data Fig. 4c). By contrast, DNA topoisomerase I,

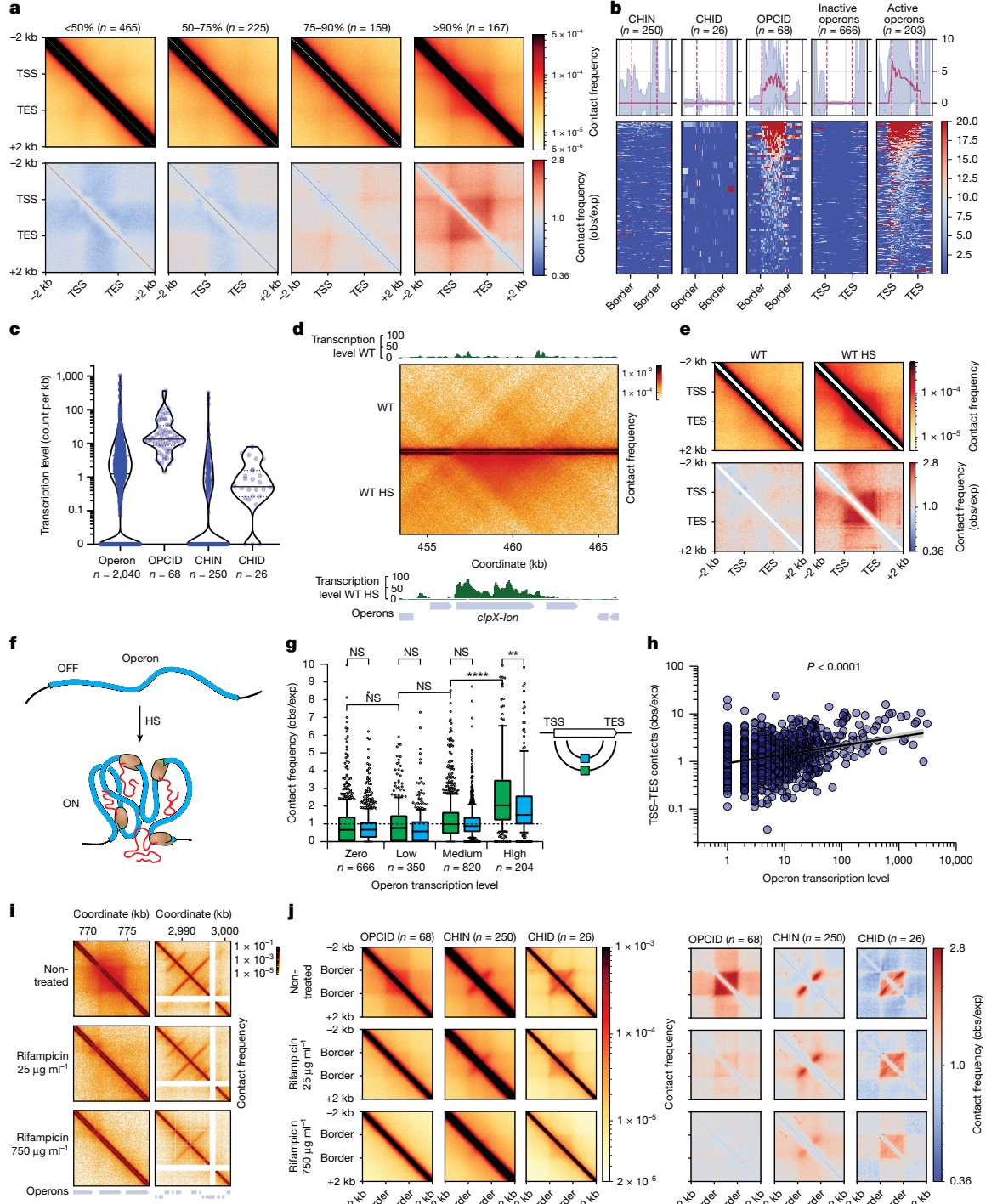

**Fig. 2** | See next page for caption.

DNA gyrases A/B and Fis are notably excluded from CHINs and CHIDs (Fig. 3a and Extended Data Fig. 4c–g). DNA within the stems of CHIN appears to be positively supercoiled, as indicated by preferential binding of GapR, a protein that senses positive supercoils[44], to these regions (Fig. 3a). However, the prominent deposition of GapR is observed only in a small fraction of CHINs, suggesting that the level of supercoiling within CHIN stems is not uniform (Extended Data Fig. 4h).

H-NS and StpA show peaks on CHINs that have a noticeable drop in the centre, suggesting that these proteins interact specifically with the stem of CHINs (Fig. 3a,b and Extended Data Fig. 4a). This observation aligns with the proposed role of H-NS in stabilizing plectonemic

DNA structures in the *E. coli* genome[45]. The loop regions of CHINs are associated with HupA/B (Fig. 3a and Extended Data Fig. 4i,j), proteins known to promote DNA curving[46].

A model illustrating the proposed organization of CHINs is shown in Fig. 3d. CHINs have a medial length of 2 kb, and neighbouring CHINs may spatially interact (Fig. 3a,e). When the loop of CHIN is sufficiently large, the two arms of the stem appear as distinct H-NS peaks in ChIP–seq profiles, whereas the characteristic signal of the stem structure is detected at a distance from the diagonal in Micro-C maps (Fig. 3f). As CHINs and CHIDs localize to non-transcribed regions and the association of H-NS with these structures is unaffected by low doses of Rif

**Fig. 2 | Transcription-driven OPCIDs on active operons and transcription-independent CHINs and CHIDs in repressed genomic regions. a,** Average contact map for operons grouped by transcription activity, based on the Red-C signal. **b,** Transcription level around CHINs, CHIDs, OPCIDs, inactive operons (zero Red-C signal) and active operons (top 10% by Red-C signal). The curves above the piles-up represent median signal, and the shaded areas denote the bootstrap standard error. **c,** Transcription level within specified genomic features, normalized by feature length. Solid and dashed lines represent the median and 25th to 75th percentiles, respectively. **d,** Contact maps for a region harbouring the heat shock (HS) operon *clpX-lon* in normal and HS conditions. **e,** Average contact maps for $\sigma^{32}$ operons under normal and HS conditions. $n = 26$. **f,** Schematic illustrating the assembly of an operon into OPCID on transcription activation under HS conditions. **g,** Frequency of promoter–terminator contacts for operons grouped by Red-C signal into non-transcribed (zero Red-C signal), lowly transcribed (bottom 50% excluding zero), moderately transcribed (50–90%) and highly transcribed (top 10%). $n$ represents the number of operons in each group. Also shown is the frequency of contacts between internal operon regions: one located 20% downstream of the TSS and another 20% upstream of the TES. The boxplots extend from the 25th to 75th percentiles, whiskers are drawn down to the 10th and up to the 90th percentiles, and centre lines represent the median. The $P$ values were calculated using one-way analysis of variance followed by Tukey's multiple comparisons test. $**P < 0.01$ and $****P < 0.0001$. Exp, expected; NS, not significant; obs, observed. The dashed line indicates the obs/exp contact frequency level of 1. **h,** Correlation between operon transcription level and the frequency of promoter–terminator contacts. The black line shows linear regression, and the shaded area denotes 95% CI. **i,** Contact maps for two representative regions harbouring an OPCID and CHINs, shown before and after treatment with 25 µg ml$^{-1}$ or 750 µg ml$^{-1}$ Rif. **j,** Average contact maps for OPCIDs, CHINs and CHIDs before and after exposure to low (25 µg ml$^{-1}$) or high (750 µg ml$^{-1}$) concentrations of Rif, illustrating the effect of transcriptional inhibition on these genomic structures.

(Extended Data Fig. 4k), the reduced intensity of CHINs and CHIDs observed at high doses of Rif may be a secondary effect caused by the eventual depletion of the H-NS pool and other scaffolding proteins.

To validate the existence of CHINs in vivo using an orthogonal approach, we performed proximity ligation assay (PLA) analysis in conjunction with super-resolution microscopy. Four genomic loci were selected for recruitment of dCas9-based PLA probes (Extended Data Fig. 5a,b). One probe (C) was positioned near the base of a long CHIN. A second probe (D) was placed on the opposite shoulder of CHIN, 6 kb away from probe C, such that both probes would be positioned across from each other within the CHIN stem when assembled. A third probe (A) was located 6 kb away from probe C but outside the CHIN region, serving as a negative control. Finally, a fourth probe (B) was placed near probe C to serve as a positive control. We hypothesized that probes C and D, brought into proximity by CHIN folding, would yield a PLA signal, whereas probes A and C, 6 kb apart on linear DNA not involved in CHIN formation, would not. Probes B and C, positioned adjacently on DNA, were expected to yield a PLA signal regardless of 3D organization. The experimental results (Extended Data Fig. 5c,d) confirmed our predictions: although probes A and C produced only sparse PLA signals, probes C and D generated a robust signal in a substantial fraction of cells (approximately 50% of the positive control). These results indicate that the CHIN is formed in at least half of the cells in the examined population within the timeframe of the PLA experiment.

## CHINs and CHIDs are assembled on HTGs

CHINs and CHIDs reside in non-transcribed areas of the genome (Fig. 2b,c) and probably serve as basic repressive structures. However, not all repressed genes reside in these domains. Given their exclusive localization in AT-rich regions (Fig. 3a and Extended Data Fig. 4l) and the established role of H-NS in repressing HTGs[31,47], we conclude that CHINs and CHIDs are predominantly formed on horizontally acquired genes. To test this hypothesis, we compared the location of CHINs (including those within CHIDs) to previously annotated HTGs in the *E. coli* genome[48] and found a strong correlation (Fig. 3h; $P < 0.001$, shuffle test; see also Extended Data Fig. 6a). Moreover, we noted that CHINs frequently colocalize with white crosses on the Micro-C map (Extended Data Fig. 6b), which mark the positions of repetitive elements excluded from the contact matrix. Repetitive elements in prokaryotic genomes are often associated with HTGs[49]. Finally, CHIDs colocalize with HTG clusters (Extended Data Fig. 6c). Collectively, these observations show that CHINs and CHIDs are indeed assembled on HTGs.

## H-NS is the top manager of CHINs

To further elucidate the role of NAPs in the spatial organization of the *E. coli* genome, we generated a panel of knockout strains lacking various NAPs and performed Micro-C analysis (Fig. 3i–n). Successful knockout was confirmed by the absence of the corresponding DNA fragment visible as a white cross on the Micro-C map (Extended Data Fig. 7a). Comparing Δ*hns* cells with wild-type (WT) cells revealed extensive 3D genome reorganization on deletion of *hns*. However, the observed changes were not uniform: although the majority of CHINs (including those within CHIDs) disappeared or weakened, some remained unaffected or even intensified (Figs. 3i and 4j). Of note, spatial contacts between the CHINs that persisted in Δ*hns* cells became more pronounced (Extended Data Fig. 8).

We hypothesized that StpA, an H-NS paralogue, might partially compensate for loss of H-NS.

Indeed, in Δ*hns*Δ*stpA* double knockout cells, CHINs and CHIDs were completely lost (Figs. 3l and 4j and Extended Data Fig. 8), whereas Δ*stpA* alone had no effect (Figs. 3k and 4j and Extended Data Fig. 8). Similarly, knockouts of the gene encoding Fis and MukBEF had no effect on CHINs and CHIDs (Fig. 3m,n). These findings establish H-NS as the principal organizer of CHINs, with StpA providing partial redundancy at specific loci.

To further validate this conclusion, we treated WT cells with netropsin, a drug that disrupts H-NS and StpA binding to AT-rich DNA[50]. Netropsin treatment mimicked the Δ*hns*Δ*stpA* knockout, leading to complete CHIN and CHID disassembly (Fig. 3o).

We also attempted to reconstitute CHINs in vitro by incubating a plasmid containing a DNA fragment known to form CHINs in vivo with purified H-NS. CHIN-like structures of varying lengths readily assembled on this fragment, as observed by transmission electron microscopy, but not on a control plasmid of similar length carrying a non-CHIN-forming DNA fragment or on the CHIN-containing plasmid in the absence of H-NS (Extended Data Fig. 5g–j).

## Disassembly of CHINs activates HTGs

As CHINs are linked to the repression of HTGs, we hypothesized that their loss or reorganization would activate these genes. To test this, we performed transcriptome sequencing of WT, Δ*hns*, Δ*stpA*, Δ*hns*Δ*stpA* cells and WT cells treated with netropsin.

HTGs were preferentially upregulated in Δ*hns*, Δ*hns*Δ*stpA* and netropsin-treated WT cells (Fig. 4a–h and Supplementary Table 1). Full disassembly of CHINs and CHIDs in Δ*hns*Δ*stpA* and netropsin-treated WT cells led to the activation of the vast majority of HTGs (Fig. 4a,b). Among non-HTG genes, upregulated and downregulated genes were approximately equal in number. In Δ*hns* cells, approximately 46% of HTGs were upregulated (Fig. 4c), whereas in Δ*stpA* cells, only approximately 9% of HTGs showed upregulation (Fig. 4d), reinforcing the idea that H-NS is the primary effector of CHIN-mediated HTG repression, with StpA having an important axillary role.

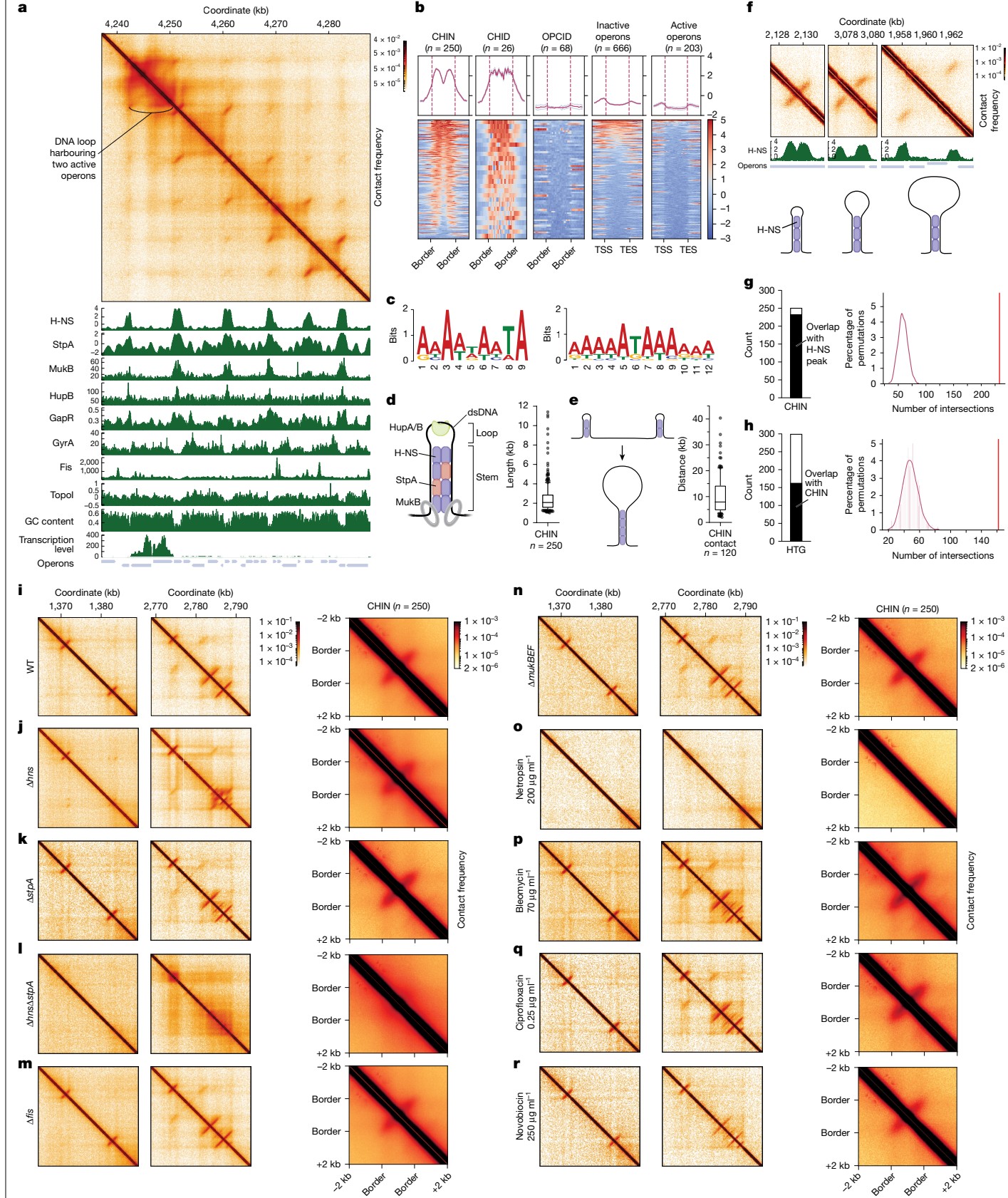

**Fig. 3** | See next page for caption.

We next analysed gene expression separately for genes overlapping and not overlapping with CHINs. In Δ*hns* and Δ*hns*Δ*stpA* cells, the expression level of genes that do not overlap with CHINs in WT cells (either all genes or HTGs) remained largely unchanged. By contrast, genes overlapping with CHINs in WT cells were markedly upregulated (Fig. 4e–h). Furthermore, CHIN-associated genes were

**Fig. 3 | Organization of CHINs and CHIDs directed by H-NS and other NAPs.**
**a**, Example contact map with profiles of NAPs distribution, GC content, transcription signal and operon track. **b**, Distribution of H-NS around CHINs, CHIDs, OPCIDs, inactive operons and active operons. Annotations are consistent with Fig. 2b. **c**, H-NS motif from the PRODORIC database (left) alongside the CHIN motif identified by HOMER (right). **d**, A model depicting the organization of a single CHIN, with CHIN length distribution shown (right; *n* represents the number of CHINs assessed). dsDNA, double-stranded DNA. **e**, A model illustrating the interaction between two CHINs, where they unfold and fuse to form a stem of a large loop. The distribution of distances at which CHINs interact is also shown (right; *n* represents the number of CHIN contacts assessed). The boxplots

more strongly repressed in WT cells than non-CHIN associated genes (Fig. 4e,g).

To further validate these findings, we examined the transcriptional changes of specific genes, including the housekeeping gene *rpoC* and HTGs (*yadC*, *pgaC*, *mcbR* and *ygeH*) using quantitative PCR with reverse transcription (RT–qPCR). The expression level of *rpoC* remained largely unchanged across all mutant strains tested, whereas *yadC*, *pgaC*, *mcbR* and *ygeH* were upregulated to varying degrees in Δ*hns* and Δ*hns*Δ*stpA* strains, but not in the Δ*stpA* strain (Fig. 4i). Of note, the pronounced upregulation of *yadC*, *pgaC* and *mcbR* in Δ*hns* and Δ*hns*Δ*stpA* strains correlated with CHIN (CHID) disassembly and the formation of OPCIDs in their place (Fig. 4j). In the case of *ygeH*, the CHID was reorganized but not fully disassembled in the Δ*hns* strain, leading to a relatively modest (approximately fivefold) upregulation. However, complete disassembly of this CHID in Δ*hns*Δ*stpA* cells resulted in a nearly 150-fold upregulation of *ygeH* and the formation of OPCID in place of CHID. Similarly, the HTG *yfjW*, which colocalizes with a CHIN in coordinate 2,773–2,775 kb (Fig. 3i), remains transcriptionally silent in Δ*hns* cells (Supplementary Table 1) where this CHIN is preserved (Fig. 3j). However, its expression increases approximately 50-fold in Δ*hns*Δ*stpA* cells (Supplementary Table 1) where the CHIN is replaced with an OPCID (Fig. 3l). These findings confirm that CHINs function as repressive structures and highlight that CHIN assembly, rather than H-NS deposition alone, is crucial for repression.

## CHINs are superhelicity independent

To determine whether the formation of CHINs and CHIDs depends on DNA superhelicity, we treated WT cells with bleomycin, a drug that introduces single-stranded breaks in DNA and relieves supercoiling[51], followed by Micro-C analysis. The profiles of CHINs and CHIDs remained unchanged after bleomycin treatment (Fig. 3p).

We then tested whether inhibiting DNA gyrase affects CHINs and CHIDs structures. Treatment with ciprofloxacin, which induces gyrase-stalled complexes on DNA, or novobiocin, a catalytic inhibitor of DNA gyrase (both used at 10× the minimum inhibitory concentration), did not cause notable changes in the CHIN and CHID profiles (Fig. 3q,r). These results indicate that the existence of CHINs and CHIDs is independent of DNA supercoiling.

## Discussion

Here we present a Micro-C map of the bacterial genome, achieving resolution up to 10 bp. This map uncovers several previously unrecognized spatial structures. The critical ability to observe these intricacies hinges on the proximity ligation step in the Micro-C protocol (Extended Data Fig. 9a), confirming that these structures indeed represent spatial interactions among distant genomic elements. Although traces of these structures are detectable in Hi-C maps (Extended Data Fig. 1b), the higher resolution of our Micro-C maps offers more detailed insights (Fig. 1). This improved resolution may arise from the more uniform distribution of DNA cuts introduced by MNase, along with the absence of detergent treatment of the cells in the Micro-C protocol (see Methods).

The most prominent structures identified in this study are CHINs and the larger CHIDs, which are primarily assembled on HTGs and appear to have a central role in their repression. The arms of CHINs are associated with H-NS and StpA, NAPs known to form both homo-oligomers and hetero-oligomers[41]. These interactions probably stabilize the hairpin stems and contribute to the ability of CHINs to form various higher-order spatial structures (Fig. 5a,b). Moreover, spatial contacts between CHINs create DNA loops capable of isolating active operons from the rest of the genome (Fig. 3a). It should be noted, however, that Micro-C analysis does not capture the frequency or duration of these contacts across individual cells; interactions between H-NS-bound loci may occur only in a subset of the population at any given time. Nevertheless, the existence of CHINs was independently validated using PLA, which confirmed that the CHIN base regions identified by Micro-C are indeed in close spatial proximity in a substantial proportion of cells (Extended Data Fig. 5a–d).

The essential role of histone-like nucleoid proteins H-NS and StpA in maintaining the structure of CHINs and CHIDs is evident from the drastic reorganization or complete disassembly of these structures in Δ*hns* and Δ*hns*Δ*stpA* strains, respectively (Fig. 3j,l). The absence of CHINs and CHIDs is also observed when cells are treated with netropsin, an antibiotic that competes with H-NS and StpA for DNA-binding sites[50]. Of note, single knockout of *stpA* does not affect the integrity of CHINs and CHIDs, which led us to consider H-NS as the primary regulator of these structures. The central role of H-NS in CHIN formation was further supported by in vitro reconstitution experiments, in which CHIN-like structures readily assembled upon incubating purified H-NS with a plasmid containing a DNA fragment known to form CHIN in vivo (Extended Data Fig. 5g–j). Other NAPs, such as MukBEF and Fis, as well as DNA supercoiling, do not appear to be necessary for the assembly of CHINs and CHIDs.

The disassembly of CHINs and CHIDs correlates with the derepression of previously silenced genes, particularly HTGs (Fig. 4a–j). This derepression results in a growth delay (Fig. 4k), probably due to the toxicity associated with many HTGs[52]. In many cases, this derepression is manifested by the emergence of new OPCIDs at operons (Figs. 3l and 4j and Extended Data Fig. 8). The contact probability within the 10–100-kb range increases in Δ*hns* and Δ*hns*Δ*stpA* strains (Extended Data Fig. 7g). This increase can be attributed to enhanced clustering of active genomic regions − a phenomenon not observed when OPCIDs and CHINs were disrupted by transcription inhibition with Rif, which resulted in decreased long-range contacts (Extended Data Fig. 7f).

It is worth noting that all CHINs within a CHID cannot coexist simultaneously at the same locus owing to overlapping regions. Therefore, CHIDs represent a population average of alternative CHINs formed in individual cells. Some of the other complex interaction patterns observed in our Micro-C maps may also reflect population averages. The observation that all CHINs within CHIDs fit neatly into a contact domain on the Micro-C map implies the presence of boundaries restricting CHIN formation and/or the deposition of H-NS–StpA. These boundaries probably arise from sharp differences in GC content between the CHID

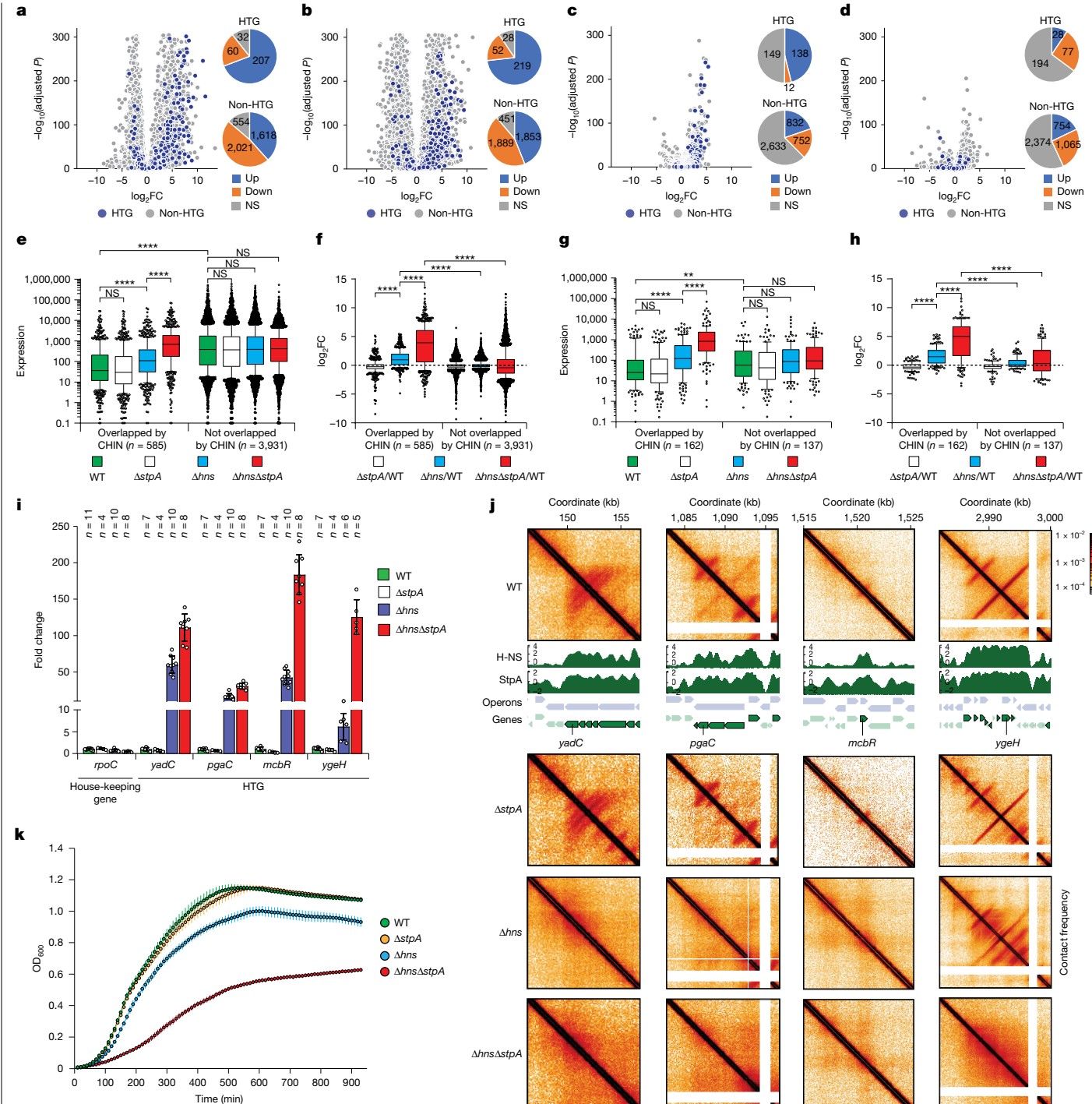

**Fig. 4 | Activation of HTGs on H-NS and double *hns stpA* knockout. a–d,** Gene expression changes following deletion of *hns* and *stpA* (**a**), exposure to 200 µg ml⁻¹ netropsin (**b**), deletion of *hns* (**c**) and deletion of *stpA* (**d**) as determined by RNA sequencing. The distribution of HTGs and non-HTGs that are upregulated, downregulated or insignificantly altered in Δ*hns*, Δ*stpA*, Δ*hns*Δ*stpA* and netropsin-treated *E. coli* cells is also shown. The *P* values were calculated using two-sided Wald test and adjusted for multiple testing using Benjamini–Hochberg correction with false discovery rate set to 0.05. FC, fold change. **e,** Expression levels of genes overlapping with CHINs and those not overlapping with CHINs in WT *E. coli* cells (green) compared with the expression levels of the same gene groups in Δ*hns* (blue), Δ*stpA* (white) and Δ*hns*Δ*stpA* (red). **f,** Expression changes of CHIN-associated and CHIN-non-associated genes in response to Δ*hns*, Δ*stpA* and Δ*hns*Δ*stpA*. **g,h,** Similar to panels **e** and **f**, but focusing specifically on HTGs.

The number of genes (*n*) is indicated. The boxplots extend from the 25th to 75th percentiles, whiskers are drawn down to the 10th and up to the 90th percentiles, and the centre lines represent the median. The *P* values were calculated using the two-sided Kolmogorov–Smirnov test: **\**P* < 0.01 and **\*\**P* < 0.0001. The black dashed line indicates the fold change value of 1. **i,** Changes in expression of selected HTGs upon Δ*hns*, Δ*stpA* and Δ*hns*Δ*stpA*, as determined by RT–qPCR. Signals are normalized to 16S rRNA, with the expression level in WT cells set to 1. Data are presented as mean ± s.d. **j,** Micro-C contact maps of regions harbouring HTGs selected for RT–qPCR analysis in WT, Δ*stpA*, Δ*hns* and Δ*hns*Δ*stpA* cells. **k,** Growth rates of WT, Δ*stpA*, Δ*hns* and Δ*hns*Δ*stpA* *E. coli* cells measured using Bioscreen. Data are presented as mean ± s.d. from *n* = 7 biological replicates for WT, Δ*stpA* and Δ*hns*, and *n* = 14 for Δ*hns*Δ*stpA*. OD₆₀₀, optical density at 600 nm.

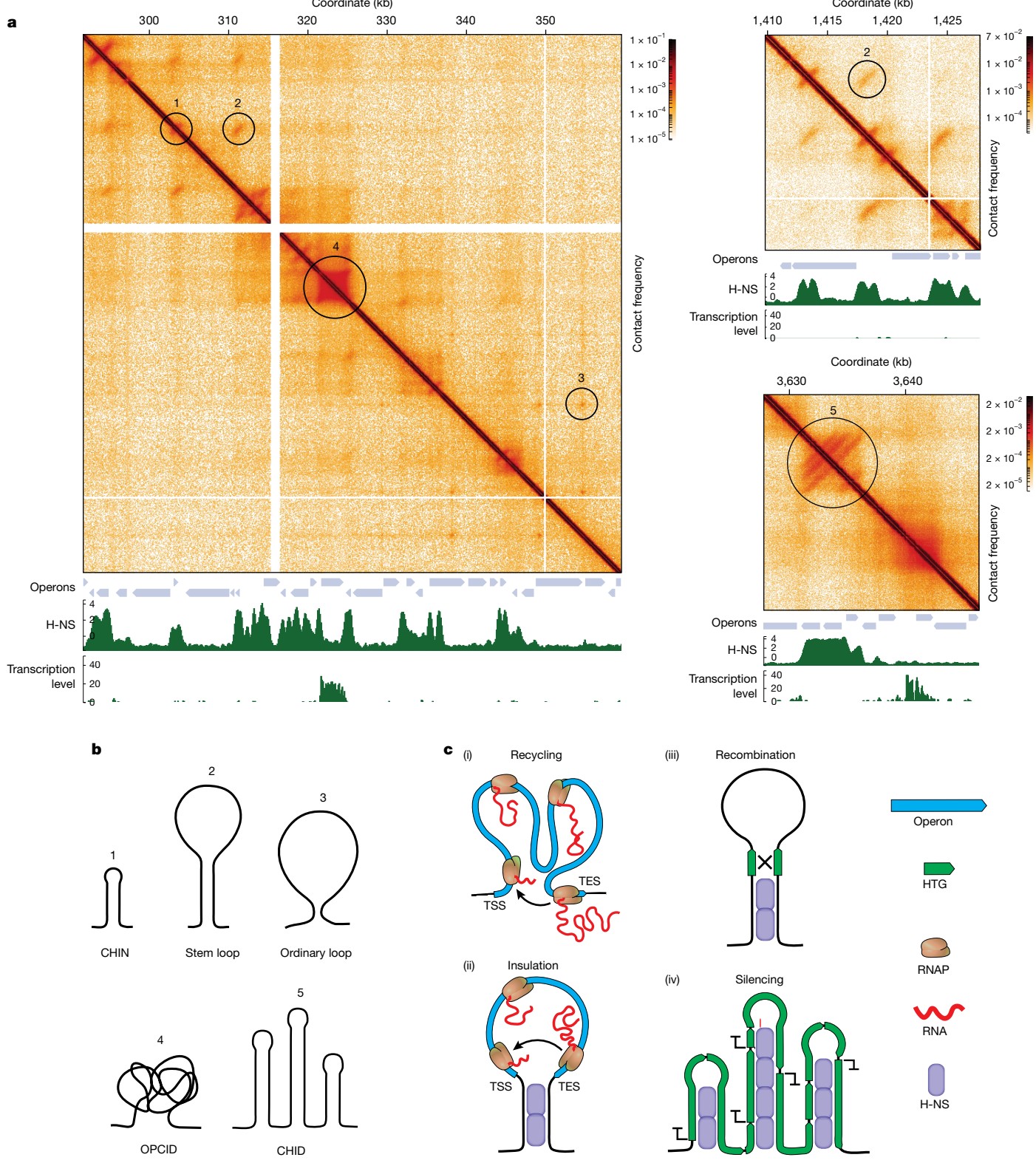

**Fig. 5 | Elementary and higher-order structures captured by Micro-C in the *E. coli* genome. a**, Example contact maps from WT *E. coli* cells showing various contact patterns. **b**, Interpretation of the different contact patterns. **c**, OPCIDs and CHIN networks, as mapped in this study, can facilitate RNAP recycling by bringing promoters (TSSs) and terminators (TESs) into close proximity (i and ii). They can also establish spatial insulation of active operons (ii), enable recombination between distant HTGs (iii) and promote HTG silencing (iv).

and its flanking regions (Extended Data Fig. 4l). As both H-NS and StpA preferentially recognize AT-rich motifs, their dense binding should be confined to these AT-rich areas.

Previous studies have shown that the *E. coli* genome exhibits a distinct spatial organization linked to transcription[5,6,8]. However, even the highest resolution of Hi-C analysis (500 bp)[8] has been insufficient

for revealing basic 3D structures or correlating them with individual operons across the genome. Our ultra-high-resolution Micro-C analysis has uncovered operon-linked OPCIDs that form in a strictly transcription-dependent manner. Unlike previously described bundled domains[8], OPCIDs are characterized by typical contact domains (squares on contact maps) where all regions are in contact with one another. OPCIDs probably represent a superposition of microloops and coiled-coil structures that develop within transcribed regions of individual cells. The integrity of OPCID was not compromised by bleomycin, a drug that introduces single-strand breaks in DNA and thus releases DNA supercoiling (Extended Data Fig. 3j).

The high frequency of direct promoter–terminator interactions within an OPCID (Fig. 2g) might facilitate the recycling of RNAP (Fig. 5c), thereby enhancing transcription efficiency. The relationship between OPCIDs and bundled domains[8] remains to be fully elucidated. Currently, we cannot rule out the possibility that the Hi-C protocol previously used[8] may fail to detect all interactions within operons owing to the use of detergents that could disrupt liquid condensates.

In conclusion, we have demonstrated that the spatial organization of the *E. coli* genome is far more intricate than previously recognized, being governed by both transcription activity and gene silencing. Key features of the *E. coli* 3D genome include operon-linked OPC-IDs, which form in a strictly transcription-dependent manner, and a chromosomal hairpin network (CHINs and CHIDs) that assemble on silenced HTGs. The ability of CHINs to stick together may facilitate recombination between HTGs in both *cis* and *trans* (Fig. 5c), explaining the previously observed high recombination frequency between these genes[53,54] and suggesting an elegant mechanism that accelerates bacterial evolution.

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

# Methods

## Cell strains

The *E. coli* K-12 MG1655 strain was grown in Luria–Bertani medium at 37 °C with shaking at 200 rpm until reaching an $OD_{600}$ of approximately 0.6. For heat shock treatment, cells were incubated in a water bath at 44 °C for 10 min. For transcription inhibition, cells were treated with 25 or 750 µg ml$^{-1}$ Rif for 40 min at 37 °C and 200 rpm. To induce DNA damage, cells were exposed to 70 µg ml$^{-1}$ bleomycin for 10 min at 37 °C and 200 rpm. For DNA gyrase inhibition, cells were treated with 0.025 or 0.25 µg ml$^{-1}$ ciprofloxacin or 25 or 250 µg ml$^{-1}$ novobiocin for 15 min at 37 °C and 200 rpm. For netropsin treatment, cells were grown to an $OD_{600}$ of approximately 0.4, followed by the addition of netropsin to a final concentration of 20 or 200 µg ml$^{-1}$ and incubation for 30 min at 37 °C and 200 rpm.

## Construction of deletion mutants

The *E. coli* Δ*hns*-knockout strain was generated using a modified λ-Red phage-assisted recombineering approach[56]. The phage λ *att-L* and *att-R* site-flanked chloramphenicol (Cm) resistance *cat* gene[57] was amplified by PCR using primers with 36-nucleotide extensions homologous to the upstream and downstream chromosomal sequences of *hns*. The purified PCR product was electroporated into *E. coli* MG1655 cells carrying the pKD46 plasmid, which encodes the λ-Red recombination genes *bet*, *exo* and *gam*. To induce λ-Red genes, 50 mM L-arabinose was added to SOB cultures with ampicillin (Ap), and cells were grown to an $OD_{600}$ of approximately 0.5 at 30 °C. Electrocompetent cells were prepared using the glycerol–mannitol density step centrifugation method[58]. Recombinant *hns::Cm*$^r$ cells were selected on LB agar plates containing Cm and verified by PCR with specific primers. The pKD46 plasmid was cured by growth at 37 °C, and Cm$^r$Ap$^s$ clones were selected. To eliminate the *cat* gene flanked by λ *att-L* and *att-R* sites, mutants were transformed with the pInt/Xis plasmid, which encodes the λ Int/Xis-dependent recombination system[57,59]. Ap-resistant transformants were selected at 30 °C and verified by PCR for the loss of the *cat* gene and the deletion of the *hns* gene. The clones were colony purified at 37 °C and tested for Ap and Cm sensitivity to confirm the loss of both the helper plasmid and the *cat* gene.

*E. coli* MG1655 *fis::Km*$^r$ and *stpA::Km*$^r$ deletion mutants were generated by transducing the corresponding alleles from the Keio collection[60] into MG1655 using phage P1*vir*, followed by selection for kanamycin resistance. To eliminate FRT-flanked Km$^r$ cassette from the *fis::Km*$^r$ strain, the pCP20 plasmid[56] was used. Ampicillin-resistant transformants were selected at 30 °C, then purified without selection at 37 °C to cure the helper plasmid. Successful removal of all antibiotic resistance markers was confirmed.

The *stpA::Km*$^r$ region was moved into the Δ*hns* strain by P1 transduction, generating the double knockout strain through selection for kanamycin.

The *muk(FEB)::Cm*$^r$ strain, in which all three genes of the *mukF–mukE–mukB* operon were knocked out, was constructed using the same method as for the *hns* deletion mutant. A relevant pair of primers with 36-nucleotide extensions homologous to the start of *mukF* and the end of *mukB* was used. Owing to the temperature-sensitive phenotype resulting from *muk(FEB)* inactivation, all incubations were performed at 20–22 °C. After selection for Cm$^r$, the pKD46 was cured from the strain by overnight growth in LB medium without antibiotics at 20 °C, followed by plating on LB agar with Cm. Colonies were screened for ampicillin sensitivity and verified by PCR to confirm insertion of the Cm$^r$ cassette in place of the *muk(FEB)* genes. The resulting strain forms colonies only at 22 °C and fails to grow at 30 °C or 37 °C.

## Growth curves

Individual colonies of *E. coli* strains were cultured in LB medium to the stationary phase and then diluted with fresh LB medium to an $OD_{600}$ of approximately 0.08. Cultures were grown at 37 °C until reaching an $OD_{600}$ of approximately 0.4. A 150 µl aliquot of each culture, further diluted to an $OD_{600}$ of approximately 0.01, was transferred into honeycomb wells and incubated at 37 °C with vigorous shaking using the Bioscreen C automated growth analyzer. $OD_{600}$ values were recorded automatically every 10 min, and growth curves were plotted based on the collected data.

## Micro-C

Micro-C was adapted from ref. 11. *E. coli* cell cultures (2–3 ml) were transferred to PBS-diluted formaldehyde (28906, Thermo Fisher) to achieve a final formaldehyde concentration of 3% and a total volume of 13.5 ml. Fixation was performed for 30 min at room temperature with gentle rotation, followed by quenching with 375 mM Tris (pH 7.5).

Previous studies have shown that H-NS bridging is temperature dependent, with maximum bridging observed below 30 °C (ref. 61). To determine whether the detection of H-NS-dependent features in Micro-C contact maps depends on cultivation or fixation temperature, we compared contact maps for cells cultivated and fixed at room temperature, cells cultivated and fixed at 37 °C, and cells cultivated at 37 °C but fixed at room temperature following the standard protocol. No differences in the intensity of CHINs were observed between the three conditions (Extended Data Fig. 9b–d).

Following formaldehyde crosslinking, cells were centrifuged at 3,220*g* for 20 min at 4 °C, resuspended in 1 ml of cold PBS and centrifuged again at 16,100*g* for 5 min at 4 °C. The cell pellet was resuspended in 205 µl PBS and mixed with an equal volume of PBS-diluted disuccinimidyl glutarate (DSG) (20593, Thermo Fisher; used as 300 mM stock in DMSO, freshly made) to a final DSG concentration of 2 mM. Fixation was carried out for 45 min at room temperature with rotation, followed by quenching with 375 mM Tris (pH 7.5).

To assess whether DSG crosslinking is effective in *E. coli*, we compared it with disuccinimidyl suberate (DSS), a cell-permeable crosslinker commonly used in *E. coli* studies[62,63]. Both crosslinkers produced a similar number of protein–protein crosslinks in *E. coli* cells (Extended Data Fig. 9e). In addition, the inclusion of DSG was essential for stabilizing Micro-C material, as MNase treatment of cells fixed with formaldehyde alone resulted in the solubilization of all DNA fragments (Extended Data Fig. 9f).

After DSG crosslinking, cells were centrifuged at 16,100*g* for 5 min at 4 °C, washed with 1 ml cold PBS and centrifuged again. Cells were resuspended in 250 µl of TE buffer (10 mM Tris pH 8.0 and 1 mM EDTA) supplemented with 1× protease inhibitors (Bimake). Lysozyme (L6876, Sigma) was added to a final concentration of 4 µg ml$^{-1}$, and the mixture was incubated at 37 °C with shaking at 1,000 rpm for 30 min. Cells were then pelleted at 16,100*g* for 5 min at room temperature and resuspended in MNase buffer (10 mM Tris pH 7.5 and 1 mM CaCl$_2$). MNase (EN0181, Thermo Fisher; used as 2 or 5 U µl$^{-1}$ stock in 20 mM Tris pH 7.5, 50 mM NaCl and 50% glycerol) was added and samples were incubated for 25 min at 37 °C with shaking at 1,000 rpm. The reaction volume ranged from 475 µl to 1,160 µl, and MNase amounts varied from 3 U to 12 U, with conditions optimized individually for each sample to obtain DNA fragments ranging from approximately 75 to 1,000 bp (Extended Data Fig. 9g).

Samples showing excessive DNA degradation, characterized by reduced DNA yield, were typically discarded (for example, sample 4 rep1 in Extended Data Fig. 9h). However, even heavily overdigested samples retained characteristic Micro-C contact patterns (Extended Data Fig. 9i), demonstrating the robustness of the procedure despite varying levels of MNase digestion.

DNA digestion was terminated by adding EGTA to 5 mM followed by centrifugation for 5 min at 16,100*g* at room temperature. The pellet was resuspended in 220 µl 1× PNK buffer (NEB), and 20 µl was set aside and subjected to DNA purification for MNase digestion control (Fig. 1a). The remaining sample was centrifuged for 5 min at 16,100*g* at room

temperature and resuspended in 145 µl 1.03× PNK buffer (NEB) followed by adding 5 µl PNK (10 U µl⁻¹, NEB) to dephosphorylate DNA 3′ ends. The mixture was incubated for 30 min at 37 °C with shaking at 900 rpm and centrifuged as above. The sample was resuspended in 90 µl solution composed of 10 µl 10× NEB buffer 2.1, 2 µl 100 mM ATP, 5 µl 100 mM dithiothreitol and 73 µl water. The mixture was supplemented with 5 µl PNK (10 U µl⁻¹, NEB) and incubated for 15 min at 37 °C with shaking at 900 rpm to phosphorylate DNA 5′ ends. The mixture was then supplemented with 5 µl Klenow (5 U µl⁻¹, NEB) and incubated for 15 min at 37 °C with shaking at 900 rpm to generate 3′ recessed DNA ends in the absence of dNTPs. The 3′ recessed DNA ends were labelled with biotin by adding 50 µl solution composed of 5 µl 10× T4 DNA ligase buffer (NEB), 10 µl 1 mM biotin–dATP, 10 µl 1 mM biotin–dCTP, 1 µl 10 mM dGTP, 1 µl 10 mM dTTP, 0.25 µl BSA (10 mg ml⁻¹, NEB) and 22.75 µl water. The mixture was incubated for 45 min at 25 °C with shaking at 900 rpm and centrifuged as above.

The pellet was resuspended in 95 µl 1.05× T4 DNA ligase buffer (NEB) followed by adding 5 µl PNK (10 U µl⁻¹, NEB) and incubating for 1 h at 37 °C with shaking at 900 rpm to ensure complete phosphorylation of DNA 5′ ends. The sample was pelleted as above and resuspended in 478 µl 1.02× T4 DNA ligase buffer (Thermo Fisher) followed by adding 12.5 µl T4 DNA ligase (5 Weiss U µl⁻¹, Thermo Fisher). DNA proximity ligation was carried out overnight at 22 °C with rotation, and the mixture was centrifuged as above. The pellet was resuspended in 148.5 µl 1.01× NEB buffer 1 followed by adding 1.5 µl exonuclease III (100 U µl⁻¹, NEB) and incubating for 15 min at 37 °C with shaking at 900 rpm to remove biotin from non-ligated ends.

To reverse crosslinks, the sample was supplemented with 125 µl solution composed of 15 µl 10× NEB buffer 2, 4.5 µl 5 M NaCl, 30 µl 10% SDS and 75.5 µl water followed by adding 25 µl proteinase K (20 mg ml⁻¹, Ambion) and incubating for 1 h at 55 °C and then for 7 h at 65 °C. The sample was extracted with an equal volume of phenol:chloroform:isoamyl alcohol (25:24:1), pH 8.0, followed by centrifugation for 5 min at 12,000g and room temperature. The water phase was supplemented with 1/10 volume 3 M NaAc, 3 µl glycogen (20 mg ml⁻¹, Thermo Fisher) and 2.5 volumes 96% ethanol. The sample was incubated for 30 min at room temperature and centrifuged for 30 min at 21,000g and 4 °C. The pellet was washed with 1 ml cold 70% ethanol and centrifuged for 10 min at 21,000g and 4 °C. The pellet was air dried and diluted in 45 µl 10 mM Tris pH 8.0. RNase A (10 mg ml⁻¹, Thermo Fisher) was added to 40 µg ml⁻¹ and the sample was incubated for 30 min at 37 °C. A 5 µl aliquot was set aside for ligation control (Fig. 1a). The remaining sample was further purified with 2 volumes of AMPure XP beads (Beckman Coulter) and finally eluted into 40 µl 10 mM Tris pH 8.0 followed by measuring DNA concentration with the Qubit dsDNA broad range kit. A typical DNA yield was 1–2 µg.

The sample was brought to 250 µl by water followed by adding 250 µl 2× sonication buffer (100 mM Tris pH 8.0, 20 mM EDTA and 0.2% SDS). The sample was sonicated on ice with five 30-s pulses followed by 3-min rest periods using a VirTis VirSonic 100 sonicator at high power (setting 15). The sample was passed through Amicon 30 K Ultra-0.5-ml centrifugal filters (Millipore) by centrifugation for 5 min at 16,100g and 4 °C. Then, 450 µl 10 mM Tris pH 8.0 was added into the filter and centrifugation was repeated. Another 450 µl 10 mM Tris pH 8.0 was added into the filter and centrifugation was repeated. The concentrated sample was removed from the filter, brought to 100 µl by 10 mM Tris pH 8.0 and subjected to biotin pull-down.

For this process, 10 µl Dynabeads MyOne Streptavidin C1 beads (10 mg ml⁻¹, Thermo Fisher) was washed twice with 400 µl Tween washing buffer (TWB; 5 mM Tris pH 7.5, 0.5 mM EDTA, 1 M NaCl and 0.05% Tween 20) by repeating the resuspension–magnet separation. Streptavidin beads were resuspended in 100 µl 2× binding buffer (10 mM Tris pH 7.5, 1 mM EDTA and 2 M NaCl) and mixed with sonicated DNA followed by incubation for 15 min at room temperature. Streptavidin beads with tethered DNA fragments were washed twice

with 600 µl TWB, once with 100 µl 1× NEB buffer 2 and once with 100 µl 1× T4 DNA ligase buffer (NEB), and resuspended in 75 µl end repair solution containing 1× T4 DNA ligase buffer (NEB), 0.25 mM each dNTP, 5 µl PNK (10 U µl⁻¹, NEB), 1 µl Klenow (5 U µl⁻¹, NEB) and 4 µl T4 DNA polymerase (3 U µl⁻¹, NEB). The mixture was incubated for 30 min at 22 °C. Streptavidin beads were washed twice with 600 µl TWB and twice with 100 µl 1× NEB buffer 2, and resuspended in 95 µl 1.05× NEB buffer 2 supplemented with 0.21 mM dATP. The mixture was supplemented with 5 µl Klenow exo-minus (5 U µl⁻¹, NEB) and A-tailing was carried out for 40 min at 37 °C. Streptavidin beads were washed twice with 600 µl TWB, once with 100 µl 1× NEB buffer 2 and once with 100 µl 1× T4 DNA ligase buffer (Thermo Fisher), and resuspended in 46.5 µl 1.08× T4 DNA ligase buffer (Thermo Fisher). The mixture was supplemented with 2.5 µl Illumina TruSeq single index adapter and 1 µl T4 DNA ligase (5 Weiss U µl⁻¹, Thermo Fisher) and incubated for 2.5 h at 22 °C with occasional mixing. After ligation, streptavidin beads were washed twice with 600 µl TWB, once with 100 µl 1× NEB buffer 2 and once with 100 µl 10 mM Tris pH 8.0, and resuspended in 16 µl water.

DNA was amplified in 50 µl PCR containing 1× KAPA HiFi Fidelity Buffer, 0.3 mM each dNTP, 0.5 µM Illumina forward primer, 0.5 µM Illumina reverse primer, 1 U KAPA HiFi DNA polymerase and 4 µl streptavidin beads from the above step. PCR was performed as follows: 95 °C for 5 min, [98 °C for 20 s, 65 °C for 15 s and 72 °C for 20 s] × 8–10 cycles, and 72 °C for 3 min. For ligase minus control, 12 PCR cycles were used. PCR products of two reactions were pooled and purified twice with 1.2 volumes of AMPure XP beads. Purified PCR products were paired-end sequenced on the Illumina NovaSeq or BGI DNBSEQ-T7 platforms.

## Micro-C read processing

Sequencing reads were processed using distiller-nf (v0.3.4). Mapping was performed using BWA with default pipeline parameters to the reference genome NC_000913.3. Statistics of read filtering and mapping are presented in Supplementary Table 2.

The generation of a large number of reads (approximately 1.5 billion for main samples) and the relatively small size of the *E. coli* genome enabled us to construct DNA–DNA contact maps with a resolution of up to 10 bp. At this resolution, CHINs remained detectable in contact maps; however, further increases in resolution resulted in the structures becoming less discernible (Extended Data Fig. 1a).

The sample correlation was computed using HiCRep[64] with a smoothing parameter $h = 30$ and resolution of 100 bp.

Scalings were calculated using the expected *cis* module from cooltools (v0.7.0)[65], using the following parameters: smooth = True, aggregate_smoothed = True. Contacts at distances less than 100 bp were excluded. The resulting plot was generated using double logarithmic coordinates.

## Operons

We use operons from RegulonDB[66] (Supplementary Table 3). The information on sigma factors was extracted from the annotated TSS dataset from RegulonDB. The level of transcription activity was inferred from the Red-C data (see below).

In the piles-up analysis (Figs. 2b and 3b and Extended Data Fig. 4), we used the top 10% active operons and operons with zero Red-C signal (inactive operons). In the average contact map analysis presented in Fig. 2a, operons were divided into four groups by the Red-C signal (bottom 50%, 50–75%, 75–90% and top 10%) followed by selecting only operons with the length higher than the median length of all operons (1,343 bp) to reduce the contribution of the Micro-C signal near the diagonal. In the average contact map analysis of σ³² operons (Fig. 2e), we also used operons longer than 1,343 bp (26 out of 44). In the analysis of promoter–terminator contacts presented in Fig. 2g, operons were grouped by transcription activity, determined by the Red-C signal normalized to feature length.

## Annotation of CHINs, CHIDs and OPCIDs

Manual annotation of CHINs, CHIDs and OPCIDs was performed on the contact map for WT *E. coli* cells under normal growth conditions (37 °C). The contact map was examined using HiGlass browser in 'blind' mode, with no operons or other tracks displayed to avoid bias during the search. In annotating CHINs and CHIDs, we aimed to identify as many instances as possible, whereas only a representative fraction of OPCIDs was annotated. We also focused on identifying CHIN contacts by searching for elongated spots located away from the diagonal. The coordinates of the annotated OPCIDs, CHINs, CHIDs and CHIN contacts can be found in Supplementary Tables 4–7.

To validate the accuracy of the manual annotation of CHINs, we also used the Chromosight program[67] for annotation (Supplementary Table 8). We found a good correlation between the positions of the manually annotated and Chromosight-annotated CHINs (Extended Data Fig. 1e,f). The discrepancies between the two sets may be attributed to false positives and false negatives present in the Chromosight dataset (Extended Data Fig. 1e). Of note, CHINs identified by Chromosight also exhibited the characteristic bimodal distribution of H-NS around them (Extended Data Fig. 1g).

## Average contact map

To create an average contact map, we utilized coolpuppy (v1.1.0)[68] with the following parameters: local = True, rescale = True, rescale_size = 401 and rescale_flank = 1. For the observed-over-expected maps, contact frequencies were normalized by dividing them by 'expected' values, which were calculated using the expected_cis function from cooltools (v0.5.4) with ignore_diags = 2. The plotpup module from coolpuppy was used to generate figures.

The average contact maps were generated from Micro-C contact maps with a resolution of 10 bp. Using lower-resolution Micro-C maps for plotting the average map leads to more blurred features and thicker diagonal (Extended Data Fig. 1c,d).

## TSS–TES contacts calculation

To calculate the observed number of TSS–TES contacts for each operon, we summed the ICE-balanced contacts between the 40 bp region (4 × 10 bins): one centred at the TSS and the other at the TES. To estimate the expected number of contacts for an operon, we selected 10 random genomic intervals of the same length as the operon. We then calculated the contact counts between the ends of these intervals using the same method and averaged the values to obtain the expected contact count.

As a control, we also calculated the number of contacts between more internal regions of the operon (intra-operon contacts). Specifically, we selected a region located 20% of the operon length downstream of the TSS and another region 20% of the operon length upstream of the TES (Fig. 2g).

The observed–expected frequencies of TSS–TES and intra-operon contacts are provided in Supplementary Table 3.

## Piles-up

The average ChIP–seq signal around features were plotted using the custom function plot_around_loop with some imports from the pybbi package (v0.4.0). The following length of flanking regions was used: 1,000 bp (Extended Data Fig. 1g) and 500 bp (all other piles-up).

We used the following ChIP–seq datasets: GSE157512 for H-NS, H-NS under Rif, StpA and RpoB; GSE182473 for TopoI; GSE182079 for GyrA and GyrB; E-MTAB-332 (Array Express) for Fis; GSE181767 for MukB, HupA and HupB; and GSE152880 for GapR.

ChIP–seq for TopoI was initially aligned to the W3110Mu assembly, so the liftOver chain file was generated using pyoverchain to transfer the signal to the NC_000913.3 assembly.

## Shuffle test

To assess whether a genomic feature (*A*) non-randomly colocalizes with or avoids another feature (*B*), a shuffle (permutation) test was conducted. For a summary of the procedure:

(1) The intersection between features *A* and *B* ($A \cap B$) was calculated using bedtools intersect with a fraction overlap threshold of 0.1 (option f = 0.1).

(2) To perform shuffling, bedtools shuffle was utilized, with the option chrom=True to maintain the chromosome structure while randomizing the locations of the features.

(3) For each shuffle, the intersection between the shuffled feature *A* and feature *B* ($A \cap B$) was calculated.

(4) The number of shuffle iterations was set to 1,000 ($N_s = 1,000$).

(5) The *P* value was computed as the minimum of the ratios $N_{A_s \cap B > A \cap B}/N_s$ and $N_{A_s \cap B < A \cap B}/N_s$, where $N_{A_s \cap B > A \cap B}/N_s$ represents the number of shuffles where the intersection between shuffled *A* and *B* is greater than the number of observed intersection $A \cap B$, and $N_{A_s \cap B < A \cap B}/N_s$ represents the number of shuffles where the intersection is smaller than the observed intersection.

To assess the colocalization of H-NS peaks with hairpins, we used H-NS peaks from the GSE157512 dataset. To assess the colocalization of HTGs with hairpins, we retrieved HTGs from the database HGT-DB[47] (https://usuaris.tinet.cat/debb/HGT/ecoli.d/HGTList.html).

## Motifs

The motif analysis was performed using HOMER (v4.11.1). The identified motifs were compared with the prodoric_2021.9.meme database using tomtom (v5.5.5) from the MEME suite.

## RNA sequencing

Total RNA was isolated from 0.5 ml cell culture using the Qiagen RNease Mini RNA Isolation Kit, following the manufacturer's protocol. rRNA was removed from 200 ng of the starting material using the NEBNext rRNA Depletion Kit (bacteria), and libraries were prepared using the NEBNext Ultra II Directional RNA-Seq library Preparation Kit. Libraries were sequenced on an Illumina Nextseq 2000 instrument in paired-end mode (2 × 61 bp) to a depth of approximately 20 million read pairs per sample. Data processing was conducted using the nf-core/rnaseq (v3.16.0) workflow (https://doi.org/10.5281/zenodo.1400710), part of the nf-core collection[69], utilizing reproducible software environments from Bioconda[70] and Biocontainers[71]. Differential gene expression analysis and data visualization were performed with the DESeq2 package[72].

## RT–qPCR

Cell cultures were mixed with ice-cold ethanol–phenol stop solution to inactivate cellular RNases, pelleted at 4 °C, washed with cold saline and stored at −80 °C. Total RNA was extracted using the hot phenol method[73], and RNA quality was assessed by agarose gel electrophoresis and the Qubit RNA IQ assay. RNA quantitation was performed with the Qubit RNA BR Assay Kit. RNA samples were treated with ezDNase (Invitrogen) to remove contaminating genomic DNA. Complementary DNA synthesis was performed using 0.25 µg of total RNA, MMLV reverse transcriptase (Evrogen) and specific oligonucleotide primers according to the manufacturer's instructions. RT–qPCR was performed using qPCRmix-HS SYBR (Evrogen) on a QuantStudio 5 Real-Time qPCR system (Applied Biosystems). *rpoC* (RNA polymerase β′-subunit) and *rrn* (pre-rRNA) served as housekeeping genes. All samples were normalized to the 3′ end of 16S rRNA, and fold changes in expression levels were calculated using the Pfaffl method[74].

## Red-C

Red-C was performed as described[38] with minor modifications relating to cell fixation and lysis conditions. Cell culture (5 ml) was transferred to 35 ml PBS-diluted formaldehyde (F8775, Sigma-Aldrich)

to obtain a final formaldehyde concentration of 3%. Fixation was allowed to proceed for 30 min at room temperature with rotation followed by quenching with 375 mM glycine. Cells were centrifuged for 10 min at 3,220g and 4 °C, washed with 10 ml cold PBS and centrifuged again.

Cells were resuspended in 500 µl TE (10 mM Tris pH 8.0 and 1 mM EDTA) supplemented with 1× protease inhibitors (Bimake) and 100 U SUPERase.In RNase inhibitor (Invitrogen) followed by adding lysozyme (L6876, Sigma) to 4 µg ml⁻¹. The mixture was incubated for 30 min at 37 °C with shaking at 1,000 rpm followed by adding 25 µl 10% SDS and incubating for 15 min at 37 °C with shaking at 1,000 rpm. The sample was supplemented with 500 µl TE and 87 µl 20% Triton X-100 followed by incubation for 15 min at 37 °C with shaking at 1,000 rpm. After adding 370 µl 4× NEB buffer 4, the sample was pelleted for 20 min at 16,100g and room temperature and resuspended in 500 µl 1× NEB buffer 4. DNA was digested with 20 µl NlaIII (10 U µl⁻¹, NEB) for 2 h at 37 °C with shaking at 1,200 rpm. Downstream steps of the Red-C procedure including RNA-to-DNA bridge ligation, reverse transcription, library construction and sequencing were done as previously described[38]. The raw Red-C reads were processed using the RedClib computational pipeline (https://github.com/agalitsyna/RedClib) as previously described[38].

## Estimation of transcription level from Red-C data

The Red-C data on RNA–DNA contacts obtained in this work were used to estimate the level of transcription in operons and other genomic features (CHINs, CHIDs and OPCIDs), as previously suggested[38]. As a measure of transcriptional activity of a feature, we calculated the number of contacts that RNA transcribed from this feature establishes near the feature (the distance between mapping coordinates of RNA and DNA reads was required to be no more than 10 kb). We reasoned that RNA captured near its parental locus may better convey the transcriptional activity of this locus than, for example, the total RNA output as determined by RNA sequencing. Of note, a good correlation has been shown between the level of a transcript in the RNA sequencing data and the number of contacts it showed in Red-C data[38]. The Red-C signal for operons, OPCIDs, CHINs and CHIDs is presented in Supplementary Tables 3–6, respectively. We also generated transcription activity profiles by calculating for each genomic bin how many contacts the RNA transcribed from this bin establishes near the bin (Extended Data Fig. 10).

## Hi-C

Hi-C was performed as previously described[75] with minor modifications. Cell fixation, lysis and restriction enzyme digestion were performed as described above for Red-C with the only difference that 10 µl HpaII (50 U µl⁻¹, NEB) was used instead of NlaIII and the digestion was carried out for 3 h instead of 2 h.

After HpaII digestion, the sample was pelleted for 10 min at 21,000g and 10 °C and resuspended in 250 µl 1× NEB buffer 2 followed by adding 42 µl fill-in solution composed of 15 µl 1 mM biotin–dCTP, 1.5 µl 10 mM dAGTTP (mix of dATP, dGTP and dTTP at 10 mM each) and 25.5 µl water. The mixture was supplemented with 8 µl Klenow (5 U µl⁻¹, NEB) and incubated for 2 h at 25 °C with shaking at 900 rpm. The sample was pelleted as above and resuspended in 300 µl 1× T4 DNA ligase buffer (Thermo Fisher) followed by adding 12.5 µl T4 DNA ligase (5 Weiss U µl⁻¹, Thermo Fisher). DNA proximity ligation was carried out overnight at 20 °C with shaking at 1,000 rpm. The sample was pelleted as above and resuspended in 375 µl proteinase K buffer composed of 4 µl 1 M Tris pH 8.0, 8 µl 5 M NaCl, 1.6 µl 0.5 M EDTA, 40 µl 10% SDS and 321 µl water followed by adding 25 µl proteinase K (20 mg ml⁻¹, Ambion) and incubating for 8 h at 65 °C. Downstream steps of the Hi-C procedure including DNA purification, sonication, biotin pulldown, library construction, sequencing and read processing were done as described above for Micro-C.

## In vivo crosslinking and mass spectrometry

In vivo crosslinking and mass spectrometry-assisted identification of covalent crosslinks formed by DSG were performed as previously described[62], with minimal modifications. In brief, E. coli MG1655 rpoC::His₁₀ cells were grown in 0.5× terrific broth (Difco) at 37 °C with shaking at 250 rpm to an OD₆₀₀ of approximately 0.5. The culture was then supplemented with a 300 mM stock solution of DSG (CF Plus) in DMSO, to a final concentration of 2 mM. After addition of DSG, the culture was incubated with shaking at 37 °C for 30 min, followed by quenching with 20 mM Tris-HCl pH 7.5. Cells were harvested by centrifugation and processed as previously described[62]. DSG-crosslinked RNA polymerase was purified by a combination of Ni-affinity and size-exclusion chromatography. Crosslinked peptides were identified from mass spectrometry raw data using pLink2 software[62]. For comparative analysis of crosslinking efficiency, data obtained from DSS-crosslinked samples prepared under identical conditions were analysed in parallel[62].

## H-NS cloning and purification

The open reading frame of the E. coli H-NS protein was cloned into the pTYB-1 vector (NEB) and transformed into Bl21 (DE3) cells. A single colony was used to inoculated 5 ml of LB medium supplemented with 100 mg ml⁻¹ carbenicillin, and the culture was incubated overnight at 37 °C. The following day, 0.5 ml LB medium of overnight culture was used to inoculate two 0.5 l LB cultures in 2-l flasks. Each flask was supplemented with 25 ml of 20× NPS solution (0.5 M (NH₄)₂SO₄, 1 M KH₂PO₄, 1 M Na₂HPO₄; final pH approximately 6.75), 10 ml of 50× 5052 autoinduction solution (25% glycerol, 2.5% glucose and 10% lactose) and 0.5 ml of 1 M MgSO₄. Cultures were grown overnight at 30 °C in the presence of 100 µg ml⁻¹ carbenicillin following the autoinduction protocol[76]. Cells were collected by centrifugation at 5,000g and the pellet was resuspended in 75 ml of chitin–intein column (CIC) buffer (20 mM Tris-HCl pH 8.0, 500 mM NaCl and 1 mM EDTA) containing two protease inhibitor cocktail tablets (Roche). Lysozyme was added to a final concentration of 0.5 mg ml⁻¹ and the suspension was incubated at 37 °C for 30 min with gentle stirring. Sodium deoxycholate was then added to 0.05% and stirring continued for another 15 min. The suspension was sonicated on ice, and the lysate was clarified by centrifugation at 20,000g for 30 min and 4 °C. The cleared lysate was applied by gravity flow to a chitin–agarose column (approximately 5 ml dry bed, NEB) pre-equilibrated in CIC buffer at room temperature. After washing with 60 ml CIC, the column was incubated with 15 ml of CIC buffer supplemented with 50 mM dithiothreitol to induce on-column intein-mediated cleavage. The column was sealed and left overnight at room temperature. The following day, the eluted protein was collected with 20 ml CIC buffer. The eluate was concentrated using an Amicon centrifugal filter unit (15 ml, 3 kDa MWCO) at 5,000g and 4 °C to a final concentration of approximately 0.7–1 ml. The protein was diluted 1:1 with glycerol and stored at −80 °C.

## Transmission electron microscopy

The control DNA template (plasmid without CHIN) was pVS10 (ref. 77) containing the E. coli genes rpoA, rpoB and rpoC. To generate a CHIN-containing plasmid, the rpo gene region of pVS10 was replaced via Gibson Assembly (NEB) with a genomic fragment encompassing yqeG, yqeH, yqeL, yqeJ, yqeK, ygeG, ygeH and ygeL (E. coli coordinates 2,985,045–2,996,445; Extended Data Fig. 5e,f). Plasmids were purified using the Qiagen Midiprep kit. To relax supercoiling, plasmids were nicked before imaging. For the control plasmid, nickase Nb.BbvCI (NEB) was used; for the CHIN-containing plasmid, Nt.BspQI (NEB) was used. Nicked DNA was purified by phenol–chloroform extraction and ethanol precipitation (no carrier).

H-NS–DNA complexes were formed following a modified version of a previously described protocol[78]. In brief, 100 ng of plasmid DNA

was incubated with 420 ng of purified H-NS in 10 µl of binding buffer (40 mM HEPES pH 8.0, 10 mM MgCl$_2$ and 60 mM KCl) for 30 min at 30 °C. The reaction mixture was then diluted 20-fold with a deposition buffer to a final concentration of 5 mM HEPES pH 8.0, 5.5 mM MgCl$_2$ and 3 mM KCl and applied to glow-discharged carbon-coated copper transmission electron microscopy grid (PELCO, 01844-F, Ted Pella Inc.). In some experiments, the dilution step was omitted to increase the density of visualized molecules (Extended Data Fig. 5h,i). Plasmid without H-NS was prepared and deposited under identical conditions. Grids were negative stained with 1% (w/v) uranyl acetate for 5 min, air dried and imaged using a FEI Talos 120 C transmission electron microscope equipped with a Gatan OneView 4k × 4k camera.

## PLA

Construction of the dCas9 expression plasmid was based on the pdCas9-bacteria plasmid[79], modified by Gibson Assembly to append a carboxy-terminal 6×histidine (6×His) tag. Dual small guide RNA (sgRNA) expression cassettes were constructed by replacing the single sgRNA module of the pgRNA-bacteria plasmid[79] with synthetic DNA (Twist Bioscience) encoding extra-long sgRNA arrays (ELSA) as previously described[80].

Each dual cassette included the following elements: SHP050 promoter – sequence-specific gRNA1 spacer – ID46 Handle – ECK120010868 terminator – SHP038 promoter – sequence-specific gRNA2 spacer – ID26 Handle – ECK120017009 terminator. Three dual cassettes were assembled to target combinations A–C, B–C or D–C (Extended Data Fig. 5b). In all constructs, gRNA2 targeted position C, whereas gRNA1 targeted positions A, B or D. Plasmids encoding dCas9-6×His and the dual ELSA arrays were sequentially transformed into E. coli strain MG1655 to generate the strains used in PLA experiments. Full sequences of all cassettes are listed in Supplementary Table 9.

PLA was performed as previously described[81–83]. Strains carrying both plasmids were grown in LB medium at 37 °C to an OD$_{600}$ of approximately 0.3, then induced with 0.1 µg ml$^{-1}$ anhydrotetracycline (aTc) to express dCas9-6×His. After 30 min of incubation, cells were harvested by centrifugation and washed three times with PBS. Pellets were resuspended in PBS and immobilized on cleaned poly-L-lysine-coated coverslips for 30 min. Cells were fixed in 4% paraformaldehyde in PBS for 20 min at room temperature. After fixation, cells were lysed with a buffer consisting of 25 mM Tris pH 8.0, 10 mM EDTA, 0.5% Triton X-100 and 0.1 mg ml$^{-1}$ lysozyme, and incubated for 30 min with gentle agitation. Samples were washed once with PBS and then subjected to the Duolink In Situ PLA protocol (Millipore Sigma), using mouse anti-6×His antibodies (66005-1-Ig, Proteintech) and rabbit anti-6×His antibodies (ab9108, Abcam), each at 1:300 dilution. Duolink PLA probes and detection reagents (Millipore Sigma) were used per the manufacturer's instructions.

Fluorescence imaging was performed on a Zeiss LSM700 confocal microscope. The number of PLA foci per cell was quantified in an unbiased manner using ImageJ (Extended Data Fig. 5d).

To confirm DNA colocalization of PLA foci, coverslips prepared for confocal imaging were repurposed for super-resolution imaging on a custom-built optical system based on an ASI-RAMM inverted microscope, equipped for multilaser excitation[81]. The lasers were adjusted to a highly inclined and laminated optical sheet mode. Each colour was filtered using single-bandpass filters (FF01-676/37 and FF01-607/36 for AF647, Semrock), and filters were switched using a motorized filter wheel (FW-1000, ASI) for sequential illumination. Photons were collected with a Teledyne Photometrics Prime 95B sCMOS camera at 33 Hz (30 ms per frame) for a minimum of 1,000 frames. Images were rendered from localization coordinates on a 10-nm pixel canvas and displayed after Gaussian blurring (σ = 10 nm; Extended Data Fig. 5c).

## Data reproducibility

All experiments were performed with a minimum of two biological replicates, except for the Micro-C experiments without added DNA ligase and those with varying cultivation or fixation temperatures, which were performed as single repeats. In addition, the Micro-C experiments with the antibiotics ciprofloxacin, novobiocin and netropsin were performed at two concentrations, with each concentration run as a single repeat.

The Micro-C contact maps showed strong concordance between replicates (Extended Data Fig. 7c,d), as indicated by a stratum-adjusted correlation coefficient of more than 0.98 (Extended Data Fig. 7b). Consequently, all analyses were performed using the combined data from the replicates.

## Reporting summary

Further information on research design is available in the Nature Portfolio Reporting Summary linked to this article.

## Data availability

Micro-C and Hi-C fastq raw reads and cool contact matrices; Red-C fastq raw reads and tsv files with contacts; and RNA sequencing raw reads are available under the Gene Expression Omnibus accession code GSE272161. Source data are provided with this paper.

## Code availability

The code used in analysis has been deposited in GitHub (https://github.com/irzhegalova/ecoli_microc).

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

**Acknowledgements** We thank the Center for Precision Genome Editing and Genetic Technologies for Biomedicine, IGB for their support; members of the E.N. laboratory, G. Gurman and V. Svetlov for conducting the in vivo crosslinking experiments and data analysis; the NYU Langone Microscopy Laboratory for access to the transmission electron microscope; and F.-X. (A.) Liang, J. Sall and A. Pisarev for their technical assistance. This work was supported by the Russian Science Foundation (grant 21-64-00001 to S.V.R.), the Blavatnik Family Foundation (to E.N.) and the Howard Hughes Medical Institute (to E.N.).

**Author contributions** A.A.G. and E.N. conceived the study. A.A.G. performed the Micro-C, Hi-C and Red-C experiments. S.P. constructed the knockout *E. coli* strains and performed RT–qPCR. Y.S. performed RNA sequencing. I.S. and I.Z. analysed the data under the supervision of A.A.G. and S.V.R. A.A.G. and G.E. performed manual annotation of CHINs, CHIDs and OPCIDs. V.E. prepared the substrates for transmission electron microscopy. A.R. and A.B. prepared the strains for PLA. S.L. optimized sample and slide preparation and imaging using confocal and HILO microscopy under the supervision of E.R. A.A.G. and I.Z. prepared the figures. S.V.R. and E.N. supervised the project. S.V.R. and E.N. wrote the manuscript with input from all the authors.

**Competing interests** The authors declare no competing interests.

**Additional information**
**Correspondence and requests for materials** should be addressed to Sergey V. Razin or Evgeny Nudler.

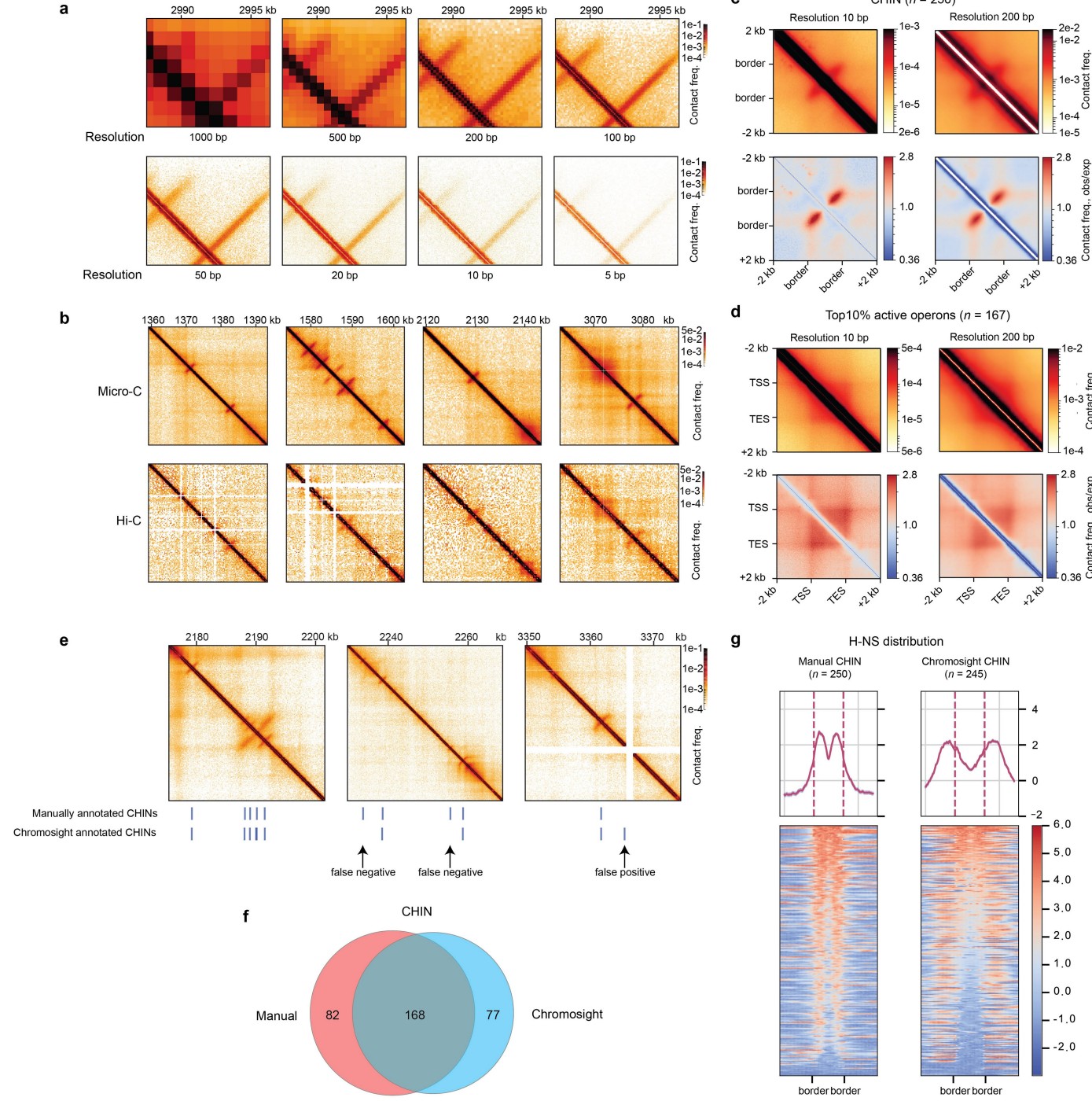

**Extended Data Fig. 1 | Comparison of Micro-C data at different resolutions and detection of CHINs in Micro-C and Hi-C data. a**, Micro-C contact map of a region harboring two CHINs displayed at different map resolution. **b**, Examples of CHINs seen in Hi-C contact maps obtained in this study, alongside corresponding Micro-C contact maps of the same regions. All maps are provided at a resolution of 200 bp. **c,d**, Average contact map for CHINs (**c**) and the top10% of active operons (**d**) generated from Micro-C contact maps with resolution of 10 and 200 bp. **e**, Representative examples of Micro-C contact maps showing the positions (centers) of CHINs, annotated either manually or using the Chromosight program[67]. **f**, Intersection of CHINs mapped manually and those identified by the Chromosight program. **g**, Distribution of H-NS around CHINs, as mapped manually and by the Chromosight program.

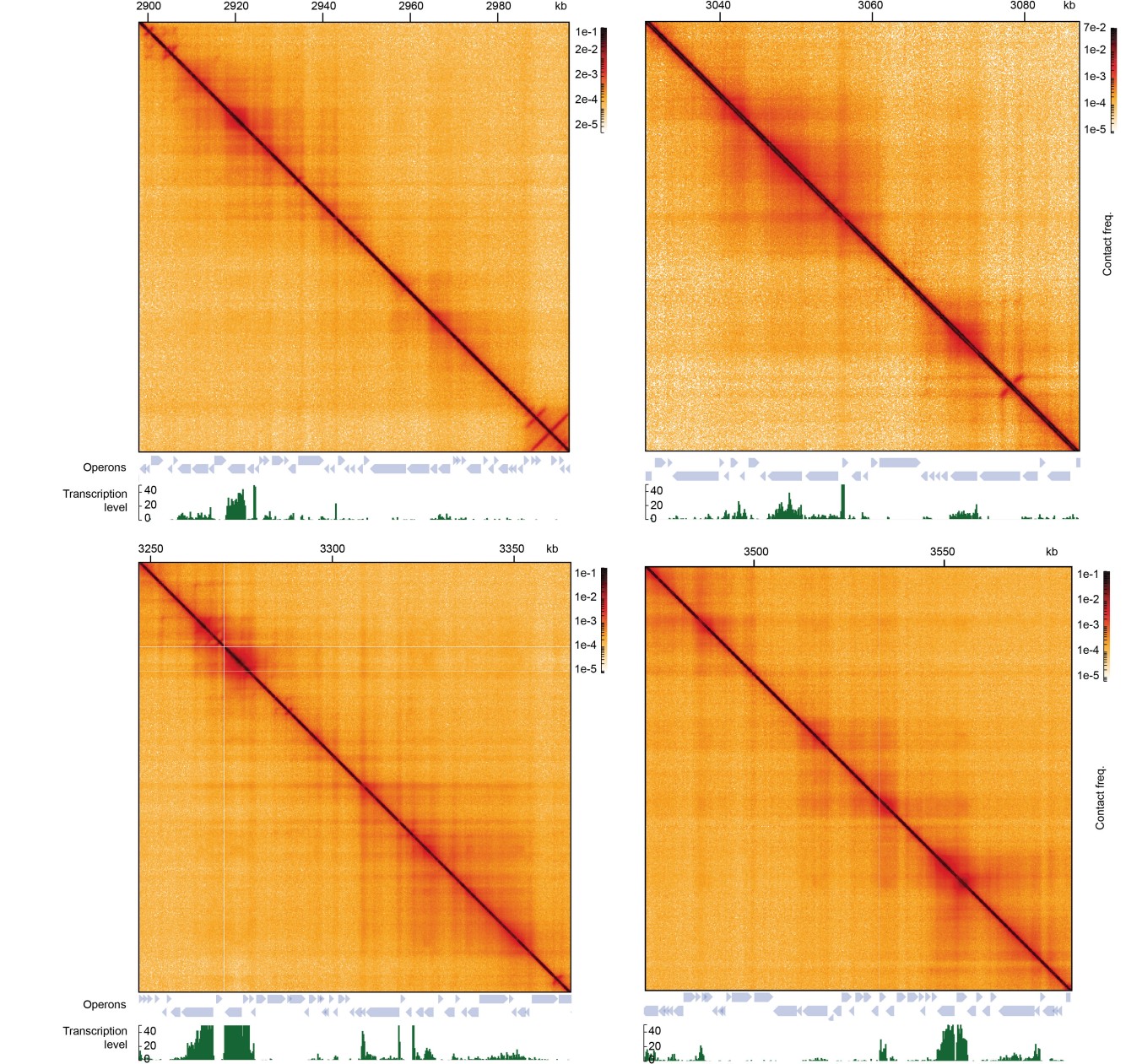

**Extended Data Fig. 2 | Plaid patterns observed in contact maps.** Examples of Micro-C contact maps for wild-type *E. coli* cells in normal growth conditions are shown.

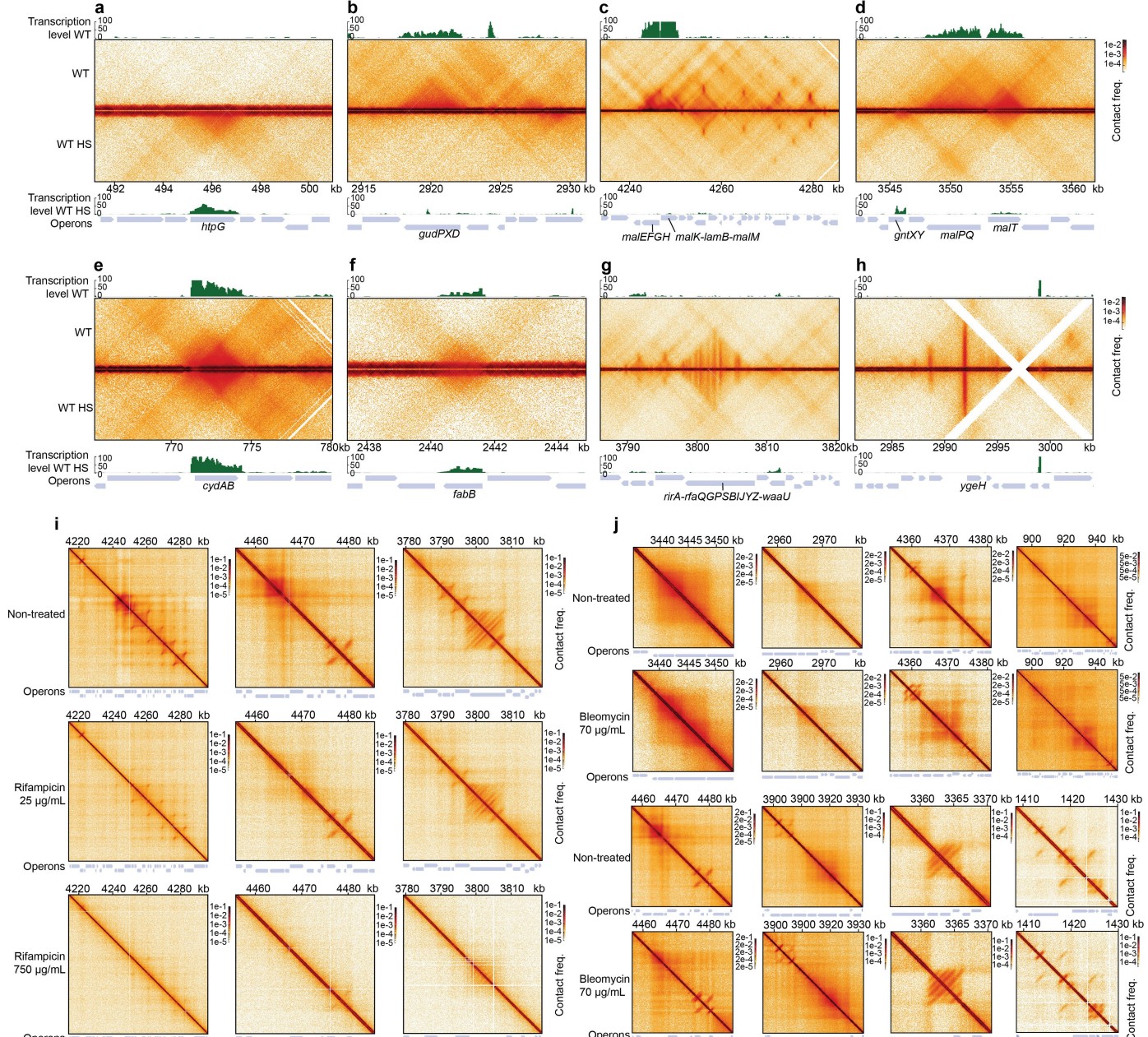

**Extended Data Fig. 3 | Effect of heat shock, rifampicin, and bleomycin treatment on OPCIDs, CHINs, and CHIDs. a**, Example of OPCID appearance under HS. **b,c**, Examples of OPCID disappearance under HS. **d**, A region harboring three OPCIDs, left (at *gntXY*) appearing under HS, middle (at *malPQ*) disappearing under HS, and right (at *malT*) weakening under HS conditions.

**e,f**, Examples of OPCIDs that are invariant between normal and HS conditions. **g,h**, Preservation of CHIN structures under HS conditions. **i**, Examples of Micro-C contact maps for non-treated and rifampicin-treated *E. coli* cells. **j**, Examples of Micro-C contact maps for non-treated and bleomycin-treated *E. coli* cells.

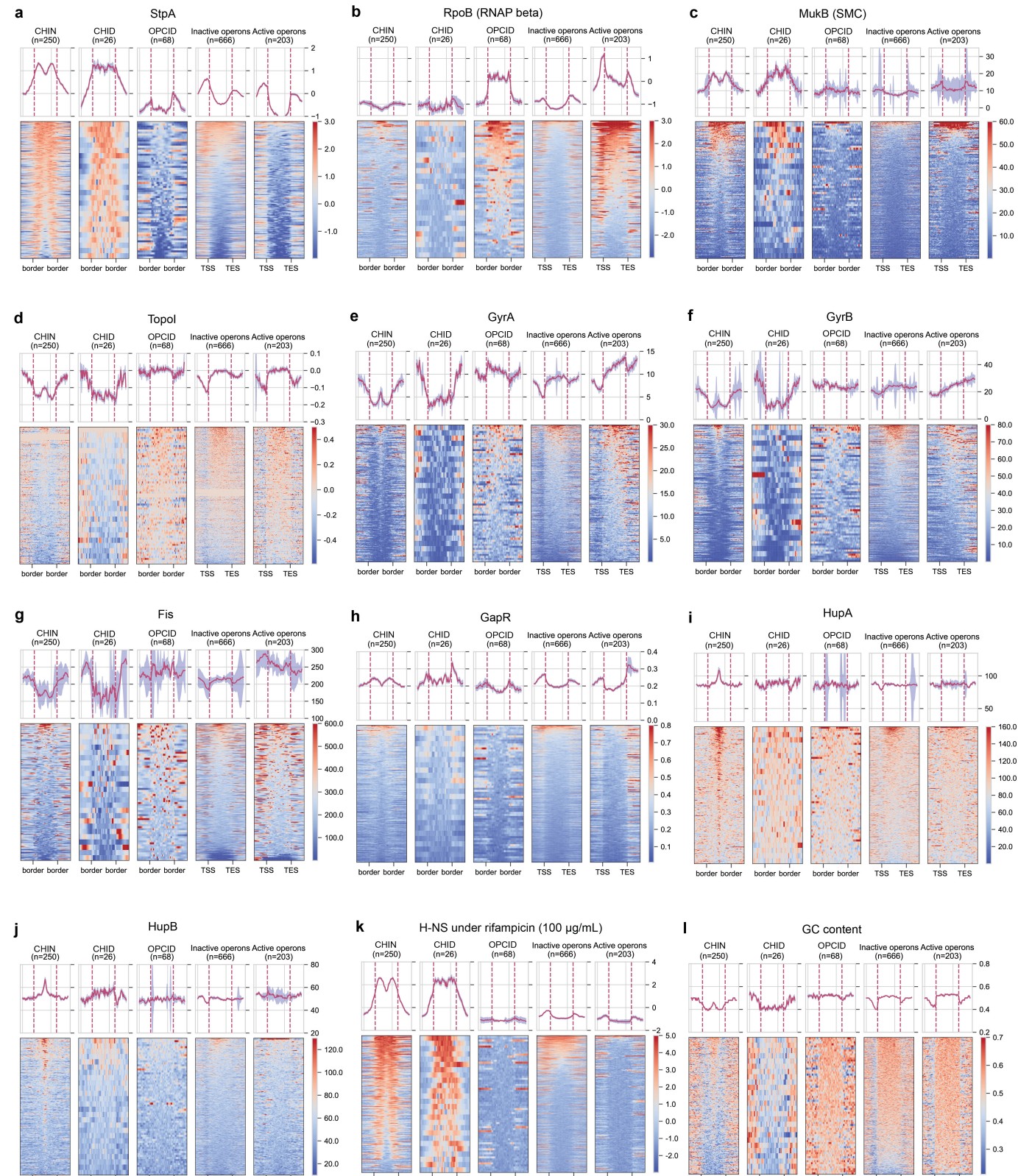

**Extended Data Fig. 4 | Distribution of proteins (a-k) and GC content (l) around different genomic features in wild-type *E. coli* cells.** All designations are as in Fig. 2b.

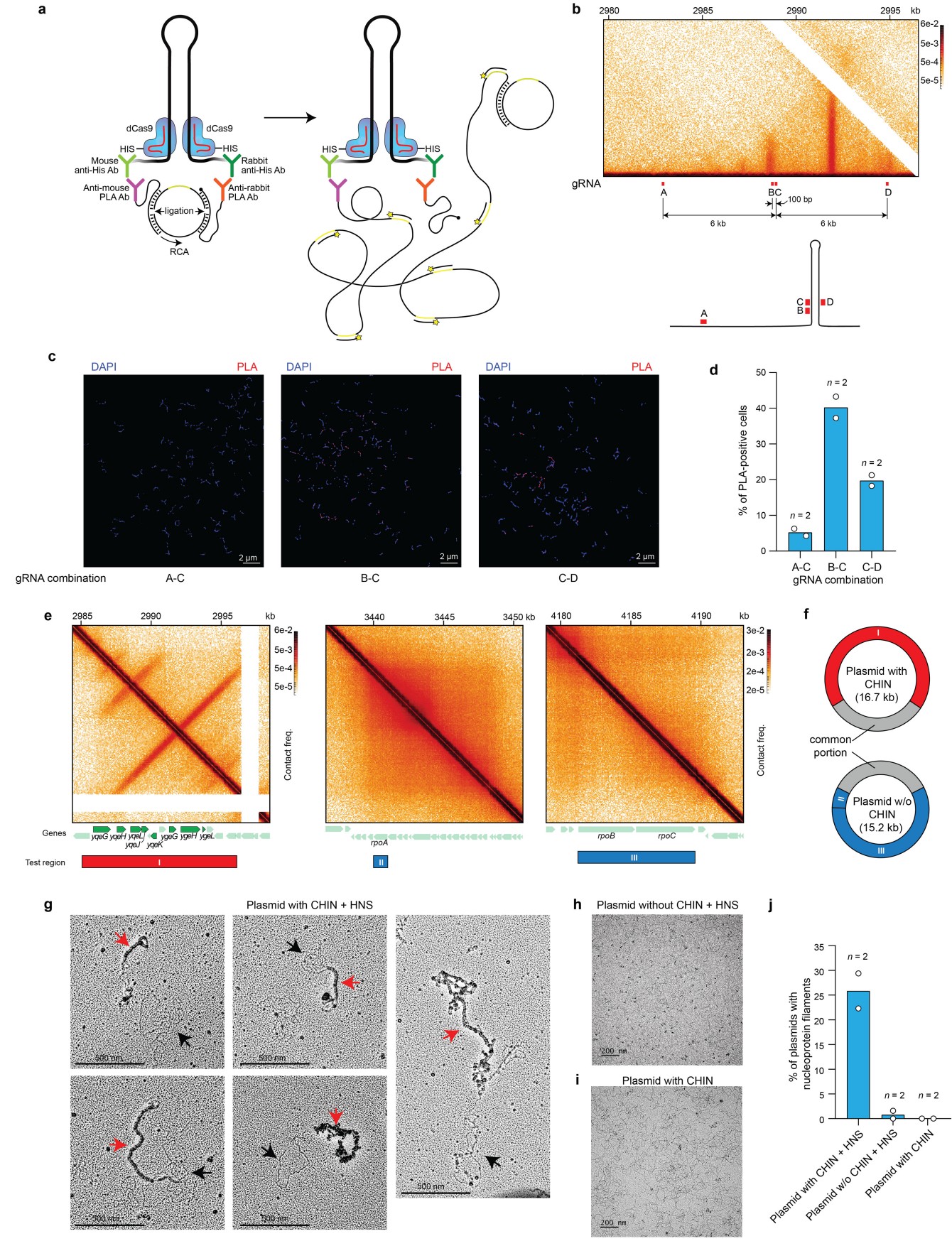

**Extended Data Fig. 5** | See next page for caption.

**Extended Data Fig. 5 | CHIN validation in vivo and in vitro. a-d**, Detection of CHIN assembly in vivo using proximity ligation assay (PLA). **a**, Schematic of the PLA approach. Two His-tagged dCas9 molecules are targeted to the CHIN bases via specific guide RNAs. Mouse and rabbit anti-His primary antibodies are added, followed by oligonucleotide-conjugated anti-mouse and anti-rabbit secondary antibodies. Each His-tag is recognized by only one antibody at a time, so the two dCas9 molecules are bound by either homotypic (mouse-mouse or rabbit-rabbit) or heterotypic (mouse-rabbit or rabbit-mouse) antibody pairs. If the dCas9 molecules are brought in close spatial proximity (<40 nm), the oligonucleotides on the heterotypic antibody pair are annealed to connector oligonucleotides, enabling their ligation into a circular DNA template. This DNA circle is then amplified by rolling circle amplification (RCA), generating a micrometer-sized DNA bundle with multiple binding sites for a fluorophore-labeled oligonucleotide, which is subsequently visualized by HILO microscopy. **b**, Micro-C contact map of the studied *E. coli* genomic region harboring a strong CHIN, with positions of the guide RNAs indicated. The inferred CHIN configuration and guide RNA locations are shown below. **c**, Representative PLA images of *E. coli* cells expressing His-tagged dCas9 and the indicated guide RNA pairs. **d**, Quantification of cells showing PLA foci under indicated conditions. Data are presented as mean values from two biological replicates, with at least 200 cells analysed per condition in each replicate. **e-j**, Transmission electron microscopy (TEM) analysis of H-NS complexes with plasmids containing *E. coli* genomic regions either prone or not prone to CHIN formation. **e**, Micro-C contact maps of the *E. coli* genomic regions inserted into plasmids. **f**, Schematic of the plasmids. The CHIN-containing plasmid harbors region I, which shows a strong CHIN signal in the Micro-C map (same region as analysed in **a-d**). The control plasmid lacks CHINs and contains regions II and III, which do not exhibit CHIN signatures. **g**, Representative TEM images of the CHIN-containing plasmid incubated with H-NS, showing nucleoprotein filaments (red arrows). Plasmid DNA is marked by black arrows. **h**, TEM image of the control plasmid (no CHINs) after incubation with H-NS. No nucleoprotein filaments are detected. **i**, TEM image of the CHIN-containing plasmid in the absence of H-NS. No filaments are observed. **j**, Quantification of plasmids displaying nucleoprotein filaments under the indicated conditions. Data are presented as mean values from two independent experiments, with at least 100 plasmids analysed per condition in each experiment.

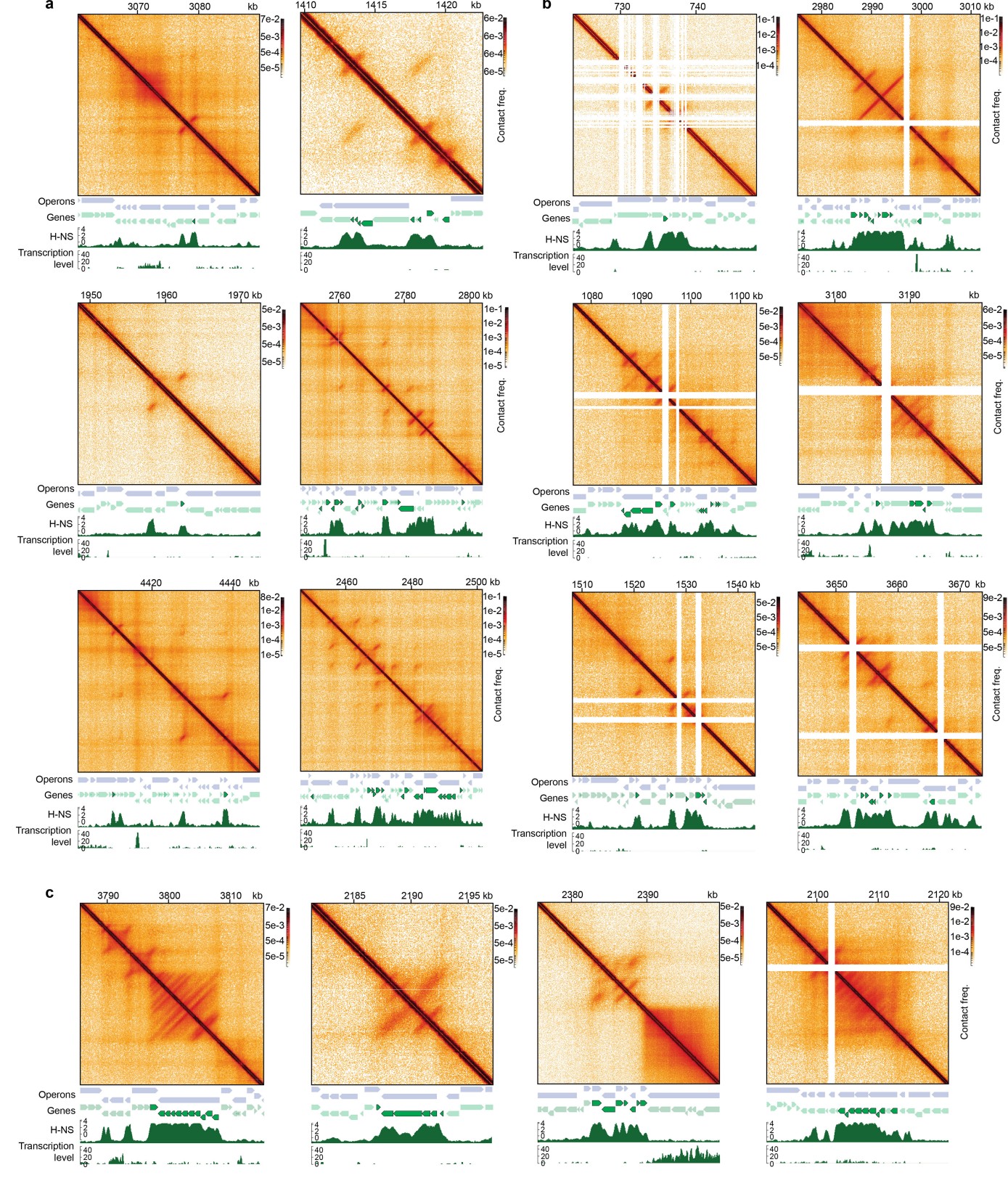

**Extended Data Fig. 6 | Colocalization of CHINs and CHIDs with HTGs.**
**a**, Examples of colocalization of HTGs (dark green filled arrows on the gene tracks) with individual CHINs and CHIN networks. **b**, CHINs and CHIDs formed on HTGs near repetitive elements. **c**, Colocalization of CHIDs with HTG clusters. Shown are Micro-C contact maps for wild-type *E. coli* cells in normal growth conditions.

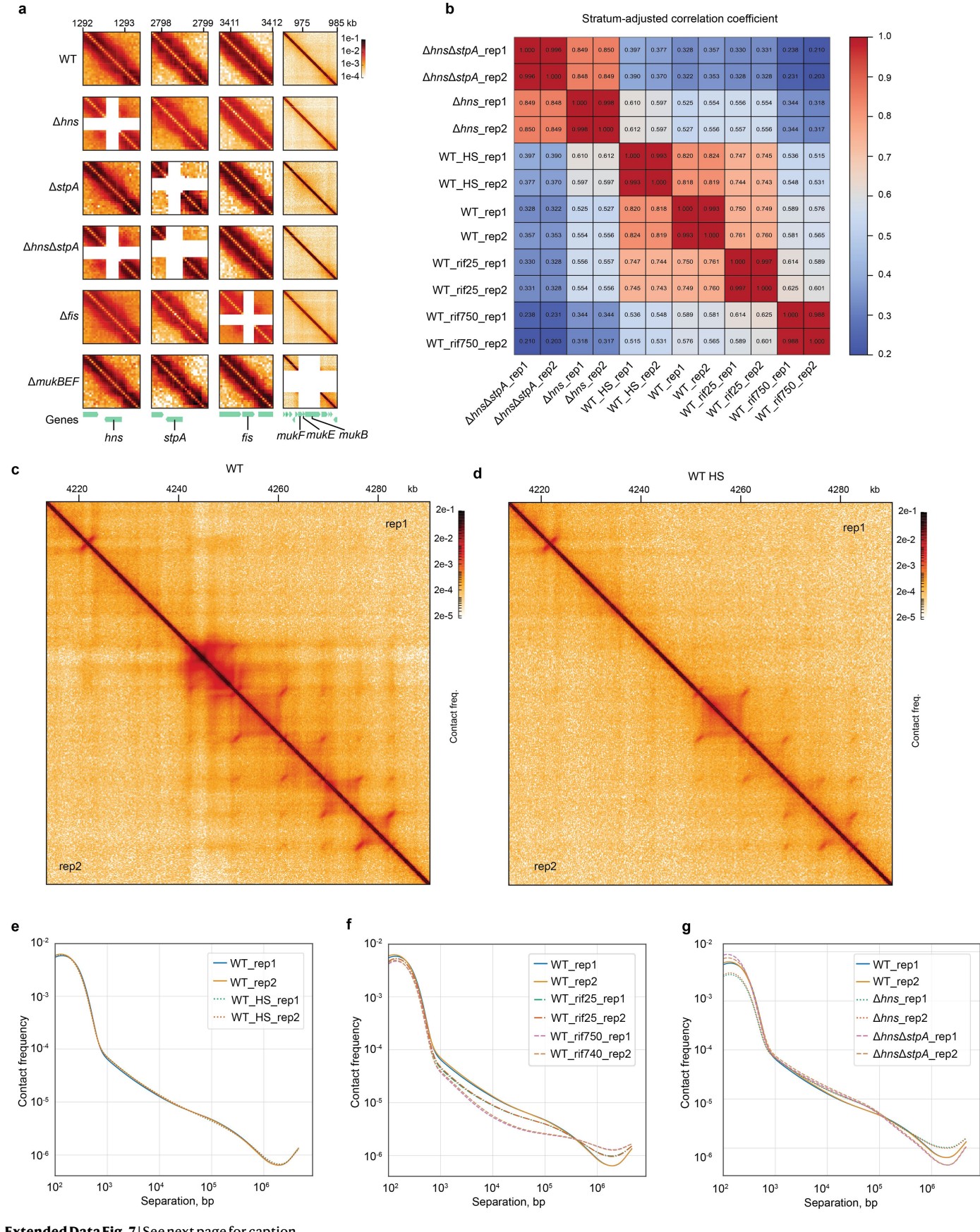

**Extended Data Fig. 7** | See next page for caption.

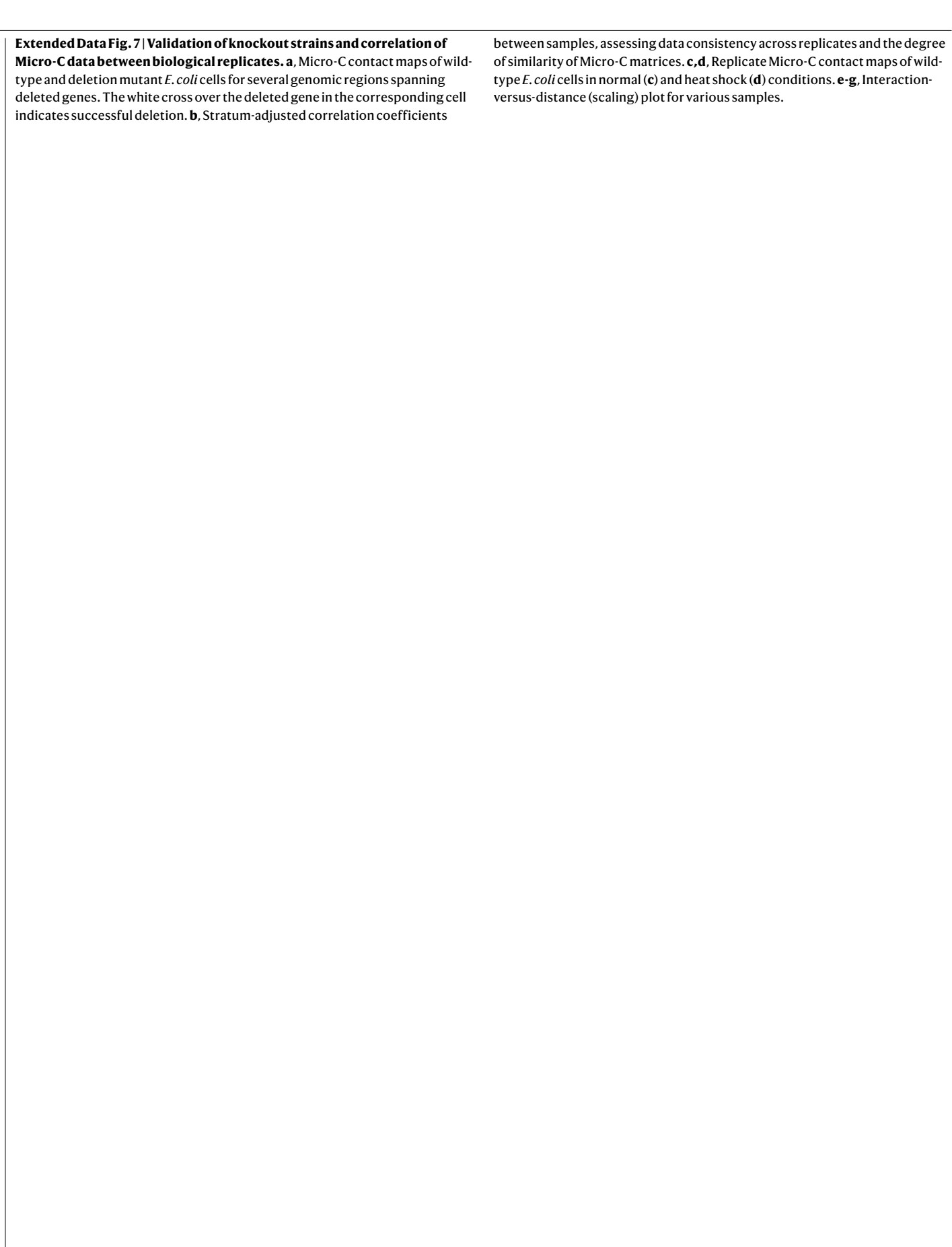

**Extended Data Fig. 7 | Validation of knockout strains and correlation of Micro-C data between biological replicates. a**, Micro-C contact maps of wild-type and deletion mutant *E. coli* cells for several genomic regions spanning deleted genes. The white cross over the deleted gene in the corresponding cell indicates successful deletion. **b**, Stratum-adjusted correlation coefficients between samples, assessing data consistency across replicates and the degree of similarity of Micro-C matrices. **c,d**, Replicate Micro-C contact maps of wild-type *E. coli* cells in normal (**c**) and heat shock (**d**) conditions. **e-g**, Interaction-versus-distance (scaling) plot for various samples.

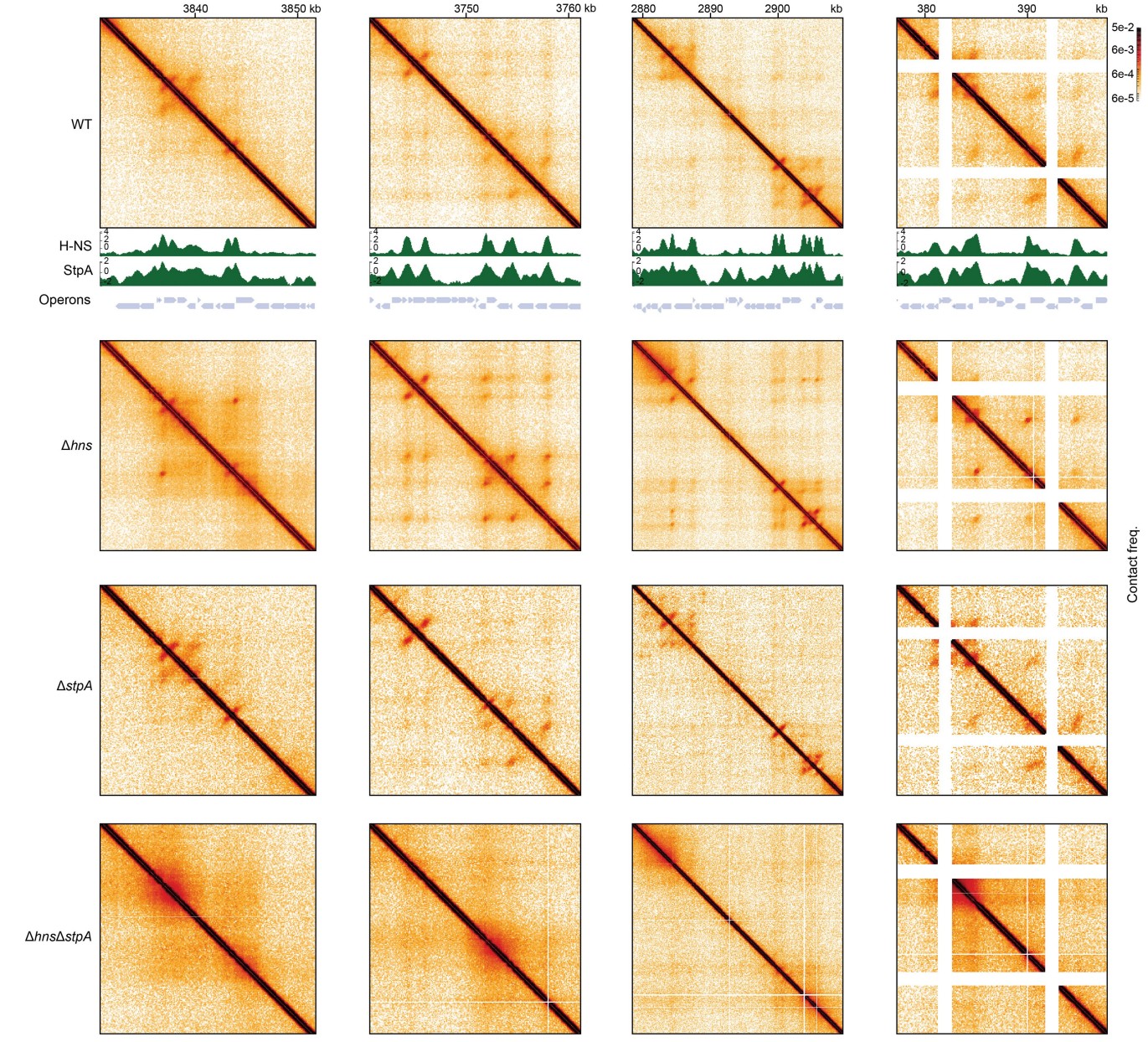

**Extended Data Fig. 8 | Intensification of CHIN spatial contacts in Δ*hns* cells.** Examples of Micro-C contact maps for wild-type, Δ*hns*, Δ*stpA*, and Δ*hns*Δ*stpA* *E. coli* cells are presented. Below the wild-type contact maps, the distribution profiles for H-NS and StpA in wild-type *E. coli* are provided alongside the operon track.

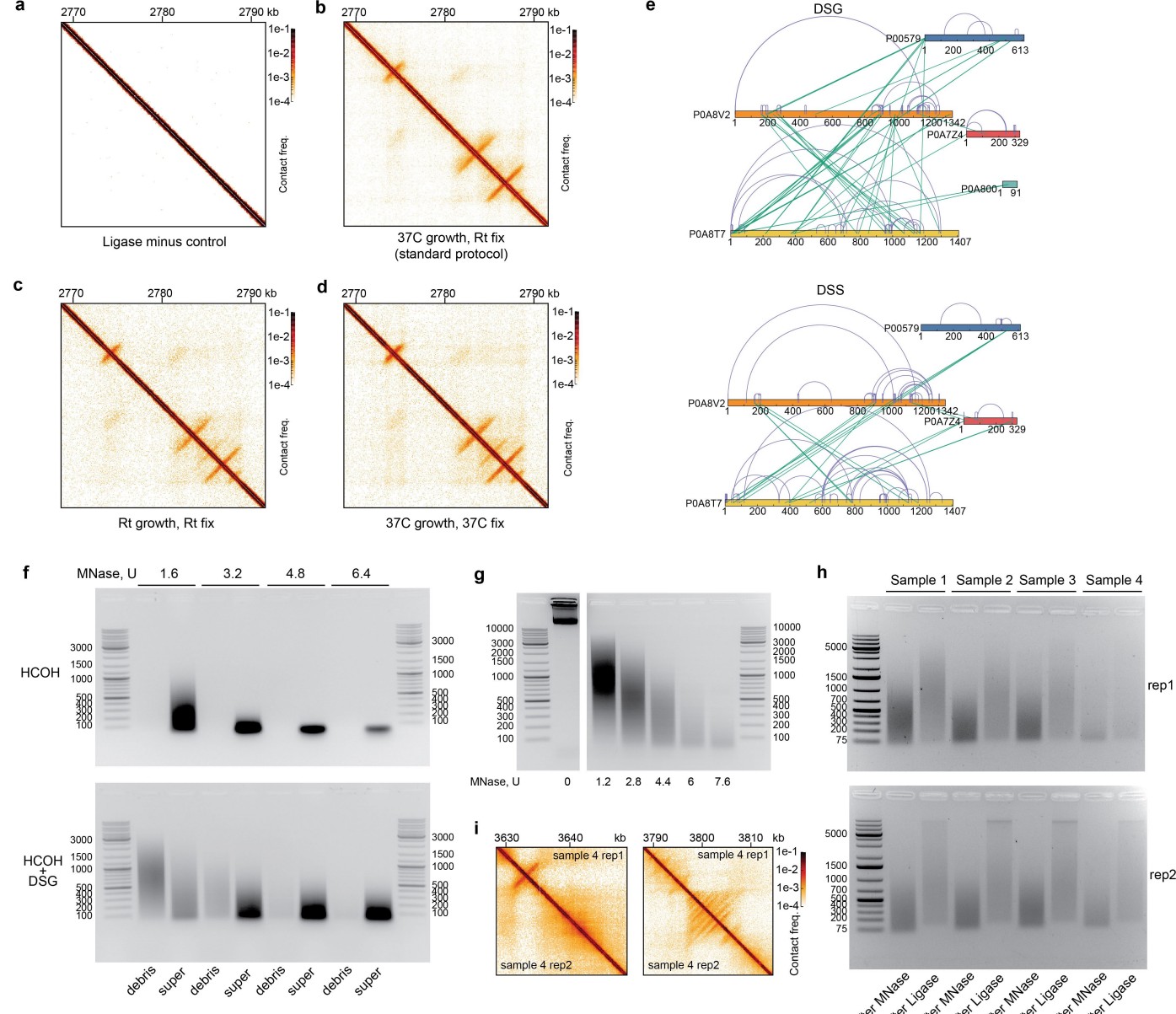

**Extended Data Fig. 9 | Micro-C control experiments and optimization of MNase digestion. a**, Micro-C contact map for wild-type *E. coli* cells processed according to the standard Micro-C protocol, but without the addition of DNA ligase. **b-d**, Comparison of Micro-C contact maps for wild-type *E. coli* cells grown at 37 °C and fixed with formaldehyde at room temperature, as per the standard Micro-C protocol (**b**), those grown and fixed at room temperature (**c**), and those grown and fixed at 37 °C (**d**). **e**, In vivo protein-protein crosslinks induced by treatment of *E. coli* cells with DSG and DSS, assessed by mass spectrometry. Protein names are according to UniProt. **f**, Distribution of DNA between soluble (super) and insoluble (debris) fractions following treatment with increasing amounts of MNase of wild-type *E. coli* cells fixed with formaldehyde alone (upper gel) or with formaldehyde and DSG (lower gel). **g**, Gel shows a typical MNase titration experiment. An amount of 5 U was selected as optimal, producing the desired fragment sizes range. **h**, Gels show four different samples tested after MNase digestion and ligation for two biological replicates. Sample 4 (non-relevant to the present study) indicates over-digestion, leading to excessive fragmentation, in replicate 1. For uncropped gels, see Supplementary Fig. 1. **i**, Micro-C contact maps for replicate 1 and 2 of sample 4, demonstrating similarity between contact maps (stratum-adjusted correlation coefficient 0.978).

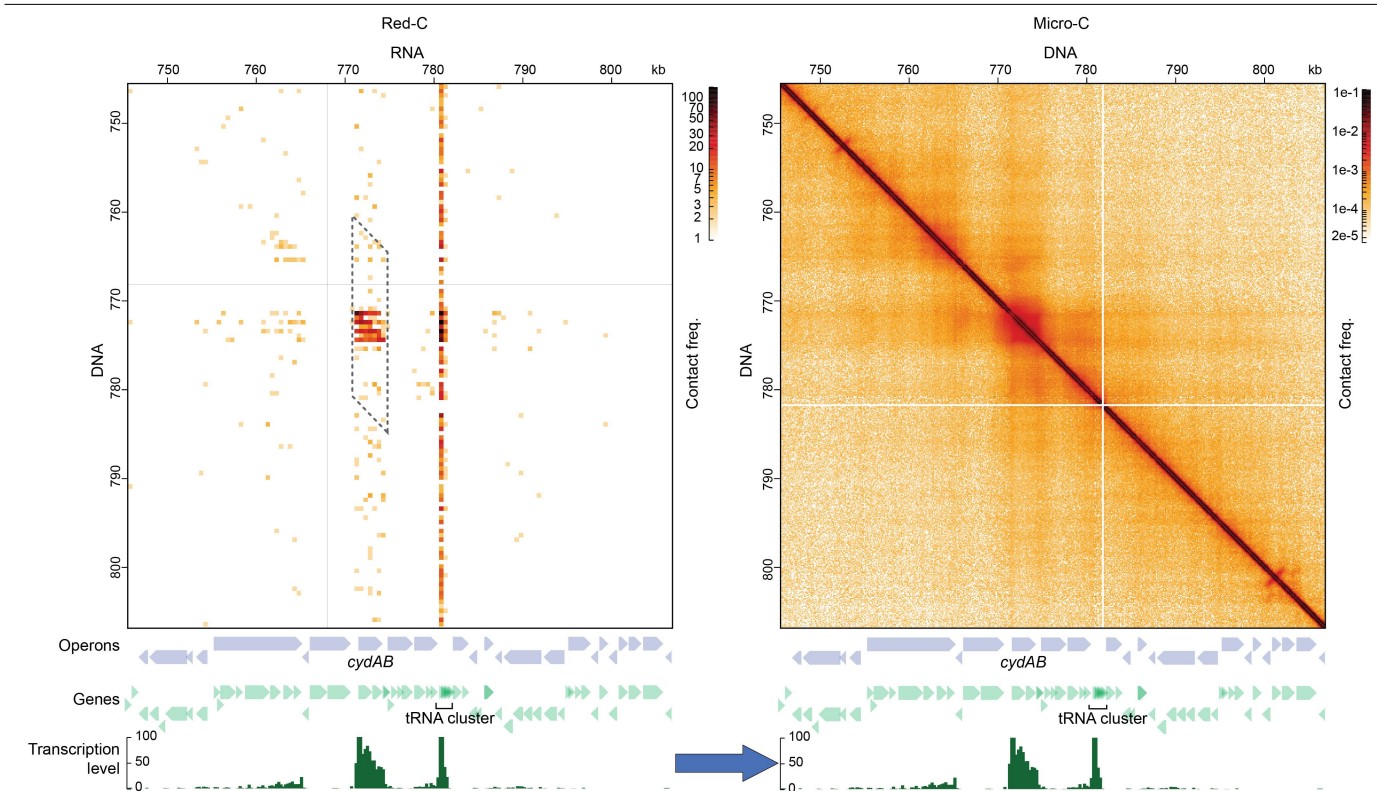

**Extended Data Fig. 10 | Inferring transcription level from Red-C data.** RNA-DNA (Red-C) and DNA-DNA (Micro-C) contact maps for the same genomic region are shown, along with the operon track, gene track, and transcription activity profile. The transcription activity is determined by summarizing, for each DNA bin, the number of contacts that RNA produced from this bin makes within ±10 kb from the bin. For example, the transcription activity of the *cydAB* operon is visualized by the signal in the dotted area of the RNA-DNA contact map. The red vertical line in the RNA-DNA contact map marks tRNA contacts.

# Reporting Summary

## Statistics

For all statistical analyses, confirm that the following items are present in the figure legend, table legend, main text, or Methods section.

| n/a | Confirmed | |
|---|---|---|
| ☐ | ☒ | The exact sample size (*n*) for each experimental group/condition, given as a discrete number and unit of measurement |
| ☐ | ☒ | A statement on whether measurements were taken from distinct samples or whether the same sample was measured repeatedly |
| ☐ | ☒ | The statistical test(s) used AND whether they are one- or two-sided<br>*Only common tests should be described solely by name; describe more complex techniques in the Methods section.* |
| ☐ | ☒ | A description of all covariates tested |
| ☐ | ☒ | A description of any assumptions or corrections, such as tests of normality and adjustment for multiple comparisons |
| ☐ | ☒ | A full description of the statistical parameters including central tendency (e.g. means) or other basic estimates (e.g. regression coefficient) AND variation (e.g. standard deviation) or associated estimates of uncertainty (e.g. confidence intervals) |
| ☐ | ☒ | For null hypothesis testing, the test statistic (e.g. $F$, $t$, $r$) with confidence intervals, effect sizes, degrees of freedom and $P$ value noted<br>*Give P values as exact values whenever suitable.* |
| ☒ | ☐ | For Bayesian analysis, information on the choice of priors and Markov chain Monte Carlo settings |
| ☒ | ☐ | For hierarchical and complex designs, identification of the appropriate level for tests and full reporting of outcomes |
| ☐ | ☒ | Estimates of effect sizes (e.g. Cohen's *d*, Pearson's *r*), indicating how they were calculated |

*Our web collection on statistics for biologists contains articles on many of the points above.*

## Software and code

Policy information about availability of computer code

| | |
|---|---|
| Data collection | No custom software was used. |
| Data analysis | Micro-C and Hi-C reads were processed using distiller-nf version 0.3.4. Mapping was performed using BWA. The sample correlation was computed using HiCRep. Scalings were calculated using cooltools v. 0.7.0. Average contact maps were plotted using coolpuppy v1.1.0 and cooltools v0.5.4. Piles-ups were generated using the custom function plot_around_loop with some imports from pybbi package v0.4.0. The motif analysis was performed using HOMER v4.11.1 and tomtom version 5.5.5 from the MEME suite. RNA-seq data processing was conducted using the nf-core/rnaseq v3.16.0 workflow. Differential gene expression analysis was performed with the DESeq2 package. Red-C sequencing reads were processed using the RedClib computational pipeline (https://github.com/agalitsyna/RedClib). |

For manuscripts utilizing custom algorithms or software that are central to the research but not yet described in published literature, software must be made available to editors and reviewers. We strongly encourage code deposition in a community repository (e.g. GitHub). See the Nature Portfolio guidelines for submitting code & software for further information.

## Data

Policy information about availability of data

All manuscripts must include a data availability statement. This statement should provide the following information, where applicable:

- Accession codes, unique identifiers, or web links for publicly available datasets
- A description of any restrictions on data availability
- For clinical datasets or third party data, please ensure that the statement adheres to our policy

Micro-C and Hi-C fastq raw reads and cool contact matrices are available under GEO accession: GSE272161.
Red-C fastq raw reads and tsv files with contacts are available under GEO accession: GSE272161.
RNA-seq raw reads are available under GEO accession: GSE272161.

## Research involving human participants, their data, or biological material

Policy information about studies with human participants or human data. See also policy information about sex, gender (identity/presentation), and sexual orientation and race, ethnicity and racism.

| | |
|---|---|
| Reporting on sex and gender | Not applicable to this study. |
| Reporting on race, ethnicity, or other socially relevant groupings | Not applicable to this study. |
| Population characteristics | Not applicable to this study. |
| Recruitment | Not applicable to this study. |
| Ethics oversight | Not applicable to this study. |

Note that full information on the approval of the study protocol must also be provided in the manuscript.

# Field-specific reporting

Please select the one below that is the best fit for your research. If you are not sure, read the appropriate sections before making your selection.

☒ Life sciences          ☐ Behavioural & social sciences          ☐ Ecological, evolutionary & environmental sciences

For a reference copy of the document with all sections, see nature.com/documents/nr-reporting-summary-flat.pdf

# Life sciences study design

All studies must disclose on these points even when the disclosure is negative.

| | |
|---|---|
| Sample size | Sample sizes were based on standard practice in similar studies and were deemed sufficient to ensure reproducibility and detect consistent effects across independent replicates. |
| Data exclusions | No data was excluded. |
| Replication | Micro-C experiments were performed in two biological replicates except the experiments without adding DNA ligase and with varying cultivation/fixation temperature and experiments with antibiotics ciprofloxacin, novobiocin and netropsin, which were performed in one repeat. Hi-C and Red-C experiments were performed in two biological replicates. RT-qPCR experiments were performed in at least four biological replicates. RNA-seq experiments were performed in five biological replicates. TEM and PLA experiments were performed in two repeats. All experiments showed high reproducibility between replicates. |
| Randomization | Randomization was not applicable, as the study involved controlled experiments using cell cultures with uniformly applied conditions, eliminating the need for random allocation. |
| Blinding | Blinding was not required in this study, as it did not involve a clinical trial with human participants. |

# Reporting for specific materials, systems and methods

We require information from authors about some types of materials, experimental systems and methods used in many studies. Here, indicate whether each material, system or method listed is relevant to your study. If you are not sure if a list item applies to your research, read the appropriate section before selecting a response.

## Materials & experimental systems

| n/a | Involved in the study |
|-----|----------------------|
| ☒ ☐ | Antibodies |
| ☒ ☐ | Eukaryotic cell lines |
| ☒ ☐ | Palaeontology and archaeology |
| ☒ ☐ | Animals and other organisms |
| ☒ ☐ | Clinical data |
| ☒ ☐ | Dual use research of concern |
| ☒ ☐ | Plants |

## Methods

| n/a | Involved in the study |
|-----|----------------------|
| ☒ ☐ | ChIP-seq |
| ☒ ☐ | Flow cytometry |
| ☒ ☐ | MRI-based neuroimaging |

## Plants

| | |
|---|---|
| Seed stocks | *Report on the source of all seed stocks or other plant material used. If applicable, state the seed stock centre and catalogue number. If plant specimens were collected from the field, describe the collection location, date and sampling procedures.* |
| Novel plant genotypes | *Describe the methods by which all novel plant genotypes were produced. This includes those generated by transgenic approaches, gene editing, chemical/radiation-based mutagenesis and hybridization. For transgenic lines, describe the transformation method, the number of independent lines analyzed and the generation upon which experiments were performed. For gene-edited lines, describe the editor used, the endogenous sequence targeted for editing, the targeting guide RNA sequence (if applicable) and how the editor was applied.* |
| Authentication | *Describe any authentication procedures for each seed stock used or novel genotype generated. Describe any experiments used to assess the effect of a mutation and, where applicable, how potential secondary effects (e.g. second site T-DNA insertions, mosiacism, off-target gene editing) were examined.* |

