## [Peer Review File · Nature]

Elementary 3D organization of active and silenced *E. coli* genome

Corresponding Author: Professor Evgeny Nudler

Version 1:

Reviewer comments:

Referee #1

(Remarks to the Author)

The ms describes the adaptation of microC to investigate the 3D organization of the *E. coli* nucleoid. There have been many studies that have used HiC for this purpose but the resolution of microC is at least one order of magnitude higher. With this resolution, the authors claim to observe novel elementary patterns describing the 3D organization of the *e.coli* genome.

A number of these elementary patterns were already observed in the literature: Higher order structures encompassing interactions within genes were recently described using HiC (Bingaud, 2024) (they are called OPCID here) which also reported plaid-like patterns as in the current study; long-range loops were observed in the *B. subtilis* chromosome (Marbouty, 2015; Dugar, 2022); and hairpins formed by SMCs were also described in the past (Marbouty, 2015; Wang, 2017). Thus, the main novel pattern is that of CHINs and long-range CHIN interactions (here defined as CHIDs) which are hairpins related to transcriptional silencing instead of being related to replication initiation as the hairpins described in *Bacillus*.

This is a very novel discovery and the authors do an excellent job at showing how these patterns change with transcriptional perturbations (heat shock, rifampicin treatment) or by H-NS deletions. However, I have multiple concerns related to interpretation and controls, and to the lack of evidence for many of the statements and conclusions.

Typically, discovery of new chromosome structures is validated with other experimental methods. For instance long-range interactions should be tested using FROS tracking. Large CHINs should be visible by HiC.

Do these structures exist in real life or are they the product of ensemble averaging effects? One can imagine the situation where, for instance CHINs are the product of contacts derived from different *E.coli* cells where proximity between H-NS bound sites exist. In this scenario, there would be no cell where all the contacts necessary for creating a CHID would actually exist. This of course puts into question the existence of CHIDs and CHINs.

It is unclear what the typical sizes of CHINs are, or at what distances they interact to form CHIDs. A quantification of this would be important. From the example provided in the figures (e.g. fig 2, 4) CHINs seem to be 2-10kb in size. If this is the case, why are they not visible by HiC experiments with 100-500bp resolution?

Genetic validation is lacking. For instance, removal of H-NS binding sites within CHINs, or of CHIN borders should remove the CHIN signal from the micro-C map. This would allow the authors to consolidate many of the statements they make about the mechanisms behind the formation of CHINs.

CHIN-like structures were also previously visualized in *Bacillus* (e.g Marbouty, 2015; Wang, Science 2017). The formation of hairpins in *Bacillus* was demonstrated to be mediated by SMC. Similarly, the current study argues that MukBEF may be associated with the formation of CHINs, however it shows no experimental evidence.

The authors state that TSS-terminator contacts are related to transcription. However, $\log(\text{obs/expected})$ TSS-TER contacts are >1 only for a subset of highly transcribed genes (Fig 2g). For the other groups there does not seem to be any correlation. Also, many non-transcribing genes display the same proportions of TSS-TER contacts as transcribing genes (see outliers for the first 3 groups in 2g). Thus, this statement needs to be toned down considerably or further proof should be provided.

The authors state that DNA within CHINs is (+) supercoiled but no evidence is provided. If this was the case, these regions should be topologically insulated from neighboring chromosomal regions. This could be tested in gyrase mutant strains.

I find the conclusions regarding plectonemic DNA or DNA curvature within CHINs is not well supported by data. Therefore, I would recommend to keep this speculation to the discussion of the ms, or to produce data supporting their statements.

While the role of HNS in the formation of CHINs is clear, the role of other important factors, such as Muk-BEF or Fis is not. These could be easily investigated using existing deletion strains.

The conclusion that CHINs and CHIDs establish genome silencing is purely based on a correlation between the locations of these structures and of silenced regions. If the authors want to make such a strong statement, they should provide supporting data testing this hypothesis. The authors hypothesize that CHINs may represent barriers to the extrusion by MukBEF however no evidence is presented. The experiments proposed above would address this question.

(Remarks on code availability)

Referee #2

(Remarks to the Author)

The 3D organization of genomes, from bacteria to humans, has indeed been a growing field of research with exciting advance over the last decade. This was revolutionised by the introduction of whole-genome 3D assays, like Hi-C, and Micro-C and its derivatives now increase resolution to resolve sub-kbp structures. Along these lines, Gavrillov and colleagues apply Micro-C (for the first time, to my knowledge) in a bacterium to identify fine-scale organizational features. This is certainly an improvement compared to previous Hi-C studies. In my view, the Micro-C data quality is high and the improvement in resolution is particularly useful in structures of transcriptionally-inert chromatin. Therein, hairpin like structures (vertical lines) emerge that often also interact with one another. Despite the adequate description of these structures, the manuscript generally lacks sufficient functional advance. For example, although structures like CHIDs/CHINs were not really observed before, the authors only speculate about their function or about the contribution of SMC-like complexes and transcription in their formation (even if secondary). I also see no genomic perturbation experiments that might have revealed novel mechanisms of genome folding, and (to be fair to works by others in the field) most of the insights around the role of active loci and transcription have been largely described before (e.g., see papers by the Koszul and Laub labs). As such, I find that the methodological and resolution advances described here are very welcome and can represent the new standard for bacterial 3D genome studies, but their limited biological insights are also rather incremental and would be better suited for another scientific outlet.

Secondary comments:

- The statement about 10-bp resolution data is only found in the opening paragraph and not really put to use in the data in a way that justifies this resolution being decisive. If, for instance, their effective resolution was 200-bp, would their conclusions be different? And how is this 10-bp resolution defined as adequately achieved?
- In my opinion, the introduction of numerous new acronyms in the manuscript is not helpful. First, because some are derivative (OPCIDs), while other not really necessary.
- The Micro-C to Hi-C comparison is welcome, but the Results text accompanying it is lacking any real numbers and detail that would be useful to readers to understand the extent of the advance achieved.
- Why would CHINs dissolve upon transcriptional inhibition? Are these really a mere secondary effect or is transcription somehow assisting in their shaping?
- In general, the Results text would benefit from more details and quantitative information.

(Remarks on code availability)

There is no aspect of the code that is no novel; this is very much standard Micro-C analysis tools.

Referee #3

(Remarks to the Author)

Gavrillov et al is a technically impressive study that applies Micro-C to study Escherichia coli chromosome organization. Micro-C enables a nucleosome-level portrait of eukaryotic genome organization, but this technology has not yet been adopted in bacteria. With Micro-C, the authors identify numerous new bacterial organizational features at operon and sub-operon resolution. These features include Operon-size Chromosomal Interaction Domains (OPCIDs), Chromosomal Hairpins (CHINs), and Chromosomal Hairpin Interaction Domains (CHIDs). The authors show with a series of beautiful transcription-modulating and inhibiting experiments that OPCIDs are dependent on transcription. Similarly, the authors demonstrate that many CHINs and OPCIDs are altered when the NAP HNS is deleted. Based on these observations, the authors propose that active operon looping enables RNA polymerase (RNAP) recycling and that HNS filaments create chromosomal loops that repress horizontally transferred genes (HTGs).

My enthusiasm for this work is diminished by three issues. First, the central conclusions may have alternative explanations due to technical considerations or a need for additional studies. Second, I find the proposed chromosome organization models are weakly supported by the data and the analysis. Lastly, this work is largely a descriptive study, which is not unusual for the field. However, some of the data has been overinterpreted. A more limited assessment of the data or additional biological experiments would improve the work. Specific comments concerning each issue are below.

Specific comments:

1. The central conclusions may have alternative explanations due to technical considerations or a need for additional studies

1a) The authors used a double-crosslinker approach for Micro-C (formaldehyde and disuccinimidyl glutarate [DSG], Micro-C XL) in the absence of detergent. Is *E. coli* DSG-permeable in these conditions? *E. coli* is known to be non-permeable to similar N-hydroxysuccinimidyl esters (de Jong, et al, J. Protein Res. 2017). Crosslinking controls (e.g. evidence that DSG is crosslinks cytoplasmic proteins) are critical. Additional experiments (e.g. comparisons with a formaldehyde-only Micro-C as in Hsieh, et al Nature Methods, 2016) could also shed light on this issue.

Hsieh et al showed that long-distance interactions were lost without DSG fixation. If DSG is not *E. coli* permeable, the data would under-report chromosome interaction domains (CIDs) and may explain the lack of significant CID signal in this dataset in comparison to Bignaud et al (Nature Methods, 2024; see also point 2c below). Additionally, there may be some issue with the analysis of the Bignaud dataset (Fig. 1E). In particular, the 5 kb data presented here does not have strong CID signal like 5 kb binned dataset presented in Bignaud et al.

1b) The authors do not show that the identified features are dependent on chromosomal contact (a no-ligation control) and are robust to MNase concentration (Micro-C with varying levels of MNase digestion) as in previous Micro-C studies (Hsieh 2015). These studies would help establish this assay in bacteria. This is especially important as some CHINs are similarly sized to the MNase digestion fragments. These CHINs could be MNase-resistant chromosomal loci (e.g. HNS protects against MNase digestion) rather than higher-order chromosome interactions as is proposed.

1c) HNS bridging is temperature dependent, with bridging highest below 30°C and weakening as temperature increases (Kotlajich, et al, eLife 2015). Although the cells were grown at 37°C for Micro-C, the formaldehyde fixation occurred for 30 min at room temperature. Is the Micro-C pattern observed here representative of an HNS-bridged state (room temperature) or an unbridged state (37°C)? Distinguishing between whether the data represents high- or low-temperature growth (or a mixture) is critical for interpreting the biology. Additionally, performing Micro-C at low vs high temperature could provide a natural experiment for studying how bridging influences CHIN formation (see 3b below).

2) the proposed chromosome organization models are weakly supported by the data and the analysis

2a) First, OPCIDs, CHINs, CHIDs were manually annotated. Can these events be computationally predicted? What is the rubric used to call events and how was a true event differentiated from noise? How was the reported 10 bp resolution of Micro-C determined?

2b) The authors note that previous studies show CID boundaries are driven by transcriptional activity (Le et al, Science 2013; Le and Laub, EMBO J 2016; Liou et al, Cell 2018)? How does the OPCID (an operon-sized region of high contact) lead to CID boundaries (a region of low contact)?

2c) In Bignaud, short-range “bundled domains” were associated with transcriptional activity, with these bundled domains interacting with each other to form a “plaid” pattern. Is the short-range bundle the same feature as an OPCID? A comparison should be made.

2d) What is the evidence that the transcription start and end sites within OPCIDs make more frequent contact than other sites within the OPCID? Is looping detectable with published algorithms (Rao et al, Cell 2014, Salameh et al, Nat Comm, 2020, or others)?

2e) How was the co-localization of NAPs with CHINs assessed? In Fig S4, while HNS/StpA signal is found at nearly all CHINs displayed, the MukB signal is only found at a minority (~10%). Conversely, Fig S4 shows clear enrichment of Topo I, GyrA/GyrB, and moderately for Fis in the exact center of CHINs, akin to the profile of HupA/B. Why do the authors report MukB co-localization with CHINs and exclude these alternate possibilities? I also could not understand how the heatmaps in Fig S4 were sorted vertically. Is it by ChIP signal?

2f) Why are the “levels of H-NS” significantly higher at CHINs in an H-NS knockout strain? I also don’t understand Fig S4M, which shows HNS binding in a Δ hns background. How is this possible? What is the significance test used here to determine the increased levels of HNS, StpA, and other NAPs and decreased levels of fis at CHINs in Δ hns?

2g) Overlapping/adjacent CHINs and NAPs could occur together or be mutually exclusive. In the model in 4D why are three CHIN contacts shown as a bundle when they could be mutually exclusive? It is also quite unlikely that all these NAPs would be interacting with CHINs together. 4D also appears to have supercoiling (GapR) upstream and topoisomerases downstream, with topo I and gyrase together which is not biologically probable.

2h) The authors suggest that “spatial contacts between CHINs create DNA loops that could isolate active operons from the

rest of the genome". What is the evidence for this statement? Are CHINs associated with the boundaries of OPCIDs? This should be shown explicitly.

2i) An alternative explanation HNS "looping" could be that each locus bound by HNS becomes more dynamic within the cell, leading to increased interaction with other dynamic (i.e. HNS-bound) loci. This possibility should be acknowledged.

3. some of the data has been overinterpreted. A more limited assessment of the data or additional biological experiments would improve the work

3a) Why do the authors propose that "H-NS is the top manager of CHINs"? I agree that some CHINs disappear in Δ hns cells, but without knowing how deletions in other NAPs affect CHINs, a dominant role for HNS seems premature.

3b) The authors show that CHIN-associated genes are more upregulated in Δ hns but concluding that CHINs are responsible for silencing is premature. There is extensive literature about the role of HNS and other NAPs silencing HTGs (Ueguchi and Mizuno, EMBO J 1993; Amemiya et al, EMBO J 2021; and many others). Given that HNS bridging is temperature sensitive and high levels of HNS in vitro also weaken bridging (Kotlajich, et al, eLife 2015), an in vivo experiment disrupting bridging could clarify if HNS underlies CHIN formation and silencing.

3c) As the authors note, Micro-C assays represent the population average organization. These events could be population-wide or may be restricted to a small subpopulation of cells. Validating some of their contacts with an alternative experiment (e.g. fluorescence repressor operator system, in-situ hybridization, etc) could help clarify the likelihood of these events.

3d) The authors suggest that CHINs/CHIDs co-localizes distant HTGs to mediate recombination. However, chromosome contact frequency is not completely predictive of spatial distance and discrepancies can be large (Fudenberg et al, Nature Methods, 2017; Williamson et al, Genes Dev 2014). An in vivo experiment directly measuring co-localization is necessary to make these claims.

Minor notes:

1) The extensive literature documenting exclusion of HNS and other NAPs from active transcription units should be cited (Peters et al, Genes Dev, 2012; Singh et al, Genes Dev, 2014; Freddolino et al, PLOS Biology 2021; Amemiya et al, EMBO J 2021, and others).

2) CHIN boundaries "likely [arising] from the sharp difference GC content between the CHID and its flanking regions" is too strong of a statement. There could be many mechanisms (proteins, supercoiling, transcription, etc) that could influence CHIN boundaries, with GC potentially being a secondary consequence.

(Remarks on code availability)

Version 2:

Reviewer comments:

Referee #1

(Remarks to the Author)

The authors have appropriately addressed all my concerns. Congratulations on a beautiful study.

Referee #2

(Remarks to the Author)

The authors have made significant efforts to address reviewers' comments experimentally or via textual changes. I still believe that the technical advance portrayed here is much welcome and will set a new standard of using Micro-C for bacterial 3D genomics, as I also still believe that experiments and analyses are of quite a high standard, and that the implications on HTGs are indeed interesting. At the same time though, I cannot fully steer away from the opinion that the conceptual advance, simply due to higher resolution structures, is not as large as presented here. To cite an example, OPCIDs are cited as structures that "might enable RNA Pol recycling" something that was proposed for promoter-terminator loops in human cells many years ago (e.g., work by the Proudfoot lab) without ever being really substantiated by hard evidence on such "recycling". A much simpler explanation would be that the authors are capturing variable conformations in single cells that are a result of variable polymerase positioning among the many alleles averaged out in their Micro-C data. Similarly, how various of these structures crosstalk with other "bundled domains", in other words how higher-res structures might underlie higher-order ones, is left unclear. Therefore, I remain borderline in my appraisal here, again, despite the high technical quality of the work.

Referee #3

(Remarks to the Author)

I commend the authors for their extensive efforts at addressing my earlier concerns. I find that the new experiments including:

critical Micro-C controls, proximity ligation assay (PLA), data clarifying the role of different nucleoid associated proteins (NAPs) and DNA supercoiling on CHIN formation, and studies of H-NS assembly on CHIN DNA by electron microscopy greatly improve the work.

1) Some of the conclusions remain unsupported by the data. There are two main issues:

- a. First, there is no evidence that the looping of transcription start and end sites promotes RNAP recycling and high transcription output, which is claimed as a major finding. Although looping is observed more frequently at highly expressed operons (Figure 2G), this does not mean that looping causes high expression. Looping may be a consequence of expression, due to transcription-dependent twin supercoiling domains or supercoiling-recruited nucleoid associated proteins “looping” DNA together. This interpretation is supported by the data showing enrichment of Hi-C contacts between the upstream and downstream regions of a transcription unit (Bignaud et al, 2024). All statements about RNAP recycling should be conservatively amended given this likely alternative possibility.
- b. Second, there is no evidence that CHINs facilitate “the spatial proximity of distant HTGs...[explaining] the previously observed high frequency of recombination between these genes, both in cis and trans”. No evidence is shown about association frequency of distant HTGs let alone if CHINs mediate this association. Indeed, CHINs contact is shown to be generally ~2-8 kb on average, suggesting any that CHIN-mediated recombination is restricted to within a gene or a local region. Furthermore, the diagram for this recombination (Figure 5b) again does not show distant contacts. All statements about recombination should be amended to indicate that this is a short-range phenomenon.

2) I like the new analysis presented in Figure 2G to distinguish between looping of transcription start/end sites and looping within a transcription unit. However, the triangle representing sites in the experiment is difficult to interpret. Can the authors use a linear representation like in Figure 2F instead?

3) I still am not satisfied as to why the authors suggest MukB and positive supercoiling (as inferred by GapR binding) is associated with CHINs. In Figure S2C, it appears that MukB is associated with ~10-20% of CHINs and the remaining >80% of CHINs have a MukB profile akin to that for active and inactive operons. Similarly, in Figure S2F, it appears that GapR is associated with perhaps 5% of CHINs at best. In both these cases, it seems much more likely that these profiles are random chance or that there is an association with transcription of CHINs rather than the CHIN itself.

Version 3:

Reviewer comments:

Referee #2

(Remarks to the Author)

Regarding the technical side of the work, I have no further suggestions.

The points pertaining to data interpretation and conceptual advance have also been highlighted by textual changes.

Referee #3

(Remarks to the Author)

The authors have satisfied all of my concerns. Kudos to the authors for their efforts and their intricate work. I have no further comments.

REFEREE #1 (Remarks to the Author):

The ms describes the adaptation of microC to investigate the 3D organization of the E. coli nucleoid. There have been many studies that have used HiC for this purpose but the resolution of microC is at least one order of magnitude higher. With this resolution, the authors claim to observe novel elementary patterns describing the 3D organization of the e.coli genome.

A number of these elementary patterns were already observed in the literature: Higher order structures encompassing interactions within genes were recently described using HiC (Bingaud, 2024) (they are called OPCID here) which also reported plaid-like patterns as in the current study; long-range loops were observed in the B. subtilis chromosome (Marbouty, 2015; Dugar, 2022); and hairpins formed by SMCs were also described in the past (Marbouty, 2015; Wang, 2017). Thus, the main novel pattern is that of CHINs and long-range CHIN interactions (here defined as CHIDs) which are hairpins related to transcriptional silencing instead of being related to replication initiation as the hairpins described in Bacillus.

This is a very novel discovery and the authors do an excellent job at showing how these patterns change with transcriptional perturbations (heat shock, rifampicin treatment) or by H-NS deletions. However, I have multiple concerns related to interpretation and controls, and to the lack of evidence for many of the statements and conclusions.

We thank the reviewer for their helpful comments and their enthusiasm for the novel findings of the manuscript. We have resolved the issues with experimental design and analysis as detailed below. All textual changes in the revised manuscript are also highlighted in blue.

Typically, discovery of new chromosome structures is validated with other experimental methods. For instance long-range interactions should be tested using FROS tracking. Large CHINs should be visible by HiC.

As suggested by the reviewer, in the revised manuscript, we demonstrate that some CHINs are detectable in Hi-C data as well (**new Extended Data Fig. 1b**). We describe this in the following sentence: “*While traces of these structures are detectable in Hi-C maps (Extended data Fig. 1b), the higher resolution of our Micro-C maps offers more detailed insights (Fig. 1)*”. Thus, CHINs can be identified using two methods – Micro-C and Hi-C – with the former providing superior resolution and structural details.

We note that fluorescent assays such as FROS or FISH lack the resolution to detect small structures like CHINs, whose median length is approximately 2 kb and rarely exceed 12 kb, as shown in **revised Fig. 3d**. CHIN-CHIN interactions also occur over relatively short genomic distances, typically less than 40 kb (**new Fig. 3e**). In contrast, FROS and FISH are generally for probing much greater distances (e.g.: PMID 35420890, 28774286, 38509385, 35145120), with a resolution limit of ~100 kb at best (PMID

31124784, 39237524). Thus, FROS or FISH are not ideal methods for verifying the small-scale structures such as CHINs identified by Micro-C.

To independently validate the existence of CHINs *in vivo*, we employed proximity ligation assay (PLA), a technique widely used to detect protein-protein proximity in cells (PMID 30238640). We targeted two His-tagged dCas9 molecules to the base regions of a long CHIN and assessed their spatial proximity using a standard PLA protocol. This analysis revealed the dCas9 molecules were indeed in close proximity in a significant fraction of cells (**new Extended Data Fig. 5a-d**).

Furthermore, we reconstituted CHIN structures *in vitro* by incubating purified H-NS protein with a DNA fragment known – based on Micro-C maps – to form a CHIN (**new Extended Data Fig. 5e-j**), further reinforcing the key role of H-NS in CHIN formation.

The following sentences were added to the Results: *“To validate the existence of CHINs in vivo using an orthogonal approach, we performed proximity ligation assay (PLA) analysis in conjunction with super-resolution microscopy. Four genomic loci were selected for recruitment of dCas9-based PLA probes (Extended Data Fig. 5a,b). One probe (“C”) was positioned near the base of a long CHIN. A second probe (“D”) was placed on the opposite shoulder of CHIN, 6 kb away from probe “C”, such that both probes would be positioned across from each other within the CHIN stem when assembled. A third probe (“A”) was located 6 kb away from probe “C” but outside of the CHIN region, serving as a negative control. Finally, a fourth probe (“B”) was placed near probe “C” to serve as a positive control. We hypothesized that probes “C” and “D”, brought into proximity by CHIN folding, would yield a PLA signal, whereas probes “A” and “C”, 6 kb apart on linear DNA not involved in CHIN formation, would not. Probes “B” and “C”, positioned adjacently on DNA, were expected to yield a PLA signal regardless of 3D organization. The experimental results (Extended Data Fig. 5c,d) confirmed our predictions: while probes “A” and “C” produced only sparse PLA signals, probes “C” and “D” generated a robust signal in a substantial fraction of cells (~50% of the positive control). These results indicate that the CHIN is formed in at least half of the cells in the examined population within the timeframe of the PLA experiment”.*

And further: *“We also attempted to reconstitute CHINs in vitro by incubating a plasmid containing a DNA fragment known to form CHINs in vivo with purified H-NS. CHIN-like structures of varying lengths readily assembled on this fragment, as observed by transmission electron microscopy, but not on a control plasmid of similar length carrying a non-CHIN-forming DNA fragment, or on the CHIN-containing plasmid in the absence of H-NS (Extended Data Fig. 5g-j)”.*

The following additions were made to the Discussion: *“It should be noted, however, that Micro-C analysis does not capture the frequency or duration of these contacts across individual cells; interactions between H-NS-bound loci may occur only in a subset of the population at any given time. Nevertheless, the existence of CHINs was independently validated using PLA, which confirmed that the CHIN base regions identified by Micro-C are indeed in close spatial proximity in a substantial proportion of cells (Extended Data Fig. 5a-d)”.* And further: *“The central role of H-NS in CHIN formation was further supported by in vitro reconstitution experiments, in which CHIN-like structures readily assembled upon incubating purified H-NS with a plasmid containing a DNA fragment known to form CHIN in vivo (Extended Data Fig. 5g-j)”.*

Do these structures exist in real life or are they the product of ensemble averaging effects? One can imagine the situation where, for instance CHINs are the product of contacts derived from different E.coli cells where proximity between H-NS bound sites exist. In this scenario, there would be no cell where all the contacts necessary for creating a CHID would actually exist. This of course puts into question the existence of CHIDs and CHINs.

We agree that the structures observed in the Micro-C map may represent population averages. In the initial manuscript, we acknowledged this by stating: *“It is worth noting that all CHINs within a CHID cannot coexist simultaneously at the same locus due to overlapping regions. Therefore, CHIDs represent a population average of alternative CHINs formed in individual cells”*. To further emphasize this point, we have added: *“Some of the other complex interaction patterns observed in our Micro-C maps may also reflect population averages”*.

The possibility that a CHIN represents a computer-generated average pattern is highly unlikely. If this were the case, we would expect heterogeneity in contact intensity along the CHIN stem in the Micro-C contact map, as different DNA sites would not interact with identical frequencies. Instead, all CHINs exhibit a perfectly symmetric signal (perpendicular to the diagonal, symmetric to the central axis), with a gradual decrease in intensity away from the diagonal. This distinctive symmetry, clearly visible in **new Extended Data Fig. 1a**, argues against the existence of stable, under-assembled entities and supports an “all-or-nothing” assembly model—where a CHIN is either fully assembled from head to base or not at all in a given cell at a given genomic position.

It is unclear what the typical sizes of CHINs are, or at what distances they interact to form CHIDs. A quantification of this would be important. From the example provided in the figures (e.g. fig 2, 4) CHINs seem to be 2-10kb in size.

Revised Fig. 3d,e (former **Fig. 4d**) now present the distribution of CHIN sizes and distances at which they interact to form loops or stem loops. The medium CHIN size is approximately 2 kb. Importantly, CHIDs are not formed by the interaction of CHINs; rather, they are clusters of CHINs that arise nearby within the same genomic location. Overlapping CHINs cannot coexist simultaneously and likely represent a population average, as discussed above.

In the revised manuscript, we have also corrected the incorrect expansion of the CHID abbreviation from CHIN Interaction Domain to “CHIN Domain”.

If this is the case, why are they not visible by HiC experiments with 100-500bp resolution?

We now demonstrate that some of the CHINs are indeed detectable in the HiC data at 200 bp resolution (**new Extended Data Fig. 1b**).

Genetic validation is lacking. For instance, removal of H-NS binding sites within CHINs, or of CHIN borders should remove the CHIN signal from the micro-C map. This would allow the authors to consolidate many of the statements they make about the mechanisms behind the formation of CHINs.

Given the filamentous nature of H-NS-DNA complexes, with multiple H-NS binding sites within a single CHIN and a relatively flexible H-NS binding consensus, identifying a specific H-NS binding site whose deletion would eliminate the entire CHIN appears challenging, if not impossible. Deleting the entire DNA region underlying a CHIN would remove the CHIN from the contact map, but this would hardly serve as conclusive evidence.

To address the reviewer's concerns, we conducted a competition assay using netropsin – a minor groove binder of AT-rich DNA (Gordon et al., PNAS 2011) – to displace H-NS (StpA) from DNA. Under these conditions, both CHINs and CHIDs disappeared (**new Fig. 3o**). Similarly, genetic disruption of both H-NS and StpA, histone-like proteins that strictly colocalize with CHINs (**Fig. 3a**), resulted in the complete elimination of CHINs/CHIDs (**new Figs. 3l, 4j**), further supporting the H-NS/StpA-driven mechanism of CHIN/CHID formation.

CHIN-like structures were also previously visualized in *Bacillus* (e.g Marbouty, 2015; Wang, Science 2017). The formation of hairpins in *Bacillus* was demonstrated to be mediated by SMC. Similarly, the current study argues that MukBEF may be associated with the formation of CHINs, however it shows no experimental evidence.

In the revised manuscript, we demonstrate that CHINs are unaffected by the loss of MukBEF. The new results are described as follows: “*Similarly, knockouts of Fis and mukBEF had no impact on CHINs and CHIDs (Fig. 3m,n)*”. Therefore, we have removed any statements suggesting a potential role for MukBEF in CHIN formation from the revised manuscript.

The authors state that TSS-terminator contacts are related to transcription. However, $\log(\text{obs/expected})$ TSS-TER contacts are >1 only for a subset of highly transcribed genes (Fig 2g). For the other groups there does not seem to be any correlation. Also, many non-transcribing genes display the same proportions of TSS-TER contacts as transcribing genes (see outliers for the first 3 groups in 2g). Thus, this statement needs to be toned down considerably or further proof should be provided.

Indeed, the elevated frequency of promoter-terminator contacts is observed exclusively in highly expressed operons, and we have framed our discussion accordingly. In the revised manuscript, we further demonstrate that the frequency of promoter-terminator contacts is significantly higher than that of contacts between internal operon sites – an effect observed only in highly expressed operons (**revised Fig. 2g**), which is consistent with facilitated RNAP recycling.

The authors state that DNA within CHINs is (+) supercoiled but no evidence is provided. If this was the case, these regions should be topologically insulated from neighboring chromosomal regions. This could be tested in gyrase mutant strains.

Our conclusion that DNA within CHINs is positively supercoiled was based on the ChIP-seq profile of GapR around CHINs (**Extended Data Fig. 4h**), a protein that senses positive supercoils (Guo et al, eLife 2021). In the revised manuscript, we tested the effects of two gyrase inhibitors, ciprofloxacin and novobiocin, as well as bleomycin – a drug that introduces single-stranded breaks and eliminates supercoiling – on CHIN integrity. We show that none of these treatments disrupt CHIN patterns, leading us to conclude that the existence of CHINs and CHIDs is independent of DNA supercoiling. These findings are detailed in the new section, “The integrity of CHINs and CHIDs is independent of DNA superhelicity”.

I find the conclusions regarding plectonemic DNA or DNA curvature within CHINs is not well supported by data. Therefore, I would recommend to keep this speculation to the discussion of the ms, or to produce data supporting their statements.

The statement about plectonemic DNA is mentioned solely in the context of comparing our findings with existing literature: “*This observation aligns with previous reports suggesting that, at identical bin sizes, Micro-C provides more details than Hi-C matrices*^{13,44}”. The clear enrichment of HupA/B over the CHIN loop region (**Extended Data Fig. 4i,j**) strongly supports the model of CHIN structure depicted in **Fig. 3d**, suggesting that DNA is curved in this portion of CHIN.

While the role of HNS in the formation of CHINs is clear, the role of other important factors, such as Muk-BEF or Fis is not. These could be easily investigated using existing deletion strains.

In the revised manuscript, we examined the role of MukBEF and Fis in CHIN formation by performing Micro-C in clean deletion strains that we generated and validated (see Methods and **new Extended Data Fig. 7a**). Our results confirm that CHINs can still be detected in the absence of these factors (**new Fig. 3m,n**). Thus, we stated: “*Similarly, knockouts of Fis and mukBEF had no impact on CHINs and CHIDs (Fig. 3m,n)*”.

The conclusion that CHINs and CHIDs establish genome silencing is purely based on a correlation between the locations of these structures and of silenced regions. If the authors want to make such a strong statement, they should provide supporting data testing this hypothesis. The authors hypothesize that CHINs may represent barriers to the extrusion by MukBEF however no evidence is presented. The experiments proposed above would address this question.

At the beginning of the section “CHINs and CHIDs are assembled on horizontally transferred genes”, we state: “*CHINs and CHIDs reside in non-transcribed areas of the*

genome (Fig. 2b,c) and likely serve as basic repressive structures. However, not all repressed genes reside in these domains". Indeed, we observe that genomic regions assembled in CHINs and CHIDs are consistently transcriptionally silent, supporting their suppressive nature.

In the initial manuscript, we showed that H-NS knockout, which disrupts many CHINs, activates genes within these CHINs, primarily HTGs (now **Fig. 4**). In the revised version, we show that the simultaneous knockout of H-NS and StpA completely abolishes *all* CHINs and CHIDs, leading to robust and widespread activation of HTGs (**new Fig. 4a-j**). Furthermore, we show that the exposure of cells to netropsin, a DNA-binding competitor of H-NS and StpA, mimics the effects of H-NS/StpA double knockout, resulting in dramatic HTG de-repression (**new Fig. 4b**). These findings strongly support the role of CHINs and CHIDs in establishing localized genome silencing.

REFEREE #2 (Remarks to the Author):

We thank the reviewer for their careful reading of our work and for recognizing the wealth of interesting observations, as well as the methodological and resolution advances that can establish "*the new standard for bacterial 3D genome studies*". In response to their feedback, we have now provided substantial additional data to further strengthen the new biological insights revealed in this study.

The 3D organization of genomes, from bacteria to humans, has indeed been a growing field of research with exciting advance over the last decade. This was revolutionised by the introduction of whole-genome 3D assays, like Hi-C, and Micro-C and its derivatives now increase resolution to resolve sub-kbp structures. Along these lines, Gavrillov and colleagues apply Micro-C (for the first time, to my knowledge) in a bacterium to identify fine-scale organizational features. This is certainly an improvement compared to previous Hi-C studies. In my view, the Micro-C data quality is high and the improvement in resolution is particularly useful in structures of transcriptionally-inert chromatin. Therein, hairpin like structures (vertical lines) emerge that often also interact with one another. Despite the adequate description of these structures, the manuscript generally lacks sufficient functional advance. For example, although structures like CHIDs/CHINs were not really observed before, the authors only speculate about their function or about the contribution of SMC-like complexes and transcription in their formation (even if secondary). I also see no genomic perturbation experiments that might have revealed novel mechanisms of genome folding, and (to be fair to works by others in the field) most of the insights around the role of active loci and transcription have been largely described before (e.g., see papers by the Koszul and Laub labs). As such, I find that the methodological and resolution advances described here are very welcome and can represent the new standard for bacterial 3D genome studies, but their limited biological insights are also rather incremental and would be better suited for another scientific outlet.

In the revised manuscript, we present new Micro-C analysis on *E. coli* stains with H-NS, StpA, or both H-NS and StpA knocked out, demonstrating the *central* role of H-NS in shaping CHINs and CHIDs and exerting their repressive functions, particularly in relation to horizontally transferred genes (HTGs) (**new/revised Fig. 3i-l, Fig. 4**). We also show that the antibiotic netropsin, which displaces H-NS and StpA from DNA (Gordon et al., PNAS 2011), produces similar effects (**new Fig. 3o, Fig. 4b**). These results establish H-NS-driven CHINs/CHIDs as primary DNA structural elements that silence foreign, often toxic, genomic loci. Failure to perform this function results in severe growth defect (**new Fig. 4k**).

By disrupting Fis and MukBEF (bacterial SMC) (see Methods), we demonstrate that these factors are not primarily involved in CHIN/CHID formation or maintenance (**new Fig. 3m,n**). Using antibiotics that relieve supercoiling (bleomycin) or inhibit DNA gyrase (ciprofloxacin, novobiocin), we show that CHINs and CHIDs are independent of DNA superhelicity (**new Fig. 3p-r**).

We also clarify the principal distinction between the OPCID structures identified in this study and the transcription-induced bundled domains previously reported by Koszul: *“Our ultra-high-resolution Micro-C analysis has uncovered operon-linked OPCIDs that form in a strictly transcription-dependent manner. Unlike previously described bundled domains¹¹, OPCIDs are characterized by typical contact domains (squares on contact maps) where all regions are in contact with one another. OPCIDs likely represent a superposition of microloops and coiled-coil structures that develop within transcribed regions of individual cells. The high frequency of direct promoter-terminator interactions within an OPCID (Fig. 2g) is expected to facilitate the recycling of RNAP (Fig. 5c), thereby enhancing transcription efficiency”*.

Furthermore, we provide new data confirming the increased contact frequency between promoters and terminators of the most highly expressed operons (**new Fig. 2g**), suggesting a novel model for efficient RNAP recycling to supports robust transcriptional output.

Finally, we validated the existence of CHINs using orthogonal approaches both *in vivo* and *in vitro* (**new Extended Data Fig. 5**).

Overall, the findings presented in the revised manuscript offer new functional and mechanistic insights into bacterial gene silencing and regulation.

Secondary comments:

- The statement about 10-bp resolution data is only found in the opening paragraph and not really put to use in the data in a way that justifies this resolution being decisive. If, for instance, their effective resolution was 200-bp, would their conclusions be different? And how is this 10-bp resolution defined as adequately achieved?

In the revised version of the manuscript, we compare how Micro-C map appears at different resolutions, ranging from 1 kb down to 5 bp (**new Extended Data Fig. 1a**). Using CHIN structures as an example, we show that these features remain clearly visible at resolutions as high as 10-20 bp but begin to dissipate at finer resolutions. Consequently,

we chose 10 bp as the optimal resolution for refinement, as it represents the last “10 number” (1000, 100, 10...) where the structures are still discernible.

We also compare average contact maps generated from Micro-C data at 10 bp and 200 bp resolution, showing that the higher resolution provides clearer structural details (**new Extended Data Fig. 1c,d**).

The results are described in the following sentences:

Regarding 10 bp resolution: “*The generation of a large number of reads (~1.5 B for main samples) and the relatively small size of the E. coli genome enabled us to construct DNA-DNA contact maps with a resolution of up to 10 bp. At this resolution, CHINs remain detectable in contact maps; however, further increases in resolution result in the structures becoming less discernible (Extended Data Fig. 1a)*”.

Regarding the comparison of 10 and 200 bp data: “*The average contact maps were generated from Micro-C contact maps with a resolution of 10 bp. Using lower-resolution Micro-C maps for plotting the average map leads to more blurred features and thicker diagonal (Extended Data Fig. 1c,d)*”.

- In my opinion, the introduction of numerous new acronyms in the manuscript is not helpful. First, because some are derivative (OPCIDs), while other not really necessary.

We have identified new genomic structures that have not been observed previously, necessitating the introduction of new terminology. Specifically, the use of the term “OPCIDs” is justified, as these structures are fundamentally distinct from TIDs and bundled domains described in earlier studies. Our analysis, for the first time, reveals operon-sized square contact domains within the spatial contact matrices of the *E. coli* genome, and these domains are strictly transcription-dependent.

- The Micro-C to Hi-C comparison is welcome, but the Results text accompanying it is lacking any real numbers and detail that would be useful to readers to understand the extent of the advance achieved.

We do not emphasize Hi-C data analysis, as most of the structures described in the manuscript are not detectable using Hi-C, as shown in the introductory section of the Results (**Fig. 1**). However, in the revised manuscript, we include a new analysis demonstrating that some CHINs can be observed in both Micro-C and Hi-C data (**new Extended Data Fig. 1b**).

In the Discussion, we elaborate on why Hi-C analysis fails to capture structures visible in Micro-C maps: “*The critical ability to observe these intricacies hinges on the proximity ligation step in the Micro-C protocol (Extended Data Fig. 9a), confirming that these structures indeed represent spatial interactions among distant genomic elements. While traces of these structures are detectable in Hi-C maps (Extended data Fig. 1b), the higher resolution of our Micro-C maps offers more detailed insights (Fig. 1). This improved resolution may arise from the more uniform distribution of DNA cuts introduced by MNase, along with the absence of detergent treatment of the cells in the Micro-C protocol (see Methods)*”.

Later in the Discussion, we address the relationship between OPCIDs and bundled domains: *“The relationship between OPCIDs and bundled domains¹¹ remains to be fully elucidated. Currently, we cannot rule out the possibility that the Hi-C protocol used in¹¹ may fail to detect all interactions within operons due to the use of detergents that could disrupt liquid condensates”*.

- Why would CHINs dissolve upon transcriptional inhibition? Are these really a mere secondary effect or is transcription somehow assisting in their shaping?

We consider this to be a secondary effect resulting from the eventual depletion of the H-NS pool at high rifampicin doses that completely halt transcription: *“Since CHINs and CHIDs localize to non-transcribed regions and the association of H-NS with these structures is unaffected by low doses of Rif (Extended Data Fig. 4k), the reduced intensity of CHINs and CHIDs observed at high Rif may be a secondary effect caused by the eventual depletion of the H-NS pool and other scaffolding proteins”*.

- In general, the Results text would benefit from more details and quantitative information.

In the revised manuscript, we included more quantitative analysis. Specifically, we provided data showing how Micro-C performs at different map resolutions (**new Extended Data Fig. 1a,c,d**), showed the distribution of CHIN sizes and distances at which they interact (**new Fig. 3d,e**), and illustrated quantitative effects of transcriptional changes in H-NS and StpA knockout strains as well as in cells treated with netropsin, alongside the corresponding effects on CHIN and CHIDs (**new Fig. 4j**) and cell growth (**new Fig. 4k**). Also, we compared manually annotated CHIDs with those identified by the ChromSight tool (Matthey-Doret et al., Nat Commun 2020), confirming consistency between quantitative approaches (**new Extended Data Fig. 1e-g**).

Referee #2 (Remarks on code availability):

There is no aspect of the code that is no novel; this is very much standard Micro-C analysis tools.

Although we deposit the code in the GitHub repository (https://github.com/irzhegalova/ecoli_microc), we acknowledge that the code primarily relies on previously developed tools. Therefore, we have unchecked the box stating “Did you develop code that is central to the main work described in this manuscript?” during the current submission.

REFEREE #3 (Remarks to the Author):

Gavrilov et al is a technically impressive study that applies Micro-C to study Escherichia coli chromosome organization. Micro-C enables a nucleosome-level portrait of eukaryotic genome organization, but this technology has not yet been adopted in bacteria. With Micro-C, the authors identify numerous new bacterial organizational features at operon and sub-operon resolution. These features include Operon-size Chromosomal Interaction Domains (OPCIDs), Chromosomal Hairpins (CHINs), and Chromosomal Hairpin Interaction Domains (CHIDs). The authors show with a series of beautiful transcription-modulating and inhibiting experiments that OPCIDs are dependent on transcription. Similarly, the authors demonstrate that many CHINs and OPCIDs are altered when the NAP HNS is deleted. Based on these observations, the authors propose that active operon looping enables RNA polymerase (RNAP) recycling and that HNS filaments create chromosomal loops that repress horizontally transferred genes (HTGs).

My enthusiasm for this work is diminished by three issues. First, the central conclusions may have alternative explanations due to technical considerations or a need for additional studies. Second, I find the proposed chromosome organization models are weakly supported by the data and the analysis. Lastly, this work is largely a descriptive study, which is not unusual for the field. However, some of the data has been overinterpreted. A more limited assessment of the data or additional biological experiments would improve the work. Specific comments concerning each issue are below.

We thank the reviewer for their helpful comments and acknowledging that our work is “*technically impressive*”, identifying and validating “*numerous new bacterial organizational features at operon and sub-operon resolution*”. We have addressed all of the reviewer’s concerns as detailed below. All textual changes in the revised manuscript are also highlighted in blue.

Specific comments:

1. The central conclusions may have alternative explanations due to technical considerations or a need for additional studies

- 1a) The authors used a double-crosslinker approach for Micro-C (formaldehyde and disuccinimidyl glutarate [DSG], Micro-C XL) in the absence of detergent. Is E. coli DSG-permeable in these conditions? E. coli is known to be non-permeable to similar N-hydroxysuccinimidyl esters (de Jong, et al, J. Protein Res. 2017). Crosslinking controls (e.g. evidence that DSG is crosslinks cytoplasmic proteins) are critical. Additional experiments (e.g. comparisons with a formaldehyde-only Micro-C as in Hsieh, et al Nature Methods, 2016) could also shed light on this issue. Hsieh et al showed that long-distance interactions were lost without DSG fixation. If DSG is not E. coli permeable, the data would under-report chromosome interaction domains (CIDs) and may explain the lack of significant CID signal in this dataset in comparison to Bignaud et al (Nature Methods, 2024; see also point 2c below). Additionally, there may be some issue with the analysis of

the Bignaud dataset (Fig. 1E). In particular, the 5 kb data presented here does not have strong CID signal like 5 kb binned dataset presented in Bignaud et al.

In the revised version of the manuscript, we employed mass spectrometry analysis to demonstrate that DSG induces *in vivo* protein-protein cross-links with a similar efficiency to those generated by DSS, a well-established *E. coli*-permeable cross-linker (PMID 37196657, 35355008). These results are shown in the **new Extended Data Fig. 9e** and discussed in the following sentence: “*To assess if DSG cross-linking is effective in E. coli, we compared it to DSS, a cell-permeable cross-linker commonly used in E. coli studies*^{71,72}. *Both cross-linkers produced a similar number of protein-protein cross-links in E. coli cells (Extended Data Fig. 9e)*”.

We also show in **new Extended Data Fig. 9f** that the addition of DSG, in combination with formaldehyde, is crucial for the Micro-C procedure to work in *E. coli* cells. We state: “*Also, the inclusion of DSG was essential for stabilizing Micro-C material, as MNase treatment of cells fixed with formaldehyde alone resulted in the solubilization of all DNA fragments (Extended Data Fig. 9f)*”.

We now provide the Micro-C and Hi-C maps using a slightly modified color scale, where CIDs similar to those observed by Bignaud et al. are clearly visible in both Hi-C and Micro-C contact matrices at 5 kb resolution (**revised Fig. 1c-e**). We have included the following sentence at the beginning of the Results: “*At 5 kb resolution (Fig. 1c), our Micro-C map recapitulates previously reported CIDs*¹¹”.

1b) The authors do not show that the identified features are dependent on chromosomal contact (a no-ligation control) and are robust to MNase concentration (Micro-C with varying levels of MNase digestion) as in previous Micro-C studies (Hsieh 2015). These studies would help establish this assay in bacteria. This is especially important as some CHINs are similarly sized to the MNase digestion fragments. These CHINs could be MNase-resistant chromosomal loci (e.g. HNS protects against MNase digestion) rather than higher-order chromosome interactions as is proposed.

A “no ligation” control is now included as **new Extended Data Fig. 9a** and discussed in the beginning of the Discussion: “*The critical ability to observe these intricacies hinges on the proximity ligation step in the Micro-C protocol (Extended Data Fig. 9a), confirming that these structures indeed represent spatial interactions among distant genomic elements*”.

We also show that the ability to detect contact patterns is robust across varying degrees of MNase digestion. These new results are discussed in Methods section, where we describe optimization of MNase amount for Micro-C experiments. The following sentences were added: “*Samples showing excessive DNA degradation, characterized by reduced DNA yield, were typically discarded (e.g., sample 4 rep1 in Extended Data Fig. 9h). However, even heavily over-digested samples retained characteristic Micro-C contact patterns (Extended Data Fig. 9i), demonstrating the robustness of the procedure despite varying levels of MNase digestion*”.

1c) HNS bridging is temperature dependent, with bridging highest below 30°C and weakening as temperature increases (Kotlajich, et al, eLife 2015). Although the cells were grown at 37°C for Micro-C, the formaldehyde fixation occurred for 30 min at room temperature. Is the Micro-C pattern observed here representative of an HNS-bridged state (room temperature) or an unbridged state (37°C)? Distinguishing between whether the data represents high- or low-temperature growth (or a mixture) is critical for interpreting the biology. Additionally, performing Micro-C at low vs high temperature could provide a natural experiment for studying how bridging influences CHIN formation (see 3b below).

In the revised manuscript, we obtained Micro-C maps from *E. coli* cells cultivated at 37 °C and fixed with formaldehyde at room temperature, as well as from cells cultivated and fixed at 37 °C and cells cultivated and fixed at room temperature (**new Extended Data Fig. 9b-d**). Our analysis revealed no significant differences among these conditions. This is discussed in the following paragraph: *“Previous studies have shown that H-NS bridging is temperature dependent, with maximum bridging observed below 30 °C⁷⁰. To determine whether the detection of H-NS dependent features in Micro-C contact maps depends on cultivation or fixation temperature, we compared contact maps for cells cultivated and fixed at room temperature, cells cultivated and fixed at 37 °C, and cells cultivated at 37 °C but fixed at room temperature following the standard protocol. No differences in CHINs intensity were observed between the three conditions (Extended Data Fig. 9b-d)”*.

2) the proposed chromosome organization models are weakly supported by the data and the analysis

2a) First, OPCIDs, CHINs, CHIDs were manually annotated. Can these events be computationally predicted? What is the rubric used to call events and how was a true event differentiated from noise? How was the reported 10 bp resolution of Micro-C determined?

In the revised manuscript, we performed computer annotation of CHINs using the Chromosight tool previously developed by Koszul and colleagues (Matthey-Doret et al., Nat Commun 2020). We showed a strong concordance between the positions of CHINs detected manually and those identified by the computer algorithm. The results are presented in **new Extended Data Fig. 1e-g** and are summarized in the following sentences: *“To validate the accuracy of the manual annotation of CHINs, we also employed the Chromosight program⁷⁶ for annotation (Supplementary Table 8). We found a good correlation between the positions of the manually annotated and Chromosight-annotated CHINs (Extended Data Fig. 1e,f). The discrepancies between the two sets may be attributed to false positives and false negatives present in the Chromosight dataset (Extended Data Fig. 1e). Notably, CHINs identified by Chromosight also exhibit the characteristic bimodal distribution of H-NS around them (Extended Data Fig. 1g)”*.

Also, we provide justification for selecting an ultimate Micro-C map resolution of 10 bp. By generating Micro-C maps at varying resolutions, we show that CHINs can still be

detected at resolutions of up to 10-20 bp, whereas lower resolutions result in the dissipation of these structures (**new Extended Data Fig. 1a**). Using average contact maps as an example, we compare the findings from Micro-C maps at 10 bp and 200 bp resolution, showing that the higher resolution facilitates the detection of clearer structures (**new Extended Data Fig. 1c,d**).

The results are described in the following sentences:

Regarding 10 bp resolution: *“The generation of a large number of reads (~1.5 B for main samples) and the relatively small size of the E. coli genome enabled us to construct DNA-DNA contact maps with a resolution of up to 10 bp. At this resolution, CHINs remain detectable in contact maps; however, further increases in resolution result in the structures becoming less discernible (Extended Data Fig. 1a)”*.

Regarding the comparison of 10 and 200 bp data: *“The average contact maps were generated from Micro-C contact maps with a resolution of 10 bp. Using lower-resolution Micro-C maps for plotting the average map leads to more blurred features and thicker diagonal (Extended Data Fig. 1c,d)”*.

2b) The authors note that previous studies show CID boundaries are driven by transcriptional activity (Le et al, Science 2013; Le and Laub, EMBO J 2016; Liroy et al, Cell 2018)? How does the OPCID (an operon-sized region of high contact) lead to CID boundaries (a region of low contact)?

We propose that, in contrast to larger CIDs, OPCIDs may not necessarily be defined by “active” boundaries. Instead, an OPCID – especially a standalone one – might represent a superposition of transcriptionally-dependent loops, with boundaries marking DNA regions beyond which no such looping occurs, which could be referred to as “passive” boundaries. In the Discussion we note: *“OPCIDs likely represent a superposition of microloops and coiled-coil structures that develop within transcribed regions of individual cells. The high frequency of direct promoter-terminator interactions within an OPCID (Fig. 2g) is expected to facilitate the recycling of RNAP (Fig. 5c), thereby enhancing transcription efficiency”*.

Moreover, we suggest that the formation of certain OPCIDs may be supported by spatial contacts between CHINs localized at the boundaries of the OPCID. We further elaborate: *“spatial contacts between CHINs create DNA loops capable of isolating active operons from the rest of the genome (Fig. 3a)”*.

2c) In Bignaud, short-range “bundled domains” were associated with transcriptional activity, with these bundled domains interacting with each other to form a “plaid” pattern. Is the short-range bundle the same feature as an OPCID? A comparison should be made.

In the Discussion, we now state: *“Unlike previously described bundled domains¹¹, OPCIDs are characterized by typical contact domains (squares on contact maps) where all regions are in contact with one another...The relationship between OPCIDs and bundled domains¹¹ remains to be fully elucidated. Currently, we cannot rule out the*

possibility that the Hi-C protocol used in¹¹ may fail to detect all interactions within operons due to the use of detergents that could disrupt liquid condensates”.

2d) What is the evidence that the transcription start and end sites within OPCIDs make more frequent contact than other sites within the OPCID? Is looping detectable with published algorithms (Rao et al, Cell 2014, Salameh et al, Nat Comm, 2020, or others)?

In **Fig. 2g** of the revised version, we present evidence that the transcription start sites (TSS) and end sites (TES) within OPCIDs make more frequent contacts compared to the inner regions of the OPCID. However, we do not observe distinct looping contacts at the apex of OPCIDs, as seen at the apex of mammalian TADs. Therefore, we do not expect published loop-calling algorithms to be capable of unambiguously detecting TSS-TES loops.

2e) How was the co-localization of NAPs with CHINs assessed? In Fig S4, while HNS/StpA signal is found at nearly all CHINs displayed, the MukB signal is only found at a minority (~10%). Conversely, Fig S4 shows clear enrichment of Topo I, GyrA/GyrB, and moderately for Fis in the exact center of CHINs, akin to the profile of HupA/B. Why do the authors report MukB co-localization with CHINs and exclude these alternate possibilities? I also could not understand how the heatmaps in Fig S4 were sorted vertically. Is it by ChIP signal?

Yes, the heatmaps in **Extended Data Fig. 4** represent a pile-up of ChIP-seq signals sorted in descending order, with the higher entities at the top indicating highest ChIP-seq signals. The curve above the pile-up represents the average signal as specified in the figure legend.

The curve for H-NS shows two picks at the CHIN stems, with a dip in between (the larger the loop, the larger the dip). The MukB curve shows a similarly shape to that of H-NS curve, albeit less pronounced. In contrast, the curves for Topo I, GyrA/GyrB, and Fis resemble a mirror image of the H-NS curve, inverted. While H-NS shows overall enrichment over CHINs relative to the flanking areas, Topo I, GyrA/GyrB and Fis demonstrate depletion.

Just as H-NS shows reduced enrichment between the stems, TopoI, GyrA/GyrB and Fis also display less depletion in that region, manifesting as a micro peak at the center of the CHIN. However, even at this micro peak, the levels of Topo I, GyrA/GyrB and Fis remain lower than those at the CHIN flanks. Thus, we do not depict enrichment of these proteins over the CHINs; rather, our model illustrates their depletion (**Fig. 3d**, former **Fig. 4d**).

The only protein showing a clear enrichment at the CHIN center (i.e., within the CHIN loop) is HupA/B. The level of HupA/B in this region is higher than in both CHIN stems and flanking regions, aligning well with the established role of HupA/B in DNA bending, as reflected in our model (**Fig. 3d**, former **Fig. 4d**).

2f) Why are the “levels of H-NS” significantly higher at CHINs in an H-NS knockout strain? I also don't understand Fig S4M, which shows HNS binding in a Δ hns background. How is this possible? What is the significance test used here to determine the increased levels of HNS, StpA, and other NAPs and decreased levels of fis at CHINs in Δ hns?

We apologize for any confusion. In **Extended Fig. 4m** we analyzed the distribution of H-NS in *wild-type* cells concerning positions of those CHINs that became less or more pronounced in H-NS knockout. Due to the awkwardness and complexity of this analysis, we have excluded it from the revised version of the manuscript.

To investigate the role of other NAPs and DNA supercoiling in the assembly of CHINs (CHIDs), we produced Micro-C maps of *E. coli* strains where we deleted *fis* and *mukBEF*, as well as wild-type cells treated with inhibitors of gyrase and an agent that introduces single-stranded breaks in DNA. We found that none of these knockouts or treatments affected the integrity of CHINs (CHIDs) (**new Fig. 3m,n,p-r**).

2g) Overlapping/adjacent CHINs and NAPs could occur together or be mutually exclusive. In the model in 4D why are three CHIN contacts shown as a bundle when they could be mutually exclusive? It is also quite unlikely that all these NAPs would be interacting with CHINs together. 4D also appears to have supercoiling (GapR) upstream and topoisomerases downstream, with topo I and gyrase together which is not biologically probable.

We agree that complex spatial configuration depicted in **Fig. 3e** (former **4d**) of the revised version may represent superposition of several simpler configurations realized at the level of individual cells within a population. To acknowledge this possibility, we note in the revised version: “*Some of the other complex interaction patterns observed in our Micro-C maps may also reflect population averages*”. Without considering the existence of alternative configurations, it is difficult to explain the existence of CHIDs composed of overlapping CHINs. In the Discussion section, we state: “*It is worth noting that all CHINs within a CHID cannot coexist simultaneously at the same locus due to overlapping regions. Therefore, CHIDs represent a population average of alternative CHINs formed in individual cells*”.

This concept is further addressed in the legend to **Fig. 5b**, where we interpret the different contact patterns observed in Micro-C maps. We clarify: “*Distinction is made between ordinary loops and structures involving 3 or 4 paired DNA segments, such as triple-stem structures and CHIN contact, which represent overlapping or nested configurations*”.

Regarding the positioning of various factors upstream and downstream of CHINs, we acknowledge that initial **Fig. 4d** was confusing. Our intention was to show that topo I, gyrase and Fis are depleted at CHIN positions, whereas GapR colocalizes with these positions. However, these observations do not appear to be significant, as our revisions have demonstrated that neither DNA supercoiling nor Fis are critical for CHIN maintenance (**new Fig 3m,p-r**). Consequently, we have removed this labeling from **Fig. 3d** (former **4d**).

2h) The authors suggest that “spatial contacts between CHINs create DNA loops that could isolate active operons from the rest of the genome”. What is the evidence for this statement? Are CHINs associated with the boundaries of OPCIDs? This should be shown explicitly.

The spatial interactions between CHINs are not a common characteristic of active operons. Typically, OPCIDs are not surrounded by CHINs, and their formation does not depend on CHIN contact, as depicted in **Fig. 5c(i)**. However, we do see that some active operons are spatially confined within interacting CHINs, suggesting that CHIN contact may be one of the possible mechanisms of OPCID formation (shown in **Fig. 5c(ii)**). The isolation of two active operons via CHIN contact is illustrated in **Fig. 3a**. A similar pattern of active operon isolation can be seen in **Fig. 1b** with the active operon *kbl-tdh* (3790-3792 kb).

2i) An alternative explanation HNS “looping” could be that each locus bound by HNS becomes more dynamic within the cell, leading to increased interaction with other dynamic (i.e. HNS-bound) loci. This possibility should be acknowledged.

As requested, we now acknowledge the possibility of dynamic interactions between HNS-bound loci in the Discussion: “*It should be noted, however, that Micro-C analysis does not capture the frequency or duration of these contacts across individual cells; interactions between H-NS-bound loci may occur only in a subset of the population at any given time. Nevertheless, the existence of CHINs was independently validated using PLA, which confirmed that the CHIN base regions identified by Micro-C are indeed in close spatial proximity in a substantial proportion of cells (Extended Data Fig. 5a-d)*”.

3. some of the data has been overinterpreted. A more limited assessment of the data or additional biological experiments would improve the work

3a) Why do the authors propose that “H-NS is the top manager of CHINs”? I agree that some CHINs disappear in Δhns cells, but without knowing how deletions in other NAPs affect CHINs, a dominant role for HNS seems premature.

In the revised manuscript, we performed Micro-C analysis on several *E. coli* deletion mutants, including $\Delta stpA$ and $\Delta hns\Delta stpA$ double mutant. While the deletion of *stpA* gene did not impact the CHIN pattern, the deletion of both *stpA* and *hns* genes completely abolished all CHINs and their contacts (**new Fig. 3i-l, Fig. 4, Extended Data Fig. 8**). Considering that the single deletion of *hns* gene results in a profound reorganization and disassembly of many CHINs, we conclude that H-NS plays a major role in CHIN assembly, while StpA has a subsidiary yet important function.

We also constructed and analyzed ΔFis and $\Delta MukBEF$ strains. Based on the results obtained (**new Fig. 3m,n**), we conclude that these two factors do not play primary roles in CHIN or CHID formation and maintenance. All of the above results are discussed in the sections “H-NS is the top manager of CHINs” and “Disassembly of CHINs and

CHIDs leads to HTGs activation” of the revised manuscript. The conclusions regarding the potential participation of MukBEF in CHIN formation were removed from the manuscript.

3b) The authors show that CHIN-associated genes are more upregulated in Δhns but concluding that CHINs are responsible for silencing is premature. There is extensive literature about the role of HNS and other NAPs silencing HTGs (Ueguchi and Mizuno, EMBO J 1993; Amemiya et al, EMBO J 2021; and many others). Given that HNS bridging is temperature sensitive and high levels of HNS *in vitro* also weaken bridging (Kotlajich, et al, eLife 2015), an *in vivo* experiment disrupting bridging could clarify if HNS underlies CHIN formation and silencing.

In the revised manuscript, we provide additional functional links between the localization of HTGs within CHINs/CHIDs and their silencing. In the initial version, we showed that the knockout of H-NS results in the activation of HTGs residing within CHINs, while HTGs located outside CHINs remain largely unaffected.

In the revised manuscript, we show that knocking out both H-NS and StpA, which disrupts all CHINs, results in the activation of majority of HTGs, with levels of activation surpassing those observed in the single H-NS mutant (**new Fig. 4a,c**). Conversely, the single knockout of StpA, which does not affect CHIN integrity, does not lead to HTGs activation (**new Fig. 4d**). Accordingly, the double H-NS/StpA mutant strain displays a much more severe growth defect than the single H-NS mutant, whereas StpA-deficient cells do not show significant growth limitations (**new Fig. 4k**).

Remarkably, treatment with the antibiotic netropsin – a drug used to outcompete H-NS for binding to AT-rich sequences *in vitro* (Gordon et al., PNAS 2011) – effectively disrupts CHINs *in vivo*, as evidenced by our Micro-C analysis of netropsin-treated *E. coli* cells (**new Fig. 3o**). Moreover, netropsin treatment leads to HTG activation (**new Fig. 4b**). These new results underscore the importance of H-NS for CHIN formation and silencing.

Finally, on the examples of *yfjW* and *ygeH* HTGs, we directly show that hyperactivation of transcription correlates with the loss of CHINs, rather than with the absence of H-NS per se (**new Figs. 3i-l, 4i,j**). We state: “*In the case of ygeH, the CHID was reorganized but not fully disassembled in the Δhns strain, leading to a relatively modest (~5-fold) upregulation. However, complete disassembly of this CHID in $\Delta hns\Delta stpA$ cells resulted in a nearly 150-fold upregulation of ygeH and the formation of OPCID in place of CHID. Similarly, the HTG yfjW, which colocalizes with a CHIN in coordinate 2773-2775 kb (Fig. 3i), remains transcriptionally silent in Δhns cells (Supplementary Table 1) where this CHIN is preserved (Fig. 3j). However, its expression increases ~ 50-fold in $\Delta hns\Delta stpA$ cells (Supplementary Table 1) where the CHIN is replaced with an OPCID (Fig. 3l). These findings confirm that CHINs function as repressive structures and highlight that CHIN assembly, rather than H-NS deposition alone, is crucial for repression*”.

3c) As the authors note, Micro-C assays represent the population average organization. These events could be population-wide or may be restricted to a small subpopulation of

cells. Validating some of their contacts with an alternative experiment (e.g. fluorescence repressor operator system, in-situ hybridization, etc) could help clarify the likelihood of these events.

As noted in our response to Reviewer 1, the resolution of most fluorescent-based assays is insufficient to detect small chromosomal structures such as CHINs, whose median length does not exceed two kb (**Fig. 3d**). To independently verify the existence of CHINs *in vivo*, we employed a proximity ligation assay (PLA), a method widely used to study protein-protein interactions in cells (PMID 30238640). We targeted two His-tagged dCas9 molecules to the base regions of a long CHIN and assessed their spatial proximity using a standard PLA protocol. The assay revealed that in a substantial portion of cells, the two dCas9 molecules were indeed in close proximity (**new Extended Data Fig. 5a-d**).

In addition, we reconstituted CHIN structures *in vitro* by incubating purified H-NS protein with a DNA fragment known from Micro-C to form a CHIN (**new Extended Data Fig. 5e-j**). This resulted in the formation of CHIN-like structures of various lengths, further reinforcing the critical role of H-NS in CHIN formation.

The following sentences were added to the Results: *“To validate the existence of CHINs in vivo using an orthogonal approach, we performed proximity ligation assay (PLA) analysis in conjunction with super-resolution microscopy. Four genomic loci were selected for recruitment of dCas9-based PLA probes (Extended Data Fig. 5a,b). One probe (“C”) was positioned near the base of a long CHIN. A second probe (“D”) was placed on the opposite shoulder of CHIN, 6 kb away from probe “C”, such that both probes would be positioned across from each other within the CHIN stem when assembled. A third probe (“A”) was located 6 kb away from probe “C” but outside of the CHIN region, serving as a negative control. Finally, a fourth probe (“B”) was placed near probe “C” to serve as a positive control. We hypothesized that probes “C” and “D”, brought into proximity by CHIN folding, would yield a PLA signal, whereas probes “A” and “C”, 6 kb apart on linear DNA not involved in CHIN formation, would not. Probes “B” and “C”, positioned adjacently on DNA, were expected to yield a PLA signal regardless of 3D organization. The experimental results (Extended Data Fig. 5c,d) confirmed our predictions: while probes “A” and “C” produced only sparse PLA signals, probes “C” and “D” generated a robust signal in a substantial fraction of cells (~50% of the positive control). These results indicate that the CHIN is formed in at least half of the cells in the examined population within the timeframe of the PLA experiment”.*

And further: *“We also attempted to reconstitute CHINs in vitro by incubating a plasmid containing a DNA fragment known to form CHINs in vivo with purified H-NS. CHIN-like structures of varying lengths readily assembled on this fragment, as observed by transmission electron microscopy, but not on a control plasmid of similar length carrying a non-CHIN-forming DNA fragment, or on the CHIN-containing plasmid in the absence of H-NS (Extended Data Fig. 5g-j)”.*

The following additions were made to the Discussion: *“It should be noted, however, that Micro-C analysis does not capture the frequency or duration of these contacts across individual cells; interactions between H-NS-bound loci may occur only in a subset of the population at any given time. Nevertheless, the existence of CHINs was independently validated using PLA, which confirmed that the CHIN base regions identified*

by Micro-C are indeed in close spatial proximity in a substantial proportion of cells (Extended Data Fig. 5a-d)". And further: "*The central role of H-NS in CHIN formation was further supported by in vitro reconstitution experiments, in which CHIN-like structures readily assembled upon incubating purified H-NS with a plasmid containing a DNA fragment known to form CHIN in vivo (Extended Data Fig. 5g-j)*".

3d) The authors suggest that CHINs/CHIDs co-localizes distant HTGs to mediate recombination. However, chromosome contact frequency is not completely predictive of spatial distance and discrepancies can be large (Fudenberg et al, Nature Methods, 2017; Williamson et al, Genes Dev 2014). An in vivo experiment directly measuring co-localization is necessary to make these claims.

We agree and only suggest this possibility in the Discussion as a potential explanation for the unusually high frequency of recombination events between HTGs reported previously (Oliveira et al., Nat Commun 2017; Everitt et al., Nat Commun 2014). This topic will be an interesting subject for future investigation.

Minor notes:

1) The extensive literature documenting exclusion of HNS and other NAPs from active transcription units should be cited (Peters et al, Genes Dev, 2012; Singh et al, Genes Dev, 2014; Freddolino et al, PLOS Biology 2021; Amemiya et al, EMBO J 2021, and others).

Additional articles discussing the role of H-NS in repression have been cited in the revised manuscript, including those mentioned by the reviewer. However, articles addressing the role of other NAPs do not appear to be relevant to the focus of our paper.

2) CHIN boundaries "likely [arising] from the sharp difference GC content between the CHID and its flanking regions" is too strong of a statement. There could be many mechanisms (proteins, supercoiling, transcription, etc) that could influence CHIN boundaries, with GC potentially being a secondary consequence.

In the revised manuscript, we elaborated on our supposition by noting that both H-NS and StpA recognize AT-rich DNA sequences: "*These boundaries likely arise from sharp differences in GC content between the CHID and its flanking regions (Extended Data Fig. 4l). As both H-NS and StpA preferentially recognize AT-rich motifs, their dense binding should be confined to these AT-rich area*". This represents the simplest supposition, although it does not exclude the possibility of other mechanisms, which we have also noted in the revised version.

REFEREE #1 (REMARKS TO THE AUTHOR):

The authors have appropriately addressed all my concerns. Congratulations on a beautiful study.

We thank the reviewer for their generous and positive assessment of our work.

REFEREE #2 (REMARKS TO THE AUTHOR):

The authors have made significant efforts to address reviewers' comments experimentally or via textual changes. I still believe that the technical advance portrayed here is much welcome and will set a new standard of using Micro-C for bacterial 3D genomics, as I also still believe that experiments and analyses are of quite a high standard, and that the implications on HTGs are indeed interesting. At the same time though, I cannot fully steer away from the opinion that the conceptual advance, simply due to higher resolution structures, is not as large as presented here.

We thank the reviewer for their high evaluation of the technical quality of our study. We would like to take this opportunity to reiterate several key novel biological findings enabled by our methodologies, which may have been overlooked in the review:

1. We are the first to adopt the Micro-C procedure to study the 3D organization of the prokaryotic genome, achieving unprecedented resolution (10 bp). This allowed us to identify two distinct basic structures within bacterial genomes: OPCIDs, which support gene expression, and CHINs, which silence gene expression. Importantly, repressive spatial structures of any kind have not been reported previously in bacterial 3D genomes.
2. We identified the nucleoid proteins H-NS and StpA as key factors in the assembly and maintenance of CHINs. These DNA structures mediate the repression of horizontally acquired genes—many of which are toxic—thus supporting bacterial growth. Our newly developed PLA assay using gRNA-programmed dCas9 confirmed the existence of CHINs; this is the first application of PLA in bacteria and is likely to become a valuable tool for various research applications.
3. We demonstrate that interactions of CHINs organize the genome into loops of various sizes, which in some cases insulate active operons. This suggests a potential new layer of transcriptional regulation.
4. Our work provides the first genome-wide map of transcription-driven gene-loop structures, termed OPCIDs, in bacteria. This high-resolution approach reveals an unprecedented number of OPCIDs, establishing gene looping as a common feature of actively transcribed genes and operons, with the strongest signals at promoter-termination contacts. We further show that OPCID assembly and disassembly are highly dynamic: assembly occurs soon after gene induction (e.g., upon heat shock) and is restricted to actively transcribed operons, while disassembly happens rapidly once transcription ceases.
5. In experiments with rifampicin, we found that trapping RNAP at promoters is sufficient to disassemble OPCIDs. In stark contrast, CHINs remain remarkably stable under the same conditions. Moreover, factors such as DNA supercoiling and the SMC complex—previously thought to be critical in bacterial 3D genome organization—do not significantly influence OPCIDs or CHINs in our data.

To cite an example, OPCIDs are cited as structures that "might enable RNA Pol recycling" something that was proposed for promoter-terminator loops in human cells many years ago (e.g., work by the Proudfoot lab) without ever being really substantiated by hard evidence on such "recycling". A much simpler explanation would be that the authors are capturing variable conformations in single cells that are a result of variable polymerase positioning among the many alleles averaged out in their Micro-C data. Similarly, how various of these structures crosstalk with other "bundled domains", in other words how higher-res structures might underlie higher-order ones, is left unclear. Therefore, I remain borderline in my appraisal here, again, despite the high technical quality of the work.

We agree that OPCIDs may represent a population average and describe their nature precisely as suggested by the reviewer: *"OPCIDs likely represent a superposition of microloops and coiled-coil structures that develop within transcribed regions of individual cells."*

Nevertheless, the observed frequency of promoter-terminator contacts exceeds that within inner operon regions, particularly in highly expressed operons, which suggests a potential mechanism of RNAP recycling. Given that this remains a hypothesis (as it has been in human cells for many years), we have removed the mention of RNAP recycling from the abstract and revised the relevant sections of the manuscript as follows:

In the Results section:

"Furthermore, the intensity of TSS-TES contacts increases with higher transcription levels (Fig. 2g,h), suggesting a possibility of rapid RNAP recycling to support sustained high transcriptional output."

In the Discussion section:

"The high frequency of direct promoter-terminator interactions within an OPCID (Fig. 2g) might facilitate the recycling of RNAP (Fig. 5c), thereby enhancing transcription efficiency."

We also agree that the relationship between OPCIDs and the bundled domains observed by Bignaud et al. (2024) remains unclear. **We acknowledge this in the Discussion:**

"The relationship between OPCIDs and bundled domains¹¹ remains to be fully elucidated. Currently, we cannot exclude the possibility that the Hi-C protocol used in ref. 11 may fail to detect all interactions within operons due to the use of detergents, which could disrupt liquid condensates."

REFEREE #3 (REMARKS TO THE AUTHOR):

I commend the authors for their extensive efforts at addressing my earlier concerns. I find that the new experiments including: critical Micro-C controls, proximity ligation assay (PLA), data clarifying the role of different nucleoid associated proteins (NAPs) and DNA supercoiling on CHIN formation, and studies of H-NS assembly on CHIN DNA by electron microscopy greatly improve the work.

We thank the reviewer for their positive assessment of the additional experiments conducted during the revision of our manuscript.

1) Some of the conclusions remain unsupported by the data. There are two main issues:

a. First, there is no evidence that the looping of transcription start and end sites promotes RNAP recycling and high transcription output, which is claimed as a major

finding. Although looping is observed more frequently at highly expressed operons (Figure 2G), this does not mean that looping causes high expression. Looping may be a consequence of expression, due to transcription-dependent twin supercoiling domains or supercoiling-recruited nucleoid associated proteins “looping” DNA together. This interpretation is supported by the data showing enrichment of Hi-C contacts between the upstream and downstream regions of a transcription unit (Bignaud et al, 2024). All statements about RNAP recycling should be conservatively amended given this likely alternative possibility.

We fully agree that the possibility of RNAP recycling from TES to TSS is currently just a hypothesis. Therefore, we have removed the mention of RNAP recycling from the abstract and modified the manuscript text as follows:

In the Results section:

“Furthermore, the intensity of TSS-TES contacts increases with higher transcription levels (Fig. 2g,h), suggesting a possibility of rapid RNAP recycling to support sustained high transcriptional output.”

In the Discussion section:

“The high frequency of direct promoter-terminator interactions within an OPCID (Fig. 2g) might facilitate the recycling of RNAP (Fig. 5c), thereby enhancing transcription efficiency.”

Regarding the potential contribution of DNA supercoiling in the formation of OPCIDs, we would like to note that OPCIDs are generally resistant to bleomycin treatment, which is known to alleviate supercoiling. Supporting examples are provided below and in the new Extended Data Fig. 3j.

Given that OPCIDs appear resistant to bleomycin treatment—which introduces single-strand breaks and releases DNA supercoiling—it is unlikely that OPCIDs are merely supercoiled domains, as their structural integrity would likely be compromised if that were the case. **This is now acknowledged in the Discussion section:** *“Surprisingly, the integrity of OPCID was not compromised by bleomycin, a drug that introduces*

single-strand breaks in DNA and thus releases DNA supercoiling (Extended Data Fig. 3j)."

Regarding the results of Bignaud et al. (2024), we would like to note that these authors did not demonstrate preferential contacts between TESs and TSSs. Instead, their data showed preferential interactions across entire highly transcribed operons (manifested as plaid-like patterns on Hi-C maps) and short-range interactions along the transcription units, leading to the formation of bundled domains observed in their Hi-C maps.

b. Second, there is no evidence that CHINs facilitate "the spatial proximity of distant HTGs...[explaining] the previously observed high frequency of recombination between these genes, both in cis and trans". No evidence is shown about association frequency of distant HTGs let alone if CHINs mediate this association. Indeed, CHINs contact is shown to be generally ~2-8 kb on average, suggesting any that CHIN-mediated recombination is restricted to within a gene or a local region. Furthermore, the diagram for this recombination (Figure 5b) again does not show distant contacts. All statements about recombination should be amended to indicate that this is a short-range phenomenon.

We agree that our data show CHINs interact over relatively short distances (<40 kb, see Fig. 3e, right boxplot). To clarify this, we deleted the word "*distant*" and rephrased the sentence as follows:

"The ability of CHINs to stick together may facilitate recombination between HTGs both in cis and in trans (Fig. 5c), explaining the previously observed high recombination frequency between these genes^{62,63} and suggesting an elegant mechanism that accelerates bacterial evolution."

Furthermore, the propensity of CHINs to stick together may create favorable conditions for recombination between exogenous HTG (e.g. mobile elements on plasmids) containing CHINs with host genomic DNA. Of course, this remains a hypothesis. In the revised manuscript, we limited it to the Discussion section, as quoted above.

2) I like the new analysis presented in Figure 2G to distinguish between looping of transcription start/end sites and looping within a transcription unit. However, the triangle representing sites in the experiment is difficult to interpret. Can the authors use a linear representation like in Figure 2F instead?

We thank the reviewer for this suggestion. The figure has been modified accordingly.

3) I still am not satisfied as to why the authors suggest MukB and positive supercoiling (as inferred by GapR binding) is associated with CHINs. In Figure S2C, it appears that MukB is associated with ~10-20% of CHINs and the remaining >80% of CHINs have a MukB profile akin to that for active and inactive operons. Similarly, in Figure S2F, it appears that GapR is associated with perhaps 5% of CHINs at best. In both these cases, it seems much more likely that these profiles are random chance or that there is an association with transcription of CHINs rather than the CHIN itself.

The possibility that the MukB and GapR profiles arise from random chance seems unlikely to us, as both profiles—though not as prominent as those for H-NS and StpA—still exhibit narrow standard errors and display a definite shape similar to that of H-NS and StpA. Also, the idea that MukB and GapR profiles result from transcription of CHINs

appears unlikely, since all CHINs are repressive structures, and genes organized within CHINs are not transcribed (Fig. 2b, Extended Data Fig. 4b).

At the same time, we agree that only a fraction of CHINs is bound by these proteins. To acknowledge this, we revised the text as follows: *“In addition to H-NS and StpA, we found that MukB, a bacterial SMC protein⁵¹, also colocalizes with the arms of CHINs (Fig. 3a), although only a fraction of CHIN arms demonstrates prominent association with this protein (Extended Data Fig. 4c).”*

and

“However, the prominent deposition of GapR is observed only in a small fraction of CHINs, suggesting that the level of supercoiling within CHIN stems is not uniform (Extended Data Fig. 4h).”

Point-by-point response to the editor's queries from the email

1, Please present titles and legends of supplementary tables in a PDF entitled "Supplementary Information". This file should encompass all supplementary data, including items like the raw blot referenced in the next point, as long as they can be formatted into a PDF.

Done

2, Please include the unprocessed images of blots/gels, as presented in main and extended data figures, in Supplementary Figure 1.

Done

3, Please add the statement on competing interests in the manuscript.

Done

4, Please note that the (GEO) accession no. (GSE272161) referenced in the manuscript is currently private and should be released ahead of the formal acceptance.

We have made the data publicly accessible.

5, The manuscript is not in .docx format. Currently it is in pdf format. Please organize the sections in the main text file in following order: Title, Authors & Affiliations, Main text (abstract/introduction/results/discussion) / Main text references / Main figure legends / Methods / Additional references associated with methods / Acknowledgements / Author Contributions / Competing Interests / Extended figure legends.

The manuscript has been organized accordingly and saved in .doc format.

6, Please reduce the Abstract to 230 words or less. Currently there are 284 words.

The abstract has been reduced to 235 words.

7, References only associated with the Methods should be listed at the end of the Methods section, and should be numbered sequentially to the references of the main text, rather than restarting at 1.

The references were arranged accordingly.

8, Please remove the main figures from the article file and re-supply them individually in an acceptable format such as EPS, AI, PS, PDF, PPT, PSD or XLS (for graphs) with editable vector files.

The main figures were removed from the article file and supplied as individual PDFs.

9, Please ensure all main figure legends are 300 words or less.

Confirmed.

10, Please provide a Microsoft Word file entitled “SI Guide”, containing a cover page with manuscript title and author information and a table of contents for supplementary files (preferably with page numbers).

The file “SI Guide” has been provided.

11, Please pay attention to the editorial policy regarding third-party permissions (<https://www.springernature.com/gp/policies/editorial-policies/third-party-permissions>). Please check sources or if permissions are needed for ‘third party content’ in the figures.

No permissions are needed for ‘third party content’ in the figures.

12, Please remove the Extended data figures from the article file and re-supply them individually in EPS, JPEG or TIF format.

The Extended data figures are supplied as individual JPEGs.

Point-by-point response to the “editorial requests” from the word file

2.1 Please note that this information is missing in the legend of figure 4k.

N and SD have been provided in the legend of figure 4k. The following text was added: “*Data are presented as mean +/-SD from n=7 biological replicates for wild-type, $\Delta stpa$, and Δhns , and n=14 for $\Delta hns\Delta stpa$ ”.*

2.2 Please provide a precise value of ‘n’ in the legends of figure 4i; extended data figures 5d,5j.

We provide N in the figures 4i, 5d, and 5j.

2.3 Although ‘n’ is provided, please describe the nature of entity for ‘n’ in the legends of figures 2g; 3d,3e; 4e-h.

The following sentences were added:

To the legend of figures 2g: “*N represents the number of operons in each group*”.

To the legend of figures 3d: “*n represents the number of CHINs assessed*”.

To the legend of figures 3e: “*n represents the number of CHIN contacts assessed*”.

To the legend of figures 4e-h: “*N represents the number of genes*”.

3.1 Please note that the error bars need to be defined in the legend of figure 4k.

The error bars are defined in the legend of figure 4k. The following text was added: “*Data are presented as mean +/-SD from n=7 biological replicates for wild-type, $\Delta stpa$, and Δhns , and n=14 for $\Delta hns\Delta stpa$* ”.

3.2 Please note that the measure of centre for the error bars needs to be defined in the legends of figure 4i; extended data figures 5d,5j.

The following sentences were added:

To the legend of figure 4i: “*Data are presented as mean +/-SD*”.

To the legend of extended data figure 5d: “*Data are presented as mean values from two biological replicates*”.

To the legend of extended data figure 5j: “*Data are presented as mean values from two independent experiments*”.

3.3. Please note that the error bands need to be defined in the legend of figure 2h.

In the legend of figure 2h we clarify now: “*Black line shows linear regression, shaded area – 95% CI*”.

4.1 Please indicate the statistical test used for data analysis and where appropriate, please specify whether it was one-sided or two-sided and whether adjustments were made for multiple comparisons, in the legends of figures 3g,3h; 4a-d.

We now indicate:

In the legend of figures 3g,3h: “*P-values were calculated using two-sided shuffle test*”.

In the legend of figures 4a-d: “*P-values were calculated using two-sided Wald test and adjusted for multiple testing using Benjamini-Hochberg correction with FDR set to 0.05*”.

4.2 Please note that the exact p value should be provided, when possible, in the legends of figures 2h; 3g,3h.

We prefer to report the approximate p-values for the analyses shown in figures 2h, 3g, 3h.

4.3 Please indicate what ‘*/**’/‘***’/‘****’/‘*****’ represents; if this represents p values, please specify whether adjustments were made for multiple comparisons and indicate the exact p value in the legends of figures 2g; 4e-h.

We indicated what ‘*/**’/‘***’/‘****’/‘*****’ represent in the legend of figures 2g and 4e-h. We prefer to report the approximate p-values for these analyses.

5. Reproducibility. Please note that this information is missing in the legends of figure 1a; extended data figures 9f,9g.

The following corrections have been made:

To the legend of figure 1a we added the following sentence: “*All Micro-C experiments were tested to show similar separation patterns*”.

In the legend of extended data figure 9f we now specify that the shown separation pattern is typical of all MNase titration experiments: “*Gel shows a typical MNase titration experiment*”.

Regarding figure 9h, no clarification is needed as the figure show the results of two replicates

6. Data availability. Please ensure that datasets deposited in public repositories are now publicly accessible, and that accession codes or DOI are provided in the "Data Availability" section.

We have made the data publicly accessible. We confirm that accession codes are provided in the “Data Availability” section.

7.1 Please note that units of molecular weight markers are missing for extended data figures 9f-h.

Missing units of molecular weight markers were added to extended data figures 9f-h as well as to Supplementary Figure 1 with uncropped gel images.

8. Micrographs: Please ensure that all micrographs include a scale bar and this scale bar is defined on the panels or in the figure legends.

Confirmed

9. Please note that for extended data figure 4 the subfigures (E.g: a,b,etc.) are not defined in the legend. Please rectify this.

The legend of extended data figure 4 was modified to define subfigures as follows: “*Distribution of proteins (a-k) and GC content (l) around different genomic features in wild-type E. coli cells.*”